# What One Cannot, Two Can: Two-Layer Transformers Provably Represent Induction Heads on Any-Order Markov Chains

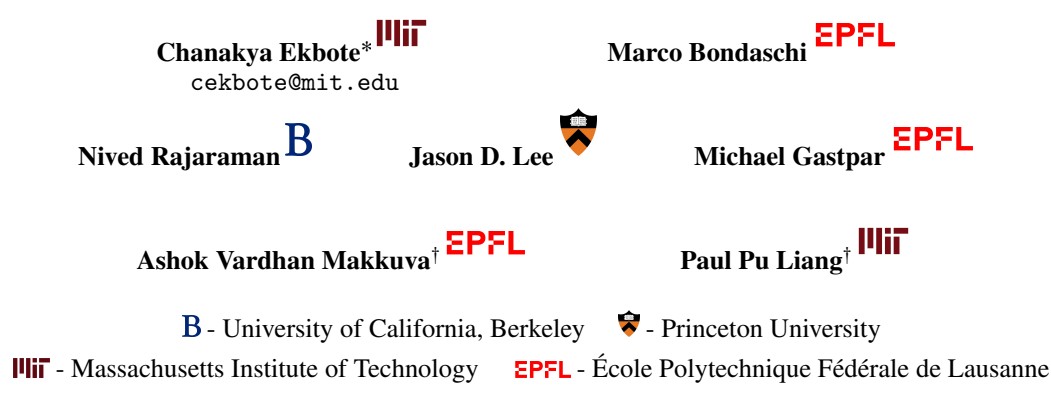

**Chanakya Ekbote*** MIT
cekbote@mit.edu

**Marco Bondaschi** EPFL

**Nived Rajaraman** B

**Jason D. Lee** 

**Michael Gastpar** EPFL

**Ashok Vardhan Makkuva**[†] EPFL

**Paul Pu Liang**[†] MIT

B - University of California, Berkeley ♟ - Princeton University

MIT - Massachusetts Institute of Technology EPFL - École Polytechnique Fédérale de Lausanne

## Abstract

In-context learning (ICL) is a hallmark capability of transformers, through which trained models learn to adapt to new tasks by leveraging information from the input context. Prior work has shown that ICL emerges in transformers due to the presence of special circuits called induction heads. Given the equivalence between induction heads and conditional $k$-grams, a recent line of work modeling sequential inputs as Markov processes has revealed the fundamental impact of model depth on its ICL capabilities: while a two-layer transformer can efficiently represent a conditional 1-gram model, its single-layer counterpart cannot solve the task unless it is exponentially large. However, for higher order Markov sources, the best known constructions require at least three layers (each with a single attention head) — leaving open the question: *can a two-layer single-head transformer represent any $k^{th}$-order Markov process?* In this paper, we precisely address this and theoretically show that a two-layer transformer with one head per layer can indeed represent any conditional $k$-gram. Thus, our result provides the tightest known characterization of the interplay between transformer depth and Markov order for ICL. Building on this, we further analyze the learning dynamics of our two-layer construction, focusing on a simplified variant for first-order Markov chains, illustrating how effective in-context representations emerge during training. Together, these results deepen our current understanding of transformer-based ICL and illustrate how even shallow architectures can surprisingly exhibit strong ICL capabilities on structured sequence modeling tasks. Code is available at the 🔗 link.

## 1 Introduction

> *"A complex system that works is invariably found to have evolved from a simple system that worked."*
> — *John Gall, Systematics (1975)*

Transformers, powered by the attention mechanism, have emerged as the dominant architecture in machine learning, achieving state-of-the-art performance across a wide range of domains, including

---

*Corresponding author.

[†]Equal contribution / equal mentorship. Authors listed alphabetically; either may be considered last author.

39th Conference on Neural Information Processing Systems (NeurIPS 2025).

natural language processing [6], computer vision [32], and complex reasoning tasks [23, 36]. A key factor underpinning this success is their ability to efficiently model sequences and perform in-context learning (ICL)—adapting to unseen tasks during inference by leveraging the relevant input context [22]. It is well known that ICL emerges in transformers due to the presence of special circuits called *induction heads* [22, 31].

Intuitively, these circuits enable transformers to implement a "copy-and-match" mechanism by copying earlier tokens from the input and matching them with the desired context for next-token prediction. For example, a first-order induction head mimics the functionality $[\ldots, A, B, \ldots, A] \to B$. Capitalizing on the connection between induction heads and conditional $k$-gram models, a recent active line of work has leveraged $k^{\text{th}}$-order Markov chains as a simple yet powerful framework to analyze how transformers learn induction heads, with the framework referred to as Markov-ICL [10, 20, 26, 7, 5]. In particular, using Markovian input sequences, the goal is to understand how transformers learn to represent conditional $k$-grams–equivalent to $k^{\text{th}}$-order induction heads (Def. 2).

Building upon the Markov-ICL setting, Bietti et al. [5], Edelman et al. [10], and Nichani et al. [20] demonstrate that a conditional $1$-gram model can be efficiently represented by a two-layer (i.e., two attention layers), single-head transformer. In contrast, Sanford et al. [29] show that a one-layer, single-head counterpart cannot solve this task unless its hidden dimension is exponentially larger than that of the two-layer model. On the other hand, for higher-order processes, Edelman et al. [10] and Nichani et al. [20] argue that low-depth transformers need the number of heads scaling linearly in $k$, in order to learn $k^{\text{th}}$-order Markov processes. However, Rajaraman et al. [26] establish a surprising result that a three-layer, single-head transformer can represent the conditional $k$-gram model for any $k \geqslant 1$, and thereby learn $k^{\text{th}}$-order Markov models in-context.

Together, these findings illustrate that while a $1$-layer transformer cannot efficiently represent an induction head, a $3$-layer model with $1$ head per layer suffices for all Markov orders $k \geqslant 1$. However, these results leave open a natural and important question:

> **Can a two-layer, single-head transformer learn $k^{\text{th}}$-order Markov processes in-context?**

In this paper, we approach this question from two perspectives—representational power and learning dynamics. In particular, we affirmatively answer this on the representation front and make partial progress on the learning dynamics through the following key contributions:

> **Main Contributions.**
>
> 1. Improving upon the best known three-layer construction, we prove that a two-layer single-head transformer is sufficient to represent the conditional $k$-gram model, and thus can learn $k^{\text{th}}$-order Markov chains in-context (Sec. 4). To the best of our knowledge, this is the tightest characterization of transformer depth and Markov order for Markov-ICL.
>
> 2. In the course of establishing this result, we uncover an interesting tradeoff between depth and width: the existing three-layer single-head construction (depth is three, width is one) is equivalent to a two-layer architecture with two heads in the first layer and one in the second (depth is two, width is two). (Sec. 3)
>
> 3. For first-order Markov chains, we prove that gradient descent on our simplified two-layer transformer learns an induction head, and thereby the in-context conditional empirical distribution. (Sec. 5)

**Comparison with other Markov-ICL works vis-a-vis transformer architecture.** In this paper, as we focus on higher-order processes, we consider the general transformer architecture (Subsec. 2.2) closer to real-world models with relative positional encodings [30], softmax attention [32], MLPs, and layer normalization [2]. For first-order processes, past literature has considered simplified variants: Bietti et al. [5] use frozen and fixed random embeddings with no MLPs in the first layer, Edelman et al. [10] consider attention-only models with no MLPs and embeddings, and Nichani et al. [20] study a variant called *disentangled transformer*, splitting the residual and attention streams, with the hidden dimension scaling with depth. While our gradient descent result for first-order processes

(Sec. 5) is similar to that of Nichani et al. [20], the architecture is fundamentally different. In Tab. 1, we compare the parameter counts of our construction against alternative constructions. Our results show that, as both the order and sequence length increase, our architecture remains the most compact in terms of parameter efficiency.

| Previous works | Markov order | Architecture | # Parameters |
|---|---|---|---|
| Bietti et al. [5] | 1 | 2-layer, 1-head | $9d^2 + dS + Td$ |
| Edelman et al. [10] | 1 | 2-layer, 1-head | $21S^2 + 3TS$ |
| Nichani et al. [20] | $k$ | 2-layer, $k$-head | $(k + (k+1)^2)(S+T)^2 + T(S+T)$ |
| Rajaraman et al. [26] | $k$ | 3-layer, 1-head | $15(6S+3)^2 + (6S+3)(3T+2S+10)$ |
| **Ours** | $k$ | 2-layer, 1-head | $9(6S+3)^2 + (6S+3)(2T+2S+9)$ |

Table 1: Comparison with prior works on transformers with Markov-ICL. Here $k \geqslant 1$ is the Markov order, $T$ the sequence length, and $S$ the state-space (vocabulary) size. Note that Bietti et al. [5] do not explicitly state what $d$ is and assume it to be large enough. To the best of our knowledge, our work obtains the tightest known characterization of transformer depth and Markov order for higher-order processes. The bit precision for our representation results is the same as that of Rajaraman et al. [26]: it suffices to have $\Omega(\log(T) + k)$ bits per parameter, with $\mathcal{O}(1/T)$ additive approximation error. A detailed breakdown of the parameter count computations can be found in App. I. Additional experiments analyzing the effect of varying bit precisions are presented in Subsec. H.4.

**Notation.** Scalars are denoted by lowercase italic letters (e.g., $x, y$), vectors by lowercase bold letters (e.g., $\mathbf{x}, \mathbf{y}$), and matrices by uppercase bold letters (e.g., $\mathbf{A}, \mathbf{B}$). The matrices $\mathbf{0}_{m \times n}$ and $\mathbf{1}_{m \times n}$ represent the all-zeros and all-ones matrices in $\mathbb{R}^{m \times n}$, respectively. For norms and inner products we use standard notation, i.e., norms are written as $|| \cdot ||$, with $||\mathbf{x}||_2$ denoting the Euclidean norm. The inner product in Euclidean space between two vectors $\mathbf{x}$ and $\mathbf{y}$ is denoted by $\langle \mathbf{x}, \mathbf{y} \rangle$. The indicator function is denoted either by $\mathbb{I}(\cdot)$ or $\mathbb{1}_{(\cdot)}$, depending on the context. The vocabulary $\mathcal{S} = \{1, 2, \ldots, S\}$ of cardinality $|\mathcal{S}| = S$ denotes the finite state-space of the Markov Chains studied in this paper. A sequence $x_{0:T}$ is defined as $x_{0:T} := x_0, x_1, \ldots, x_T$, where $T \in \mathbb{N}$. For any $x \in \mathcal{S}$, its one-hot embedding is denoted by $e_x^S \in \mathbb{R}^S$. $\mathcal{O}(\cdot)$ denotes the big O notation.

## 1.1 Related Work

In recent years, transformers have been widely studied from both theoretical and mechanistic perspectives [35, 14, 19]. Foundational results [37, 25, 33] have established their universality and Turing-completeness, with further work examining their ability to model formal languages [4, 16] and implement algorithmic behaviors such as induction heads [22]. Concurrently, recent studies have explored how transformers perform ICL, including their learning dynamics and phase transitions [31, 34, 3, 8, 27, 3, 1, 11, 15]. On this front, our work is most closely related to recent works studying Markov-ICL, where the sequential inputs are modeled as Markov processes. In Bietti et al. [5] and Edelman et al. [10], the authors discover a stage-wise learning procedure for first-order sources, and show that a 2-layer, 1-head transformer can represent a first-order induction head. Makkuva et al. [17, 18] study the optimization landscape and learning dynamics of a 1-layer, 1-head model for a first-order global Markov chain, unveiling the significance of weight-tying and Markov switching probabilities. Nichani et al. [20] studies the learning dynamics of 2-layer, 1-head disentangled transformer, illustrating it learns to implement a first-order induction head. For higher-order sources, Nichani et al. [20] constructs a 2-layer, $k$-head architecture, whereas Rajaraman et al. [26] shows that a 3-layer, 1-head model suffices to represent a conditional $k$-gram model. In contrast, we show that a 2-layer, 1-head transformer suffices to represent arbitrary-order Markov models in-context.

## 2 Background

In this section, we provide the requisite preliminaries on Markov processes, the conditional $k$-gram model, and the Transformer.

## 2.1 Markov Processes and the Conditional $k$-gram Model

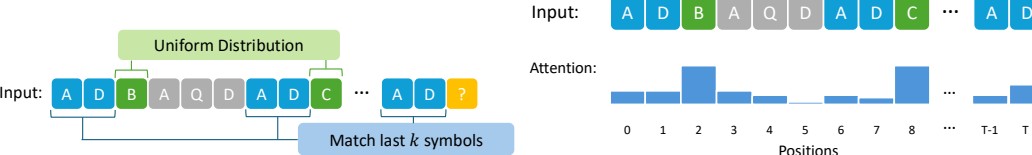

(a) The conditional $k$-gram model (Def. 2). It first (1) identifies the positions in the sequence where the preceding $k$ tokens match the current context (blue), and (2) returns the empirical distribution over symbols (green) that follow these matched positions.

(b) $k^{\text{th}}$-order induction head (Def. 4) The attention pattern concentrates on positions in the sequence where the preceding $k$ tokens exactly match the final $k$ tokens. These are the positions the model attends to most strongly, as they reflect the same context as the current prediction target.

A $k^{\text{th}}$-order Markov process is a time-homogeneous stochastic process over a finite state space, where the probability of the next symbol depends only on the most recent $k$ symbols in the sequence [21]. More formally, it is defined as follows:

> **Definition 1.** ($k^{\text{th}}$-order Markov Chains) A stochastic process $x_{0:T} := x_0, x_1, \cdots, x_T$ over a finite state space $\mathcal{S}$ is said to be a $k^{\text{th}}$-order Markov Chain with transition kernel $\pi(\cdot)$ if, for all $k \leqslant n \leqslant T$, the following condition holds:
>
> $$\pi(x_{n+1} = s \mid x_n, x_{n-1}, \ldots, x_0) = \pi(x_{n+1} = s \mid x_n, x_{n-1}, \ldots, x_{n-k+1}), \quad \forall s \in \mathcal{S}.$$

We now define the empirical equivalent of the kernel $\pi$, the *conditional $k$-gram model*:

> **Definition 2.** (Conditional $k$-gram model) Given a sequence $x_{0:T} := x_0, x_1, \cdots, x_T$ in $\mathcal{S}^{T+1}$, the conditional $k$-gram model $\widehat{\pi}_k(s \mid x_T, x_{T-1}, \ldots, x_0)$ computes the empirical probability of observing a symbol $s \in \mathcal{S}$ matching the last $k$ tokens, i.e.
>
> $$\widehat{\pi}_k(s \mid x_0, x_1, \cdots, x_T) := \frac{\sum_{i=k}^{T} \mathbb{I}(x_i = s, \, x_{i-1} = x_T, \, x_{i-2} = x_{T-1}, \ldots, \, x_{i-k} = x_{T-k+1})}{\sum_{i=k}^{T} \mathbb{I}(x_{i-1} = x_T, \, x_{i-2} = x_{T-1}, \ldots, \, x_{i-k} = x_{T-k+1})}$$

assuming the denominator is non-zero. Thus conditional $k$-gram can be interpreted as a simple count-based estimator of the transition kernel $\pi$, using the in-context count estimate for the predictive probability distribution.

**Data generation (Random Markov sequences).** To empirically generate the input data, i.e. random Markov sequences of order $k$, we first sample each row of the transition matrix $\pi$ independently from a Dirichlet prior with parameter vector $\mathbf{1}$, which is equivalent to an uniform distribution on the $S$-dimensional probability simplex. The initial $k$ tokens of each sequence are sampled uniformly from $\mathcal{S}^k$, and the remaining tokens are generated according to the sampled kernel $\pi$. To sample an input batch, we first sample multiple independent transition kernels, generate sequences from each, and then aggregate these sequences to form a batch for training or evaluation.

## 2.2 Transformer Model

While there exist several attention-based transformer architectures in literature, we adopt the formulation described in Shaw et al. [30] and Dai et al. [9], which employs relative positional encodings. Each layer in our transformer comprises softmax-based self-attention, optionally followed by a multilayer perceptron (MLP). Following the construction outlined in Rajaraman et al. [26], we define a $L$-layer transformer with multi-headed attention as follows:

**Definition 3.** (Multi-head Attention Transformer)

**Require:** Input $x_{0:T}$, number of layers $L$, number of attention heads per layer $H_\ell$

**Ensure:** Output distribution $P_\theta(\cdot \mid x_{0:T})$

$\quad \mathbf{x}_n^{(1)} \leftarrow \texttt{Emb}(x_n)$ for $n = 0, 1, \ldots, T$

$\quad$ **for** $\ell = 1$ to $L$ **do**

$\quad\quad$ **for** $n = 0$ to $T$ **do**

$\quad\quad\quad$ **for** $h = 1$ to $H_\ell$ **do**

$\quad\quad\quad\quad \widetilde{\mathbf{x}}_n^{(\ell,\texttt{head}:h)} := \sum_{i=0}^{n} \texttt{att}_{n,i}^{(\ell,\texttt{head}:h)} \cdot \left( \mathbf{W}_V^{(\ell,\texttt{head}:h)} \mathbf{x}_i^{(\ell)} + \mathbf{p}_{n-i}^{(\ell,\texttt{head}:h,V)} \right)$

$\quad\quad\quad$ **end for**

$\quad\quad\quad \widetilde{\mathbf{x}}_n^{(\ell+1)} := \mathbf{x}_n^{(\ell)} + \sum_{h=1}^{H_\ell} \widetilde{\mathbf{x}}_n^{(\ell,\texttt{head}:h)}$ $\qquad\qquad\qquad$ $\triangleright$ Residual connection

$\quad\quad\quad \mathbf{x}_n^{(\ell+1)} := \texttt{MLP}(\mathbf{x}_n^{(\ell)}, \widetilde{\mathbf{x}}_n^{(\ell+1)})$ $\triangleright$ MLP with layer normalization and skip connections

$\quad\quad$ **end for**

$\quad$ **end for**

$\quad \text{logits}_T := \mathbf{W}_o \mathbf{x}_T^{(L+1)} + b_o$

$\quad P_\theta(\cdot \mid x_{0:T}) := f(\text{logits}_T),$

Where the attention weights are defined as:

$$\texttt{att}_{n,i}^{(\ell,\texttt{head}:h)} := \texttt{softmax}_i \left( \left\langle \mathbf{W}_K^{(\ell,\texttt{head}:h)} \mathbf{x}_i^{(\ell)} + \mathbf{p}_{n-i}^{(\ell,\texttt{head}:h,K)}, \ \mathbf{W}_Q^{(\ell,\texttt{head}:h)} \mathbf{x}_i^{(\ell)} \right\rangle \right).$$

Here $\ell$ denotes the layer index and $h$ the index of the attention head within a layer. The matrices $\mathbf{W}_Q^{(\ell,\texttt{head}:h)}, \mathbf{W}_K^{(\ell,\texttt{head}:h)}, \mathbf{W}_V^{(\ell,\texttt{head}:h)} \in \mathbb{R}^{d \times d}$ are the query, key, and value projection matrices, respectively The vectors $\mathbf{p}_{n-i}^{(\ell,\texttt{head}:h,K)}$ and $\mathbf{p}_{n-i}^{(\ell,\texttt{head}:h,V)} \in \mathbb{R}^d$ represent the relative positional encodings that modulate attention based on token distance. Finally, the output of the transformer is projected using $\mathbf{W}_o \mathbf{x}_T^{(L+1)} + b_o \in \mathbb{R}^S$, mapping the representation from the embedding dimension $d$ to the output vocabulary size $S$. Although a softmax is typically used for output normalization, in our setup we define $f(\cdot) = \texttt{ReLU}(\cdot)$. Note that the attention mechanism described is causal self-attention.

# 3  Warm Up: Construction with Two Heads in Layer One, One in Layer Two

As a warm up, in this section, we prove our first main result that we can represent $k^{\text{th}}$-order Markov processes with a 2-layer, 2-head transformer. Towards the same, we recall that the $k^{\text{th}}$-order in-context counting estimator, i.e. conditional $k$-gram model, defined in Subsec. 2.1 is closely related to the classical Laplacian smoothing, which in turn is the optimal Bayes estimator for the next-token predictive distribution [12, 28]. In view of this fact, in order to estimate the optimal conditional distribution of each token in-context, it is natural to ask: *how does a transformer represent this conditional $k$-gram model?* To this end, Rajaraman et al. [26] introduces the notion of a *$k$-th order induction head*. This mechanism enables the attention layer of a transformer to assign at each time step the highest attention weight to past tokens whose length-$k$ context matches that of the current token. For the sake of completeness, we recall the formal definition.

**Definition 4.** ($k^{\text{th}}$-order induction head) An attention layer with a single head is said to implement a $k^{\text{th}}$ order induction head if, for any input sequence $(x_0, x_1, \ldots, x_T) \in \mathcal{S}^{T+1}$, and for any fixed index $i \leqslant T$, the attention score $\texttt{att}_{T,i}$ at position $i$ is maximized if and only if its preceding $k$ tokens exactly match the final $k$ tokens of the sequence. That is, $\texttt{att}_{T,i}$ is maximized if and only if $x_{i-j} = x_{T-j+1}$ for all $i \in \{1, 2, \ldots, k\}$.

The salience of this higher-order induction head is immediately reflected by the fact that such a mechanism is indeed what is precisely needed to implement the conditional $k$-gram estimator from Def. 2. Capitalizing on this, Rajaraman et al. [26] proves the best known result for higher-order Markov processes, through a 3-layer, 1-head transformer architecture, which we recall below to motivate our main result:

> **Theorem 1.** (Theorem 4 in Rajaraman et al. [26]): The conditional $k$-gram model can be represented using a transformer with three layers, each containing a single attention head. The layers are separated by MLP blocks and include relative positional encodings as well as layer normalization. The embedding dimension scales as $\mathcal{O}(S)$.

*Proof sketch.* We provide a brief outline of the proof of Thm. 1. For each position $n$ in the sequence $s_{0:T}$, the first attention layer effectively learns a hard attention map concentrated on the indices $\{n, n-1, \ldots, n-k+1\}$. Specifically, within a subspace of its embedding space, this layer computes $\mathbf{u}_n = \left( \sum_{i=0}^{k-1} 3^i \cdot e_{x_{n-i}}^S \right) / \left( \sum_{i=0}^{k-1} 3^i \right)$, with attention weights $\text{attn}_{n,i} = 3^i / \left( \sum_{j=0}^{k-1} 3^j \right)$ for $i \in \{n, n-1, \ldots, n-k+1\}$, and zero otherwise. Intuitively, this $\mathbf{u}_n$ allows to capture the past-$k$ context starting at position $n$. Similarly, the second attention layer focuses on the positions $\{n-1, n-2, \ldots, n-k\}$, and computes $\mathbf{v}_n = \left( \sum_{i=1}^{k} 3^i \cdot e_{x_{n-i}}^S \right) / \left( \sum_{i=1}^{k} 3^i \right)$, capturing the context shifted by one position. After these two layers, the embedding at position $n$ contains both $\mathbf{v}_n/\|\mathbf{v}_n\|_2$ and the normalized vector $\mathbf{u}_n/\|\mathbf{u}_n\|_2$, assuming appropriate choices of embedding dimensions, query/key/value matrices, positional encodings, and MLP layers with layer normalization. In particular, the architecture can be configured to compute the $\ell_2$-norm via layer norm, as detailed in Subsec. A.1. The third attention layer then functions as a $k$-th order induction head: at position $n$, the attention score is made proportional to the cosine similarity $\langle \mathbf{u}_T, \mathbf{v}_n \rangle / (\|\mathbf{u}_T\|_2 \|\mathbf{v}_n\|_2)$. This quantity equals 1 when $\mathbf{u}_T = \mathbf{v}_n$, and is strictly less than 1 otherwise. As the softmax temperature tends to infinity, or equivalently, as the attention logits are scaled by a constant tending to infinity, the model approaches the behavior of a conditional $k$-gram estimator. Hence, the third layer effectively implements a $k$-th order induction head. $\qquad\square$

We are now ready to prove our first main result. Specifically, we show that the aforementioned construction with 3 layers and a single head can be adapted to a two-layer counterpart with two heads in the first layer and one head in the second.

> **Theorem 2.** (Two heads in the first layer, one in the second): The conditional $k$-gram model can be represented using a transformer with two layers, the first containing two attention heads and the second containing a single attention head. The layers are separated by MLP blocks and include relative positional encodings as well as layer normalization. The embedding dimension is $6S + 3$.

**Remark 1.** (Depth-width tradeoff) Thm. 2 thus reveals an interesting tradeoff between depth (number of attention layers) and width (maximum number of attention heads per layer). While our construction here has both depth and width two, its counterpart in Thm. 1 instead has depth three and width one. Thus our new architecture trades off depth with an additional attention head, thereby increasing the width.

*Proof sketch.* We provide a brief sketch of the proof for Thm. 2, using the same notation as that of the Thm. 1, and focusing on the key differences between the two architectures.

**Layer One.** For each position $n$ in the sequence $s_{0:T}$, the first attention head in the first layer learns a focused attention pattern over the positions $\{n, n-1, \ldots, n-k+1\}$. In a subspace of the embedding space, this head computes the representation $\mathbf{u}_n = \left( \sum_{i=0}^{k-1} 3^i \cdot e_{x_{n-i}}^S \right) / \left( \sum_{i=0}^{k-1} 3^i \right)$. On the other hand, the second attention head in this layer focuses on the preceding context, attending to positions $\{n-1, n-2, \ldots, n-k\}$, and computes the representation $\mathbf{v}_n = \left( \sum_{i=1}^{k} 3^i \cdot e_{x_{n-i}}^S \right) / \left( \sum_{i=1}^{k} 3^i \right)$. After this layer, the embedding at position $n$ contains both the vectors $\mathbf{u}_n/\|\mathbf{u}_n\|_2$ and $\mathbf{v}_n/\|\mathbf{v}_n\|_2$, achieved by choosing an appropriate choice of embedding dimensions, query, key, and value matrices, positional encodings, and MLPs with layer normalization.

**Layer Two.** The second attention layer now plays the role of an induction head, analogous to the third layer in Theorem 1. At position $n$, this head computes an attention score that is proportional to the cosine similarity $\langle \mathbf{u}_T, \mathbf{v}_n \rangle / \|\mathbf{u}_T\|_2 \|\mathbf{v}_n\|_2$. This score attains the maximum value of one when

$\mathbf{u}_T = \mathbf{v}_n$, and strictly less otherwise. As the temperature parameter in the softmax becomes large, or equivalently as the inner product is multiplied by a large constant, the attention mechanism approaches the behavior of a conditional $k$-gram estimator. Therefore, this two-layer transformer, with two heads in the first layer and a single head in the second, is sufficient to represent the conditional $k$-gram model. We refer to App. A for the full proof. □

## 4 Main Result: Two-Layer Single-Head Construction

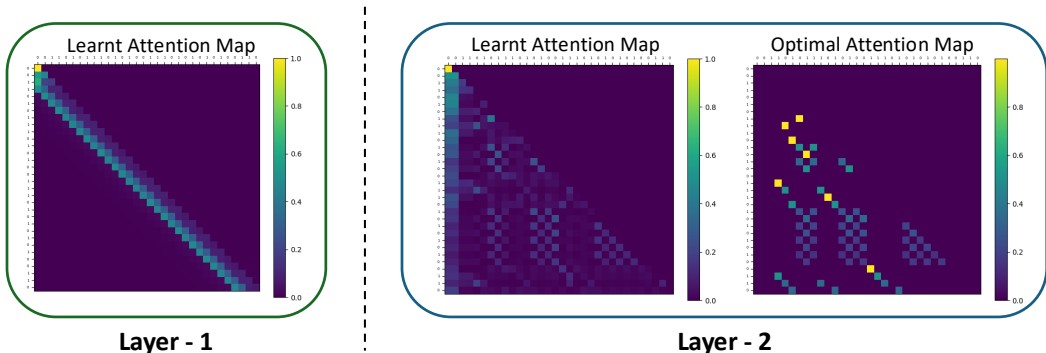

**Layer - 1**          **Layer - 2**

Figure 2: Attention maps learnt by a two-layer, single-head transformer trained on sequences generated by random Markov chains of order 3 (Subsec. 2.1). (i) in the first layer, the attention map shows a clear pattern: attention weights increase monotonically along the first three lower-diagonals and drop to zero beyond that. This suggests that the relative positional bias is maximized the diagonal with an offset of $-k = -3$, i.e., the third diagonal below the main diagonal, which is consistent with the construction in Sec. 4, (ii) in the second layer, the attention map closely resembles the ideal attention pattern required to approximate the conditional $k$-gram estimator. We note that all experiments were conducted using standard initialization schemes. Additional experiments with different orders of markov chains and experimental details are provided in App. H. In Subsec. H.2, we also experimentally demonstrate that single-layer transformers fail to solve the induction head task with the same order of parameters. Finally, in Subsec. H.3, we test the robustness of the two-layer, single-head model to noise in the input sequences.

While Thm. 2 above demonstrates that the conditional $k$-gram can be represented by a two-layer transformer, it however relies on a two-head construction, thus still leaving open our motivating question: *can a 2-layer, 1-head transformer represent the conditional $k$-gram?* In this section, we precisely address this gap and realize a 2-layer, 1-head model to represent $k^{\text{th}}$-order Markov processes for any $k \geqslant 1$. The key idea behind our approach is to leverage the MLP—and specifically, the non-linearities such as ReLU and LayerNorm—to isolate symbols at particular positions that are useful for estimating induction heads. This contrasts with prior constructions, which primarily focused on the attention mechanism while underutilizing the role of the MLP and its non-linear components. As a result, our construction highlights how these non-linear MLP elements can play a critical role in enabling in-context learning, demonstrating their significance. We now present our main result.

> **Theorem 3.** (Two-layer, single-head construction): The conditional $k$-gram model can be represented using a transformer with two layers, both containing a single attention head. The layers are separated by MLP blocks and include relative positional encodings as well as layer normalization. The embedding dimension is $6S + 3$.

**Remark 2.** (Parameter count and bit precision) We note that the above construction utilizes $9(6S + 3)^2 + (6S + 3)(2T + 2S + 9)$ parameters. A slightly more efficient construction in terms of parameter count with $8(6S + 3)^2 + (6S + 3)(2T + 2S + 9)$ parameters can be found in App. B. For both these results, it suffices to have $\Omega(\log(T) + k)$ bits per parameter, with $\mathcal{O}(1/T)$ additive approximation error, where $T$ is the sequence length.

*Proof Sketch.* We now provide an outline of the proof of Thm. 3, using notation consistent with the constructions in Thm. 1 and Thm. 2.

**Layer One.** For each position $n$ in the sequence $s_{0:T}$, the attention head in the first layer attends to the preceding context, focusing on positions $\{n-1, n-2, \ldots, n-k\}$, and computes the representation $\mathbf{v}_n = \left(\sum_{i=1}^{k} 3^i \cdot e^S_{x_{n-i}}\right) / \left(\sum_{i=1}^{k} 3^i\right)$, with attention weights $\text{attn}_{n,i} = 3^i / \left(\sum_{j=1}^{k} 3^j\right)$ for $i \in \{n-1, \ldots, n-k\}$, and zero otherwise. This matches the optimal attention pattern for Layer-1 illustrated in Fig. 2.

**MLP.** In contrast to previous constructions, the MLP situated between the two attention layers plays a central role in our proof and is structured to reconstruct the complementary representation $\mathbf{u}_n = \left(\sum_{i=0}^{k-1} 3^i \cdot e^S_{x_{n-i}}\right) / \left(\sum_{i=0}^{k-1} 3^i\right)$, which corresponds to a hard attention pattern over the positions $\{n, n-1, \ldots, n-k+1\}$. Recall that this $\mathbf{u}_n$ was constructed in Thm. 2 thanks to a second attention head, absent in our architecture.

Our main idea to tackle this issue is to isolate the one-hot embedding $e^S_{x_{n-k}}$ from $\mathbf{v}_n$ computed in the first layer. This is achieved by embedding the entire one-hot vocabulary within the weight matrix of the first MLP layer. When combined with a suitable bias and followed by a ReLU activation and layer normalization, this configuration filters out all but the component aligned with $x_{n-k}$, thereby producing $e^S_{x_{n-k}}$. At this stage, the embedding also retains the representation $\mathbf{v}_n$, computed by the first attention layer. Subsequent layers, along with appropriately configured skip connections, implement a linear transformation that combines $e^S_{x_{n-k}}$, $\mathbf{v}_n$, and the token embedding $e^S_{x_n}$, the latter of which is introduced via a skip connection. Since $\mathbf{v}_n$ and $e^S_{x_{n-k}}$ occupy orthogonal subspaces of the embedding, and all required components are explicitly available, this construction yields the intermediate representation $\mathbf{u}_n = \left(e^S_{x_n} / \sum_{i=0}^{k-1} 3^i\right) + \left(3 e^S_{x_{n-k}} \mathbf{v}_n\right) / \left(\sum_{i=0}^{k-1} 3^i\right) - \left(3^k \cdot e^S_{x_{n-k}}\right) / \left(\sum_{i=0}^{k-1} 3^i\right)$, which simplifies exactly to the desired form $\mathbf{u}_n = \left(\sum_{i=0}^{k-1} 3^i \cdot e^S_{x_{n-i}}\right) / \left(\sum_{i=0}^{k-1} 3^i\right)$. With appropriately chosen weights, layer normalization, and skip connections, the resulting embedding at position $n$ encodes both $\mathbf{u}_n / \|\mathbf{u}_n\|_2$ and $\mathbf{v}_n / \|\mathbf{v}_n\|_2$.

**Layer Two.** As in the second layer of Thm. 2, the attention mechanism in the second layer is configured to act as a $k^{\text{th}}$-order induction head by computing an attention score at position $n$ that is proportional to the cosine similarity $(\langle \mathbf{u}_T, \mathbf{v}_n \rangle / \|\mathbf{u}_T\|_2 \|\mathbf{v}_n\|_2)$. This score reaches its maximum value of one when $\mathbf{u}_T = \mathbf{v}_n$, and is strictly less than one otherwise. As the softmax temperature tends to infinity, or equivalently as the attention logits are scaled by a constant that tends to infinity, the resulting distribution converges to that of a conditional $k$-gram estimator. App. B details the complete proof. $\qquad\square$

**Importance of non-linearities.** This construction demonstrates that non-linear components such as ReLU activations and layer normalization are not merely auxiliary, but play a critical role in enabling the transformer architecture to express higher-order inductive structures that are essential for in-context learning.

**Note.** Due to the problem's non-convexity, multiple constructions may enable two-layer, single-head transformers to implement a $k^{\text{th}}$-order induction head; our approach offers one such construction motivated by two key insights. First, empirical observation: for datasets generated from $k^{\text{th}}$-order Markov chains, we observe that in a trained transformer model, the first-layer attention weights increase monotonically along the first $k$ lower diagonals before dropping sharply to near-zero (see Fig. 2 and Subsec. H.1). Second, prior theoretical construction: as demonstrated in Rajaraman et al. [26], the attention pattern can be designed to be proportional to a dyadic sum representation. Taken together, they offer both empirical and theoretical grounding for our construction of $k^{\text{th}}$-order induction heads.

## 5  Gradient Descent Analysis

Thm. 3 establishes a representation result showing that a two-layer transformer with single attention heads can implement the conditional $k$-gram model. However, this does not guarantee that current optimization algorithms such as gradient descent (or its stochastic variants) can recover such a solution in practice. To this end, in this section we focus on first-order Markov chains and show

that gradient descent on a reduced variant of our two-layer transformer indeed learns an induction head, and thereby the in-context conditional 1-gram. On this note, we would like to emphasize that first-order Markov analysis a critical first-step and a building block for challenging higher-order analysis [20].

Building towards our result, we quickly recall a few recent works [10, 20] that also analyze the gradient dynamics on first-order Markov data with simplified transformer architectures. Specifically, Edelman et al. [10] study an attention-only linear transformer without MLPs and softmax operations, while Nichani et al. [20] analyze a disentangled transformer, with residual and attention streams split. On the other hand, our transformer architecture in Def. 3 consists of relative positional encodings, softmax attention, MLP with ReLU activations, and layer normalization. While non-linearities such as ReLU and layer normalization play a crucial role in our representation result, as illustrated in Sec. 4, at the same time, they make the gradient analysis challenging and intractable even for a simple first-order Markov setup.

To tackle this, we study a reduced model wherein we only treat the salient components of our two-layer architecture in Thm. 3, such as positional encodings and attention layer, as parameters, and treating the rest as approximately optimal. This enables for a a faithful reproduction of the ICL phenomenon exhibited by the parent model, whilst being tractable. This is akin to the reparameterization strategies of Makkuva et al. [18] and Nichani et al. [20] for gradient-flow/gradient-descent analysis. Alternatively, this can also be viewed as a form of *good parameter initialization*, similar to that of [10, 20]. We now start with our technical assumptions on the data distribution and the transformer architecture. We use notation from Subsec. 2.2, dropping the head-identifying superscript.

**Assumptions.**

(i) Data distribution assumptions: Following [20], we assume the prior over transition kernels enforces non-degeneracy, positive transitions, and constant mean (see App. C).

(ii) The positional embeddings used for the keys in the first attention layer are scalar-valued, i.e. $\mathbf{p}_n^{(1,K)} \propto p_n$, where $p_n$ is a scalar.

(iii) $\mathbf{W}_{Q,K,V}^{(1)}$ and $\mathbf{p}^{(1,V)}$ are chosen so that the attention weight for position $n - i$ is given by $\mathrm{att}_{n,n-i}^{(1)} = \exp(p_i) / \left( \sum_{j=0}^{i} \exp(p_j) \right) \in \{0,1\}$, and the corresponding value output is $\mathbf{v}_n = \sum_{i=0}^{n} \mathrm{att}_{n,n-i}^{(1)} \cdot e_{x_{n-i}}^S$.

(iv) We assume the MLP is optimal—i.e., it outputs the correct $\mathbf{u}_n$ even from suboptimal $\mathbf{v}_n$; this holds for first-order Markov chains via skip connections.

(v) $\mathbf{W}_{Q,K}^{(2)}$ are chosen such that the attention in this layer is defined as $\mathrm{att}_{T,i}^{(2)} = \exp\left(a_2 \cdot \langle \mathbf{v}_i, \mathbf{u}_T \rangle \right) / \left( \sum_{j=0}^{T} \exp(a_2 \cdot \langle \mathbf{v}_j, \mathbf{u}_n \rangle) \right)$, where $a_2 \in \mathbb{R}$ is the attention scalar.

(vi) $\mathbf{W}_V^{(2)}$ is chosen so that the final logit $\mathrm{logit}_T = \sum_{i=0}^{T} \mathrm{att}_{n,i}^{(2)} e_{x_i} \in \mathbb{R}^S$,

**Loss function.** We minimize the cross-entropy loss for next-token prediction [20]:

$$\mathcal{L}(\theta) = -\mathbb{E}_{\pi \sim \mathbb{P}_\pi, \, x_{0:T} \sim \pi} \left[ \sum_{s \in \mathcal{S}} \pi\left(x_{T+1} = s \mid x_{0:T}\right) \log\left(\mathrm{logit}_T(s) + \varepsilon\right) \right], \tag{1}$$

where $\varepsilon > 0$ is a small additive constant, $\mathbb{P}_\pi(\cdot)$ denotes the prior over Markov transition kernels, and $\mathrm{logit}_T(s)$ is the logit corresponding to the symbol $s$.

**Training algorithm.** We denote the set of trainable parameters as $\theta = (\mathbf{p}, a_2)$, where the positional scalars $\mathbf{p} = [p_0, p_1, \ldots, p_T]^T$ and $a_2$ is the attention scalar. Similar to [20], we adopt a two-stage training procedure. (i) In the first stage, we optimize the positional scalars $\mathbf{p}$ using gradient descent for $T_1$ steps with learning rate $\eta_1$, (ii) in the second stage, we freeze $\mathbf{p}$ and train only $a_2$ for an additional $T_2$ steps using a separate learning rate $\eta_2$. A key distinction in our setup is how we treat layer normalization during training. While in the first stage we omit all layer norms in the MLP, during the second step we reintroduce them. This stage-wise enables a tractable theoretical analysis. We refer to App. C for further details. We now present our main result:

**Theorem 4.** (Convergence of the training Algorithm): Suppose that Assumptions (i)- (vi) hold and that the Markov sequence length $T$ satisfies $T+1 \geqslant \text{poly}(\gamma^{-1}, S)$ for some constant $\gamma > 0$. Then there exist $\varepsilon > 0$, learning rates $\eta_1, \eta_2$, and step counts $T_1, T_2$, such that the output of the above two-stage training algorithm, $\hat{\theta} = (\{\hat{p}_i\}_{i=0}^{T}, \hat{a}_2)$, satisfies

$$\mathcal{L}(\hat{\theta}) - \mathcal{L}^* \lesssim \frac{\log(T+1)}{(T+1)^{c\gamma}},$$

for a constant $c$ independent of $(\gamma, S)$, and the optimal loss $\mathcal{L}^* := -\mathbb{E}\left[(1/S) \sum_{s,s'} \pi(s' \mid s) \log \pi(s' \mid s)\right]$.

Thus Thm. 4 illustrates that the transformer model trained via the two-stage algorithm achieves near-optimal loss asymptotically. Furthermore, we can also show that $\hat{\theta}$ approximates the conditional 1-gram model, i.e. first-order induction head, with vanishing error as sequence length grows (see App. C). For completeness, perturbative analysis of the architecture consisting of two attention heads in the first layer and one in the second (see Sec. 3) for any-order Markov chains can be found in App. D.

# 6   Conclusion

In this paper, improving over prior three-layer constructions, we show that a two-layer, single-head transformer can represent any $k$-th order Markov process. Thus our result gives the tightest known depth characterization for in-context learning of conditional $k$-grams. Additionally, we provide a gradient descent analysis of how such representations emerge during training on first-order Markov data. While these results deepen our theoretical understanding of ICL and show that compact transformer architectures can be both expressive and learnable, several open questions remain. In particular, fully characterizing the learning dynamics for higher-order processes is an important avenue of future research. Further, we view our contributions as a first step toward understanding the fundamental limits of ICL in compact transformer architectures, offering not only a tractable setting for theoretical analysis, but also a potential pathway toward more efficient model designs.

## Acknowledgements

This work was supported in part by the Swiss National Science Foundation under Grant No. 200364. We are grateful to Eshaan Nichani for insightful discussions and valuable input throughout the preparation of this paper.

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

# Appendix

## Contents

# A Two Heads in Layer One, One in Layer Two Construction

## A.1 Modification of the Layer Norm

Following Rajaraman et al. [26] (Appendix C.1 in Rajaraman et al. [26]), we use a modified version of layer normalization to extract the $\ell_2$-normalized direction of the embedding $\mathbf{x}_n$. Specifically, we augment the embedding by concatenating it with its negation to form $[\mathbf{x}_n; -\mathbf{x}_n]$, which ensures that the mean of the features is zero. Applying standard layer normalization to this vector yields $\sqrt{d} \cdot \mathbf{x}_n / \|\mathbf{x}_n\|$ in the first $d$ components, since the standard deviation of the augmented vector is $\|\mathbf{x}_n\| / \sqrt{d}$. To recover the unit-normalized embedding $\mathbf{x}_n / \|\mathbf{x}_n\|$, we simply divide the first $d$ components of the output by $\sqrt{d}$, and extract the first $d$ components. Throughout this work, we assume this normalization and scaling step is handled implicitly, and we interpret layer normalization as returning the normalized embedding direction. In order to avoid explicitly repeating this transformation at each layer, we redefine the layer norm to implicitly return $\mathbf{v}/\|\mathbf{v}\|_2$ for any input vector $\mathbf{v}$. Hence, $\mathbf{v}/\|\mathbf{v}\|_2 = \mathrm{LN}(\mathbf{v})$, where, $\mathrm{LN}(\cdot)$ denotes the modified layer norm operator. Note that this modified definition is used throughout this paper.

## A.2 Construction

**Theorem 5.** (Two Layer, Two Heads in the First Layer). We claim that there exists a two-layer attention-based Transformer model $f_\theta$, consisting of two attention heads in the first layer and a single attention head in the second layer, which can be configured in one of two equivalent ways: (1) with two MLP layers inserted between the first and second attention blocks, or (2) with no intermediate MLP layers, but with layer normalization applied to the query and key projections in the second attention block, such that:

$$f_\theta(s_{1:T})_{s'} \approx \pi(s' \mid s_{1:T}) \quad \forall (s_{1:T}, s') \sim \pi, \pi \sim \mathbb{P}_\pi.$$

Note that $\mathbb{P}_\pi$ denotes an arbitrary distribution over trajectories, where each trajectory is generated using a Markovian transition kernel $\pi$. The kernel $\pi$ corresponds to a $k^{\text{th}}$-order Markov chain.

Importantly, in both configurations, the input embeddings and the structure of the first layer (**Layer 1**) remain identical. We define the input embeddings as follows:

$$\mathbf{x}_n^{(1)} = \mathrm{Emb}(x_n) = \begin{bmatrix} \mathbf{0}_{1\times 3} & e_{x_n}^S & \mathbf{0}_{1\times 5S} \end{bmatrix}^T \in \mathbb{R}^{6S+3}$$

In the first layer, for both the heads we use the following relative positional embeddings

$$\mathbf{p}_i^{(\texttt{head:1}),K} = \mathbf{p}_i^{(\texttt{head:2}),K} = \begin{cases} \kappa \cdot \begin{bmatrix} 1 & 0 & \mathbf{0}_{1\times(1+6S)} \end{bmatrix}^T, & \text{if } i = 0, \\ (i \cdot \log(3) + \kappa) \cdot \begin{bmatrix} 0 & 1 & \mathbf{0}_{1\times(1+6S)} \end{bmatrix}^T, & \text{if } i \in \{1, 2, \cdots, k-1\}, \\ (i \cdot \log(3) + \kappa) \cdot \begin{bmatrix} 0 & 0 & 1 & \mathbf{0}_{1\times(6S)} \end{bmatrix}^T, & \text{if } i = k, \\ \mathbf{0}, & \text{if } i > k \end{cases},$$

and the same, value embeddings,

$$\mathbf{p}_i^{(\texttt{head:1}),V} = \mathbf{p}_i^{(\texttt{head:2}),V} = \begin{cases} 3^i \begin{bmatrix} 1 & \mathbf{0} \end{bmatrix}^T & \text{for } i \leqslant k \\ \mathbf{0} & i > k. \end{cases}$$

**Layer 1, Head 1.** Consider the key and query matrices,

$$\mathbf{W}_K^{(\texttt{head:1})} = \begin{bmatrix} \mathbf{1}_{1\times 2} & \mathbf{0}_{1\times 1} & \mathbf{0}_{1\times 6S} \\ \mathbf{0}_{(2+6S)\times 2} & \mathbf{0}_{(2+6S)\times 1} & \mathbf{0}_{(2+6S)\times(2+6S)} \end{bmatrix}$$

$$\mathbf{W}_Q^{(\texttt{head:1})} = \begin{bmatrix} \mathbf{0}_{1\times 3} & \mathbf{1}_{1\times S} & \mathbf{0}_{1\times 5S} \\ \mathbf{0}_{(2+6S)\times 3} & \mathbf{0}_{(2+6S)\times S} & \mathbf{0}_{(2+6S)\times 5S} \end{bmatrix}$$

On computing (for $i = 0$ to $i = k - 1$)

$$\mathbf{W}_K^{(\texttt{head:1})} \texttt{Emb}(x_{n-i}) = \mathbf{0}$$
$$\mathbf{W}_K^{(\texttt{head:1})} \mathbf{p}_i^{(\texttt{head:1}),K} = \begin{bmatrix} (i \cdot \log(3) + \kappa) & \mathbf{0}_{(2+6S) \times 1} \end{bmatrix}^T$$
$$\mathbf{W}_Q^{(\texttt{head:1})} \texttt{Emb}(x_n) = \begin{bmatrix} 1_{1 \times 1} & \mathbf{0}_{1 \times (2+6S)} \end{bmatrix}^T$$

Then, observe that,

$$\left\langle \mathbf{W}_K^{(\texttt{head:1})} \left( \texttt{Emb}(x_{n-i}) + \mathbf{p}_i^{(\texttt{head:1}),K} \right), \mathbf{W}_Q^{(\texttt{head:1})} \texttt{Emb}(x_n) \right\rangle = (i \cdot \log(3) + \kappa) \cdot \mathbb{I}(0 \leqslant i \leqslant \min\{n, k-1\})$$

Letting $\kappa \to \infty$, this results in the attention pattern,

$$\texttt{att}_{n,n-i}^{(\texttt{head:1})} = \frac{3^i \mathbb{I}(0 \leqslant i \leqslant \min\{n, k-1\})}{\sum_{i'=0}^{\min\{n,k-1\}} 3^{i'}} = \frac{3^i \mathbb{I}(0 \leqslant i \leqslant \min\{n, k-1\})}{\sum_{i'=0}^{\min\{n,k\}-1} 3^{i'}} = \frac{3^i \mathbb{I}(0 \leqslant i \leqslant \min\{n, k-1\})}{C_{\texttt{attn}}}$$

Choose the value matrix as,

$$\mathbf{W}_V^{(\texttt{head:1})} = \begin{bmatrix} \mathbf{0}_{3 \times 3} & \mathbf{0}_{3 \times S} & \mathbf{0}_{3 \times 4S} \\ \mathbf{0}_{3S \times 3} & \mathbf{0}_{S \times S} & \mathbf{0}_{3S \times 4S} \\ \mathbf{0}_{S \times 3} & I_{S \times S} & \mathbf{0}_{S \times 4S} \\ \mathbf{0}_{2S \times 3} & \mathbf{0}_{S \times S} & \mathbf{0}_{2S \times 4S} \end{bmatrix}$$

Hence, the output of the attention head is:

$$\widetilde{\mathbf{x}}_n^{(\texttt{head:1})} = \begin{bmatrix} 0 & | & \mathbf{0}_{1 \times 2} & | & \mathbf{0}_{1 \times S} & | & \mathbf{u}_n & | & \mathbf{0}_{1 \times 2S} & | & \mathbf{0}_{1 \times 2S} \end{bmatrix}^T,$$
$$\text{where,} \ \mathbf{u}_n = \sum_{i=0}^{\min\{n,k-1\}} \texttt{att}_{n,n-i}^{\texttt{head:1}} e_{x_{n-i}}^S$$

**Layer 1, Head 2.** Consider the key and query matrices,

$$\mathbf{W}_K^{(\texttt{head:2})} = \begin{bmatrix} 0_{1 \times 1} & \mathbf{1}_{1 \times 2} & \mathbf{0}_{1 \times 6S} \\ \mathbf{0}_{(2+6S) \times 1} & \mathbf{0}_{(2+6S) \times 2} & \mathbf{0}_{(2+6S) \times (2+6S)} \end{bmatrix}$$
$$\mathbf{W}_Q^{(\texttt{head:2})} = \begin{bmatrix} \mathbf{0}_{1 \times 3} & \mathbf{1}_{1 \times S} & \mathbf{0}_{1 \times 5S} \\ \mathbf{0}_{(2+6S) \times 3} & \mathbf{0}_{(2+6S) \times S} & \mathbf{0}_{(2+6S) \times 5S} \end{bmatrix}$$

On computing (for $i = 1$ to $i = k$)

$$\mathbf{W}_K^{(\texttt{head:2})} \texttt{Emb}(x_{n-i}) = \mathbf{0}$$
$$\mathbf{W}_K^{(\texttt{head:2})} \mathbf{p}_i^{(1),K} = \begin{bmatrix} (i \cdot \log(3) + \kappa) & \mathbf{0}_{(2+6S) \times 1} \end{bmatrix}^T$$
$$\mathbf{W}_Q^{(\texttt{head:2})} \texttt{Emb}(x_n) = \begin{bmatrix} 1_{1 \times 1} & \mathbf{0}_{1 \times (2+6S)} \end{bmatrix}^T$$

Then, observe that,

$$\left\langle \mathbf{W}_K^{(\texttt{head:2})} \left( \texttt{Emb}(x_{n-i}) + \mathbf{p}_i^{(\texttt{head:2}),K} \right), \mathbf{W}_Q^{(\texttt{head:2})} \texttt{Emb}(x_n) \right\rangle = (i \cdot \log(3) + \kappa) \cdot \mathbb{I}(1 \leqslant i \leqslant \min\{n, k\})$$

Letting $\kappa \to \infty$, this results in the attention pattern,

$$\text{att}_{n,n-i}^{(1)} = \frac{3^i \mathbb{I}(1 \leqslant i \leqslant \min\{n,k\})}{\sum_{i'=1}^{\min\{n,k\}} 3^{i'}} = \frac{3^{i-1} \mathbb{I}(1 \leqslant i \leqslant \min\{n,k\})}{\sum_{i'=0}^{\min\{n,k\}-1} 3^{i'}} = \frac{3^{i-1} \mathbb{I}(1 \leqslant i \leqslant \min\{n,k\})}{C_{\text{attn}}}$$

Choose the value matrix as,

$$\mathbf{W}_V^{(1)} = \begin{bmatrix} I_{3\times 3} & \mathbf{0}_{3\times S} & \mathbf{0}_{3\times 4S} \\ \mathbf{0}_{3S\times 3} & \mathbf{0}_{S\times S} & \mathbf{0}_{3S\times 4S} \\ \mathbf{0}_{S\times 3} & I_{S\times S} & \mathbf{0}_{S\times 4S} \\ \mathbf{0}_{2S\times 3} & \mathbf{0}_{S\times S} & \mathbf{0}_{2S\times 4S} \end{bmatrix}$$

The output of the second attention head is,

$$\widetilde{\mathbf{x}}_n^{(\texttt{head:2})} = \begin{bmatrix} Z_n & | & \mathbf{0}_{1\times 2} & | & \mathbf{0}_{1\times S} & | & \mathbf{0}_{1\times 2S} & | & \mathbf{v}_n & | & \mathbf{0}_{1\times 2S} \end{bmatrix}^T,$$

$$\text{where, } \mathbf{v}_n = \sum_{i=1}^{\min\{n,k\}} \text{att}_{n,n-i} \, e_{x_{n-i}}^S$$

$$Z_n = \sum_{i=1}^{\min\{k,n-1\}} \text{att}_{n,n-i} \, 3^i,$$

It is straightforward to verify that $Z_n = 3^{k+1}/5$ when $n \geqslant k+1$, and that $Z_n \leqslant 3^k/5$ otherwise. This observation will be useful later, as the value of $Z_n$ serves as a signal for whether $n \geqslant k+1$ or $n \leqslant k$. In particular, this distinction enables the next layer to bypass attention computations for indices $i \leqslant k$, where the condition $x_n = x_{i-1}, \ldots, x_{n-k+1} = x_{i-k}$ is not well defined.

Therefore, with skip connections:

$$\widetilde{\mathbf{x}}_n^{(1)} = \mathbf{x}_n^{(1)} + \widetilde{\mathbf{x}}_n^{(\texttt{head:1})} + \widetilde{\mathbf{x}}_n^{(\texttt{head:2})}$$

$$= \begin{bmatrix} Z_n & | & \mathbf{0}_{1\times 2} & | & e_{x_n}^S & | & \mathbf{u}_n & | & \mathbf{0}_{1\times S} & | & \mathbf{v}_n & | & \mathbf{0}_{1\times 2S} \end{bmatrix}^T$$

### A.2.1 Two Layer MLP followed by Second Attention Layer

**MLP.** The first layer for the MLP is :

$$\mathbf{W}_{\texttt{mlp}}^{(1)} = \begin{bmatrix} \mathbf{0}_{(3+S)\times(3+3S)} & \mathbf{0}_{(3+S)\times(S)} & \mathbf{0}_{(3+S)\times(2S)} \\ \mathbf{0}_{S\times(3+3S)} & \mathbf{0}_{S\times S} & \mathbf{0}_{S\times 2S} \\ \mathbf{0}_{S\times(3+3S)} & I_{S\times S} & \mathbf{0}_{S\times 2S} \\ \mathbf{0}_{3S\times(3+3S)} & \mathbf{0}_{3S\times S} & \mathbf{0}_{3S\times 2S} \end{bmatrix}$$

$$\mathbf{b}_{\texttt{mlp}}^{(1)} = \mathbf{0}_{(3+6S)\times 1}$$

Incorporating skip connections and applying non-linearities, we obtain:

$$\widetilde{\mathbf{x}}_{n,\texttt{mlp}}^{(1)} = \widetilde{\mathbf{x}}_n^{(1)} + \text{LN}\left(\text{ReLU}\left(\mathbf{W}_{\texttt{mlp}}^{(1)}\widetilde{\mathbf{x}}_n^{(1)}\right) + \mathbf{b}_{\texttt{mlp}}^{(1)}\right)$$

$$\widetilde{\mathbf{x}}_{n,\texttt{mlp}}^{(1)} = \begin{bmatrix} Z_n & | & \mathbf{0}_{1\times 2} & | & e_{x_n}^S & | & \mathbf{u}_n & | & \frac{\mathbf{u}_n}{\|\mathbf{u}_n\|_2} & | & \mathbf{v}_n & | & \mathbf{0}_{1\times S} & | & \mathbf{0}_{1\times S} \end{bmatrix}^T$$

The second layer for the MLP is :

$$\mathbf{W}_{\mathtt{mlp}}^{(2)} = \begin{bmatrix} \mathbf{0}_{(3+S)\times(3+3S)} & \mathbf{0}_{(3+S)\times(S)} & \mathbf{0}_{(3+S)\times(2S)} \\ \mathbf{0}_{S\times(3+3S)} & \mathbf{0}_{S\times S} & \mathbf{0}_{S\times 2S} \\ \mathbf{0}_{S\times(3+3S)} & \mathbf{0}_{S\times S} & \mathbf{0}_{S\times 2S} \\ \mathbf{0}_{S\times(3+3S)} & I_{S\times S} & \mathbf{0}_{S\times 2S} \\ \mathbf{0}_{2S\times(3+3S)} & \mathbf{0}_{2S\times S} & \mathbf{0}_{2S\times 2S} \end{bmatrix}$$

$$\mathbf{b}_{\mathtt{mlp}}^{(2)} = \mathbf{0}_{(3+6S)\times 1}$$

Including skip connections and non-linearities, we get:

$$\widetilde{\mathbf{x}}_{n,\mathtt{mlp}}^{(2)} = \widetilde{\mathbf{x}}_{n,\mathtt{mlp}}^{(1)} + \mathrm{LN}\left(\mathrm{ReLU}\left(\mathbf{W}_{\mathtt{mlp}}^{(2)}\widetilde{\mathbf{x}}_n^{(2)}\right) + \mathbf{b}_{\mathtt{mlp}}^{(2)}\right)$$

$$\widetilde{\mathbf{x}}_{n,\mathtt{mlp}}^{(2)} = \begin{bmatrix} Z_n & \mathbf{0}_{1\times 2} & e_{x_n}^S & \mathbf{u}_n & \frac{\mathbf{u}_n}{\|\mathbf{u}_n\|_2} & \mathbf{v}_n & \frac{\mathbf{v}_n}{\|\mathbf{v}_n\|_2} & \mathbf{0}_{1\times S} \end{bmatrix}^T$$

**Layer 2.** In this layer, all relative position encodings are set to $\mathbf{0}$, and we use the following query and key matrices for the attention mechanism in the second layer:

$$\mathbf{W}_Q^{(2)} = \sqrt{\kappa} \begin{bmatrix} 1 & \mathbf{0} & \mathbf{0} & \mathbf{0} \\ \mathbf{0} & \mathbf{0}_{S\times(2+3S)} & I_{S\times S} & \mathbf{0} \\ \mathbf{0} & \mathbf{0} & \mathbf{0} & \mathbf{0} \end{bmatrix}$$

$$\mathbf{W}_K^{(2)} = \sqrt{\kappa} \begin{bmatrix} 1 & \mathbf{0} & \mathbf{0} & \mathbf{0} \\ \mathbf{0} & \mathbf{0}_{S\times(2+4S)} & I_{S\times S} & \mathbf{0} \\ \mathbf{0} & \mathbf{0} & \mathbf{0} & \mathbf{0} \end{bmatrix}$$

With these choices in place, we obtain:

$$\left\langle (\mathbf{W}_K^{(2)}\widetilde{\mathbf{x}}_{i,\mathtt{mlp}}^{(2)}), (\mathbf{W}_Q^{(2)}\widetilde{\mathbf{x}}_{n,\mathtt{mlp}}^{(2)}) \right\rangle = \kappa \cdot Z_i Z_n + \frac{\kappa \cdot \langle \mathbf{v}_i, \mathbf{u}_n \rangle}{\|\mathbf{v}_i\|_2 \cdot \|\mathbf{u}_n\|_2} = \kappa \cdot Z_i Z_n + 2 - \kappa \cdot \left\| \frac{\mathbf{v}_i}{\|\mathbf{v}_i\|_2} - \frac{\mathbf{u}_n}{\|\mathbf{u}_n\|_2} \right\|^2$$

Applying the softmax function, we obtain:

$$\mathrm{att}_{n,i}^{(2)} = \frac{\exp\left(\kappa Z_i Z_n + 2\kappa - \kappa \left\| \frac{\mathbf{v}_i}{\|\mathbf{v}_i\|_2} - \frac{\mathbf{u}_n}{\|\mathbf{u}_n\|_2} \right\|^2 \right)}{\sum_{j=0}^n \exp\left(\kappa Z_j Z_n + 2\kappa - \kappa \left\| \frac{\mathbf{v}_j}{\|\mathbf{v}_j\|_2} - \frac{\mathbf{u}_n}{\|\mathbf{u}_n\|_2} \right\|^2 \right)}$$

We can see that for all $k+1 \leqslant i \leqslant n$, $\mathbf{v}_i = \mathbf{u}_n$ if and only if the local context around position $i$ matches that of position $n$, i.e., $\{x_{i-1-j} = x_{n-j}$ for all $j = 0, \ldots, k\}$. Moreover, for any $k+1 \leqslant i \leqslant n$, if $\mathbf{v}_i \neq \mathbf{u}_n$, then the normalized vectors are separated by a non-negligible margin:

$$\left\| \frac{\mathbf{v}_i}{\|\mathbf{v}_i\|_2} - \frac{\mathbf{u}_n}{\|\mathbf{u}_n\|_2} \right\|_2 \geqslant \frac{1}{3^k}.$$

Note that while this gap is small, it is strictly non-zero. Recall that $Z_i = \frac{3^{k+1}}{5}$ for $i \geqslant k+1$, and $Z_i \leqslant \frac{3^k}{5}$ otherwise. As a result, the attention mechanism favors positions $i$ such that $\mathbf{v}_i = \mathbf{u}_n$ and $i \geqslant k+1$. In the limit as $\kappa \to \infty$, this results in the following attention pattern:

$$\mathrm{att}_{n,\cdot}^{(2)} = \mathrm{Unif}(\mathcal{I}_n),$$

where $\mathcal{I}_n$ denotes the set of all indices $i \in \{k+1, \ldots, n\}$ whose local context matches that of position $n$, formally defined as:

$$\mathcal{I}_n := \{i \in \{k, \ldots, n\} \mid x_{i-1-j} = x_{n-j} \text{ for all } j = 0, \ldots, k\}.$$

Moreover, $\mathrm{Unif}(\mathcal{I}_n)$ denotes the uniform distribution over the indices in $\mathcal{I}_n$; that is, each position $i \in \mathcal{I}_n$ receives equal attention weight, and all other positions receive zero. We now choose the value projection matrix in the second attention layer as:

$$\mathbf{W}_V^{(2)} = \begin{bmatrix} \mathbf{0} & \mathbf{0} & \mathbf{0} \\ \mathbf{0}_{S \times 3} & I_{S \times S} & \mathbf{0} \end{bmatrix}.$$

With this choice, the output of the second attention layer becomes:

$$\widetilde{\mathbf{x}}_n^{(2)} = \widetilde{\mathbf{x}}_{n,\mathtt{mlp}}^{(2)} + \sum_{i=0}^{n} \mathrm{att}_{n,i}^{(2)} \begin{bmatrix} \mathbf{0} \\ e_{x_i}^S \end{bmatrix}$$

$$= \widetilde{\mathbf{x}}_{n,\mathtt{mlp}}^{(2)} + \frac{1}{|\mathcal{I}_n|} \sum_{i \in \mathcal{I}_n} \begin{bmatrix} \mathbf{0} \\ e_{x_i}^S \end{bmatrix}.$$

For the output linear layer, we choose:

$$\mathbf{W}_o = \begin{bmatrix} \mathbf{0}_{S \times (5S+3)} & I_{S \times S} \end{bmatrix}, \qquad \mathbf{b}_o = \mathbf{0}.$$

This yields the output logits:

$$\mathrm{logit}_n = \frac{1}{|\mathcal{I}_n|} \sum_{i \in \mathcal{I}_n} e_{x_i}^S = \sum_{i=k}^{n} \frac{\mathbb{I}\left(\forall\, 1 \leqslant j \leqslant k,\ x_{i-j} = x_{n-j+1}\right)}{\sum_{i'=k}^{n} \mathbb{I}\left(\forall\, 1 \leqslant j \leqslant k,\ x_{i'-j} = x_{n-j+1}\right)} \cdot e_{x_i}^S.$$

Finally, by adjusting the notation and setting $n = T$ to reflect prediction at the final time step, we obtain:

$$\mathrm{logit}_T(x_{T+1}) = \frac{\sum_{n=k}^{T} \mathbb{I}\left(\forall\, 0 \leqslant i \leqslant k,\ x_{n-i} = x_{T-i+1}\right)}{\sum_{n=k}^{T} \mathbb{I}\left(\forall\, 1 \leqslant i \leqslant k,\ x_{n-i} = x_{T-i+1}\right)},$$

which precisely computes the conditional $k$-gram probability.

### A.2.2 Layer Norms in the Second Attention Block

We start with $\widetilde{\mathbf{x}}_n^{(1)}$ from the first layer as defined.

$$\widetilde{\mathbf{x}}_n^{(1)} = \mathbf{x}_n^{(1)} + \widetilde{\mathbf{x}}_n^{(\mathtt{head:1})} + \widetilde{\mathbf{x}}_n^{(\mathtt{head:2})}$$

$$= \begin{bmatrix} Z_n \mid \mathbf{0}_{1 \times 2} \mid e_{x_n}^S \mid \mathbf{u}_n \mid \mathbf{0}_{1 \times S} \mid \mathbf{v}_n \mid \mathbf{0}_{1 \times 2S} \end{bmatrix}^T$$

**Layer 2.** In this layer, all the relative position encodings are set as $\mathbf{0}$ and instead,

$$\mathbf{W}_Q^{(2)} = \begin{bmatrix} 0 & \mathbf{0} & \mathbf{0} & \mathbf{0} \\ 0 & \mathbf{0}_{S \times (2+3S)} & I_{S \times S} & \mathbf{0} \\ 0 & \mathbf{0} & \mathbf{0} & \mathbf{0} \end{bmatrix}$$

$$\mathbf{W}_K^{(2)} = \begin{bmatrix} 0 & \mathbf{0} & \mathbf{0} & \mathbf{0} \\ 0 & \mathbf{0}_{S \times (2+4S)} & I_{S \times S} & \mathbf{0} \\ 0 & \mathbf{0} & \mathbf{0} & \mathbf{0} \end{bmatrix}$$

With these choices, and adding a layer normalization after the output of the key linear layer and the query linear layer

$$\left\langle \mathrm{LN}(\mathbf{W}_K^{(2)}\tilde{\mathbf{x}}_{i,\mathtt{mlp}}^{(2)}), \mathrm{LN}(\mathbf{W}_Q^{(2)}\tilde{\mathbf{x}}_{n,\mathtt{mlp}}^{(2)}) \right\rangle = \frac{2\langle \mathbf{v}_i, \mathbf{u}_n \rangle}{\|\mathbf{v}_i\|_2 \cdot \|\mathbf{u}_n\|_2}$$

$$= 2 - \left\| \frac{\mathbf{v}_i}{\|\mathbf{v}_i\|_2} - \frac{\mathbf{u}_n}{\|\mathbf{u}_n\|_2} \right\|^2.$$

Taking the softmax with a temperature parameter $\kappa$,

$$\mathrm{att}_{n,i}^{(2)} = \frac{\exp\left(2\kappa - \kappa \left\| \frac{\mathbf{v}_i}{\|\mathbf{v}_i\|_2} - \frac{\mathbf{u}_n}{\|\mathbf{u}_n\|_2} \right\|^2\right)}{\sum_{j=0}^{n} \exp\left(2\kappa - \kappa \left\| \frac{\mathbf{v}_j}{\|\mathbf{v}_j\|_2} - \frac{\mathbf{u}_n}{\|\mathbf{u}_n\|_2} \right\|^2\right)}$$

Note that although this gap is small, it remains non-zero. In particular, as $\kappa \to \infty$, the resulting attention pattern approaches

$$\mathrm{att}_{n,\cdot}^{(2)} = \mathrm{Unif}(\mathcal{I}_n \cup \hat{\mathcal{I}}_k),$$

where $\mathcal{I}_n$ denotes the set of indices $i \in \{k, \ldots, n\}$ whose local context matches that of position $n$. Formally,

$$\mathcal{I}_n := \{i \in \{k, \ldots, n\} \mid x_{i-1-j} = x_{n-j} \text{ for all } j = 0, \ldots, k\}.$$

In contrast, $\hat{\mathcal{I}}_k$ captures earlier positions $i \in \{0, \ldots, k-1\}$ whose (normalized) key vectors are aligned with the normalized query at position $n$:

$$\hat{\mathcal{I}}_k := \left\{ i \in \{0, \ldots, k-1\} \left| \frac{\mathbf{v}_i}{\|\mathbf{v}_i\|_2} = \frac{\mathbf{k}_n}{\|\mathbf{u}_n\|_2} \right. \right\}.$$

On choosing,

$$\mathbf{W}_V^{(2)} = \begin{bmatrix} \mathbf{0} & \mathbf{0} & \mathbf{0} \\ \mathbf{0}_{S\times 3} & I_{S\times S} & \mathbf{0} \end{bmatrix}.$$

We obtain,

$$\tilde{\mathbf{x}}_n^{(2)} = \mathbf{x}_n^{(2)} + \sum_{i=1}^{n} \mathrm{att}_{n,i}^{(2)} \begin{bmatrix} \mathbf{0} \\ e_{x_i}^S \end{bmatrix} = \mathbf{x}_n^{(3)} + \frac{1}{|\mathcal{I}_n| + |\hat{\mathcal{I}}_k|} \sum_{i \in \mathcal{I}_n \cup \hat{\mathcal{I}}_k} \begin{bmatrix} \mathbf{0} \\ e_{x_i}^S \end{bmatrix}.$$

For the output linear layer, we choose:

$$\mathbf{W}_o = \begin{bmatrix} \mathbf{0}_{S\times(5S+3)} & I_{S\times S} \end{bmatrix}, \qquad \mathbf{b}_o = \mathbf{0}.$$

which results in,

$$\mathrm{logit}_n = \frac{1}{|\mathcal{I}_n| + |\hat{\mathcal{I}}_k|} \sum_{i \in \mathcal{I}_n \cup \hat{\mathcal{I}}_k} e_{x_i}^S = \sum_{i=k}^{n} \frac{\mathbb{I}(\forall 1 \leqslant j \leqslant k, \ x_{i-j} = x_{n-j+1})}{\sum_{i'=k}^{n} \mathbb{I}(\forall 1 \leqslant j \leqslant k, \ x_{i'-j} = x_{n-j+1}) + \hat{\mathcal{I}}_k} \cdot e_{x_i}^S$$

$$+ \sum_{i \in \hat{\mathcal{I}}_k} \frac{\mathbb{I}(\forall 1 \leqslant j \leqslant k, \ x_{i-j} = x_{n-j+1})}{\sum_{i'=k}^{n} \mathbb{I}(\forall 1 \leqslant j \leqslant k, \ x_{i'-j} = x_{n-j+1}) + \hat{\mathcal{I}}_k} \cdot e_{x_i}^{S}$$

Now, by substituting $n = T$, and aiming to predict the next symbol given the current context, we obtain:

$$\mathrm{logit}_T(x_{T+1}) = \frac{\sum_{n=k}^{T} \mathbb{I}(\forall 0 \leqslant i \leqslant k, \ x_{n-i} = x_{T-i+1})}{\sum_{n=k}^{T} \mathbb{I}(\forall 1 \leqslant i \leqslant k, \ x_{n-i} = x_{T-i+1}) + \hat{\mathcal{I}}_k}$$
$$+ \frac{\sum_{i \in \hat{\mathcal{I}}_k} \langle e_{T+1}^{S}, e_i^{S} \rangle}{\sum_{n=k}^{T} \mathbb{I}(\forall 1 \leqslant i \leqslant k, \ x_{n-i} = x_{T-i+1}) + \hat{\mathcal{I}}_k}$$

This corresponds to a *biased conditional* $k$-gram model, where the bias originates from the second term in the expression above. Notably, this bias vanishes in the limit as $T \to \infty$. The reason is that the numerator of the second term remains constant with respect to sequence length, since $\hat{\mathcal{I}}_k$ is fixed and does not scale with $T$. In contrast, the denominator, given by $\sum_{n=k}^{T} \mathbb{I}(\forall \, 1 \leqslant i \leqslant k, \ x_{n-i} = x_{T-i+1})$, grows with $T$, assuming the underlying Markov chain is irreducible (i.e., every state is reachable from every other state). Consequently, under this assumption of irreducibility, the influence of the second term diminishes as $T$ becomes large, yielding the approximation for a sufficiently large $T$):

$$\mathrm{logit}_T(x_{T+1}) \approx \frac{\sum_{n=k}^{T} \mathbb{I}(\forall 0 \leqslant i \leqslant k, \ x_{n-i} = x_{T-i+1})}{\sum_{n=k}^{T} \mathbb{I}(\forall 1 \leqslant i \leqslant k, \ x_{n-i} = x_{T-i+1})}$$

# B  Two-Layer Single-Head Construction

**Theorem 6.** (Two-layer, single-head construction). We assert that there exists a two-layer, single-head attention Transformer model $f_\theta$, which can be configured in one of two equivalent ways: (1) with three MLP layers placed between the first and second attention blocks, or (2) with two MLP layers between the attention blocks, but with layer normalization applied to the query and key projections in the second attention block, such that:

$$f_\theta(s_{1:T})_{s'} \approx \pi(s' \mid s_{1:T}) \quad \forall\, (s_{1:T}, s') \sim \pi, \pi \sim \mathbb{P}_\pi.$$

Note that $\mathbb{P}_\pi$ denotes an arbitrary distribution over trajectories, where each trajectory is generated using a Markovian transition kernel $\pi$. The kernel $\pi$ corresponds to a $k^{\text{th}}$-order Markov chain.

It is important to note that in both configurations, the input embeddings and the first layer (**Layer 1**) remain identical. We choose the input embeddings as follows:

$$\mathbf{x}_n^{(1)} = \texttt{Emb}(x_n) = \begin{bmatrix} \mathbf{0}_{1\times 3} & e_{x_n}^S & \mathbf{0}_{1\times 5S} \end{bmatrix}^T \in \mathbb{R}^{6S+3}$$

In the first layer we use the following relative positional embeddings:

$$\mathbf{p}_i^{(1),K} = \begin{cases} \mathbf{0}, & \text{if } i = 0, \\ (i \cdot \log(3) + \kappa) \cdot \begin{bmatrix} 0 & 1 & \mathbf{0}_{1\times(1+6S)} \end{bmatrix}^T, & \text{if } i \in \{1, 2, \cdots, k\},, \\ \mathbf{0}, & \text{if } i > k \end{cases}$$

and the value embeddings,

$$\mathbf{p}_i^{(1),V} = \begin{cases} 3^i \begin{bmatrix} 1 & \mathbf{0} \end{bmatrix}^T & \text{for } i \leqslant k \\ \mathbf{0} & i > k. \end{cases}$$

**Layer 1.**  Consider the key and query matrices,

$$\mathbf{W}_K^{(1)} = \begin{bmatrix} 0_{1\times 1} & \mathbf{1}_{1\times 2} & \mathbf{0}_{1\times 6S} \\ \mathbf{0}_{(2+6S)\times 1} & \mathbf{0}_{(2+6S)\times 2} & \mathbf{0}_{(2+6S)\times(2+6S)} \end{bmatrix}$$

$$\mathbf{W}_Q^{(1)} = \begin{bmatrix} \mathbf{0}_{1\times 3} & \mathbf{1}_{1\times S} & \mathbf{0}_{1\times 5S} \\ \mathbf{0}_{(2+6S)\times 3} & \mathbf{0}_{(2+6S)\times S} & \mathbf{0}_{(2+6S)\times 5S} \end{bmatrix}$$

On computing (for $i = 1$ to $i = K$)

$$\mathbf{W}_K^{(1)}\texttt{Emb}(x_{n-i}) = \mathbf{0}$$
$$\mathbf{W}_K^{(1)}\mathbf{p}_i^{(1),K} = \begin{bmatrix} (i \cdot \log(3) + \kappa) & \mathbf{0}_{(2+6S)\times 1} \end{bmatrix}^T$$
$$\mathbf{W}_Q^{(1)}\texttt{Emb}(x_n) = \begin{bmatrix} 1_{1\times 1} & \mathbf{0}_{1\times(2+6S)} \end{bmatrix}^T$$

Then, observe that,

$$\left\langle \mathbf{W}_K^{(1)}\left(\texttt{Emb}(x_{n-i}) + \mathbf{p}_i^{(1),K}\right), \mathbf{W}_Q^{(1)}\texttt{Emb}(x_n) \right\rangle = (i \cdot \log(3) + \kappa) \cdot \mathbb{I}(1 \leqslant i \leqslant \min\{n, k\})$$

Letting $\kappa \to \infty$, this results in the attention pattern,

$$\text{att}_{n,n-i}^{(1)} = \frac{3^i \mathbb{I}(1 \leqslant i \leqslant \min\{n,k\})}{\sum_{i'=1}^{\min\{n,k\}} 3^{i'}} = \frac{3^{i-1} \mathbb{I}(1 \leqslant i \leqslant \min\{n,k\})}{\sum_{i'=0}^{\min\{n,k\}-1} 3^{i'}} = \frac{3^{i-1} \mathbb{I}(1 \leqslant i \leqslant \min\{n,k\})}{C_{\texttt{attn}}}$$

Choose the value matrix as,

$$\mathbf{W}_V^{(1)} = \begin{bmatrix} I_{3\times3} & \mathbf{0}_{3\times S} & \mathbf{0}_{3\times4S} \\ \mathbf{0}_{3S\times3} & \mathbf{0}_{S\times S} & \mathbf{0}_{3S\times4S} \\ \mathbf{0}_{S\times3} & I_{S\times S} & \mathbf{0}_{S\times4S} \\ \mathbf{0}_{2S\times3} & \mathbf{0}_{S\times S} & \mathbf{0}_{2S\times4S} \end{bmatrix}$$

The output of the attention layer (with the residual connection) is,

$$\widetilde{\mathbf{x}}_n^{(1)} = \begin{bmatrix} Z_n \mid \mathbf{0}_{1\times2} \mid e_{x_n}^S \mid \mathbf{0}_{1\times2S} \mid \mathbf{v}_n \mid \mathbf{0}_{1\times2S} \end{bmatrix}^T,$$

$$\text{where, } \mathbf{v}_n = \sum_{i=1}^{\min\{n,k\}} \text{att}_{n,n-i} \, e_{x_{n-i}}^S$$

$$Z_n = \sum_{i=1}^{\min\{k,n-1\}} \text{att}_{n,n-i} \, 3^i,$$

It is easy to verify that $Z_n = 3^{k+1}/5$ for $n \geqslant k+1$, and that $Z_n \leqslant 3^k/5$ for $n < k+1$. This distinction will be useful later, as the value of $Z_n$ acts as a signal indicating whether $n \geqslant k+1$ or $n < k+1$. In particular, this allows the subsequent layer to skip attention computations for indices $i \leqslant k$, where the condition $x_n = x_{i-1}, \ldots, x_{n-k+1} = x_{i-k}$ is not well defined.

### B.1   Three Layer MLP followed by the Second Attention Layer

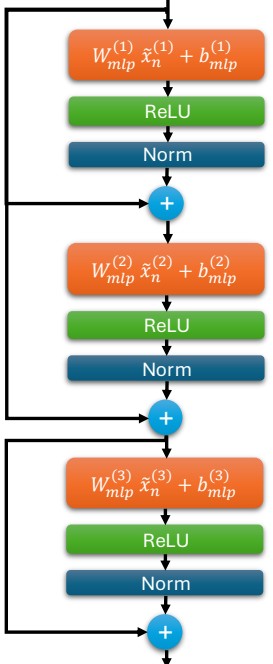

Figure 3: The MLP Architecture

The architecture of the MLP is illustrated in Fig. 3. We now explain how the first layer of the MLP extracts the symbol $e_{x_{n-k}}^S$ from $\widetilde{\mathbf{x}}_n^{(1)}$. To achieve this, the first layer of the MLP is designed as follows:

$$\mathbf{W}_{\mathtt{mlp}}^{(1)} = \begin{bmatrix} \mathbf{0}_{(3+S)\times(3+3S)} & \mathbf{0}_{(3+S)\times(S)} & \mathbf{0}_{(3+S)\times(2S)} \\ \mathbf{0}_{S\times(3+3S)} & \mathbf{A}_{S\times S} & \mathbf{0}_{S\times 2S} \\ \mathbf{0}_{S\times(3+3S)} & \mathbf{0}_{S\times S} & \mathbf{0}_{S\times 2S} \\ \mathbf{0}_{3S\times(3+3S)} & \mathbf{0}_{3S\times S} & \mathbf{0}_{3S\times 2S} \end{bmatrix}$$

$$\mathbf{b}_{\mathtt{mlp}}^{(1)} = \begin{bmatrix} \mathbf{0}_{1\times(3+S)} & -\frac{(3^{k-1}-1)}{2\cdot C_{\mathtt{attn}}}\cdot \mathbf{1}_{1\times S} & \mathbf{0}_{1\times S} & \mathbf{0}_{1\times 2S} \end{bmatrix}^T$$

$$\text{where, } \mathbf{A} = \begin{bmatrix} e_1^T \\ e_2^T \\ \vdots \\ e_S^T \end{bmatrix}$$

Note that on computing, $\hat{\mathbf{x}}_{n,\mathtt{mlp}}^{(1)} = \mathrm{LN}\left(\mathrm{ReLU}\left(\mathbf{W}_{\mathtt{mlp}}^{(1)}\,\widetilde{\mathbf{x}}_n^{(1)} + \mathbf{b}_{\mathtt{mlp}}^{(1)}\right)\right)$, effectively isolates the component associated with the symbol at position $n-k$ in the sequence. This, in turn, allows us to identify the value of $e_{x_{n-k}}^S$ for a given $x_n$. Before demonstrating this explicitly, we first state a useful fact: in a triadic sum representation, the coefficient corresponding to position $n-k$ is always greater than the sum of all coefficients at earlier positions. To illustrate this, consider the scenario in which several consecutive symbols, specifically those from positions $n-1$ to $n-k+1$, are identical. Observe that:

$$\mathrm{att}_{n,n-k} = \frac{3^{k-1}}{C_{\mathtt{attn}}}$$

$$\sum_{i=1}^{k-1} \mathrm{att}_{n,n-i} = \sum_{i=1}^{k-1} \frac{3^{i-1}}{C_{\mathtt{attn}}} = \frac{3^{k-1}-1}{2\cdot C_{\mathtt{attn}}}$$

$$\text{Hence, } \mathrm{att}_{n,n-k} > \sum_{i=1}^{k-1} \mathrm{att}_{n,n-i}$$

The operation $\mathbf{W}_{\mathtt{mlp}}^{(1)}\,\widetilde{\mathbf{x}}_n^{(1)} + \mathbf{b}_{\mathtt{mlp}}^{(1)}$ computes the dot product of each symbol with $\mathbf{v}_n$ and subtracts $\frac{3^{k-1}}{2\cdot C_{\mathtt{attn}}}$ from each coefficient. This ensures that only the $n-k^{\mathrm{th}}$ symbol retains a positive coefficient, while the rest become negative. Passing this result through a ReLU function eliminates the negative coefficients, leaving only the one corresponding to the $n-k^{\mathrm{th}}$ symbol. Normalizing the output sets the coefficient of the $n-k^{\mathrm{th}}$ symbol to one. Concretely:

$$\hat{\mathbf{x}}_{n,\mathtt{mlp}}^{(1)} = \mathbf{W}_{\mathtt{mlp}}^{(1)}\,\widetilde{\mathbf{x}}_n^{(1)} + \mathbf{b}_{\mathtt{mlp}}^{(1)} = \begin{bmatrix} \mathbf{0}_{1\times(3+S)} & \mathbf{c}_{1\times S} & \mathbf{0}_{1\times S} & \mathbf{0}_{1\times 2S} \end{bmatrix}^T$$

$$\mathrm{LN}\left(\mathrm{ReLU}(\hat{\mathbf{x}}_{n,\mathtt{mlp}}^{(1)})\right) = \begin{bmatrix} \mathbf{0}_{1\times(3+S)} & \widetilde{\mathbf{c}}_{1\times S} & \mathbf{0}_{1\times S} & \mathbf{0}_{1\times 2S} \end{bmatrix}^T$$

$$\text{Where, } \widetilde{\mathbf{c}} = \begin{bmatrix} \mathbb{I}(c_1 > \frac{3^{k-1}-1}{2\cdot C_{\mathtt{attn}}}) & \mathbb{I}(c_2 > \frac{3^{k-1}-1}{2\cdot C_{\mathtt{attn}}}) & \cdots & \mathbb{I}(c_S > \frac{3^{k-1}-1}{2\cdot C_{\mathtt{attn}}}) \end{bmatrix}$$

Note that in $\widetilde{\mathbf{c}}$, exactly one entry is equal to 1—specifically, the entry associated with the symbol at position $n-k$; all other entries are zero. After applying the skip connection, the final result becomes:

$$\widetilde{\mathbf{x}}_{n,\mathtt{mlp}}^{(1)} = \widetilde{\mathbf{x}}_n^{(1)} + \mathrm{LN}\left(\mathrm{ReLU}(\hat{\mathbf{x}}_{n,\mathtt{mlp}}^{(1)})\right)$$

$$= \begin{bmatrix} Z_n & \mathbf{0}_{1\times 2} & e_{x_n}^S & \widetilde{\mathbf{c}}_{1\times S} & \mathbf{0}_{1\times S} & \mathbf{v}_n & \mathbf{0}_{1\times 2S} \end{bmatrix}^T$$

The second layer of the MLP is designed to compute $\mathbf{u}_n$ from $\widetilde{\mathbf{x}}_{n,\mathtt{mlp}}^{(1)}$. It does so by extracting $e_{x_n}^S$ and $\mathbf{v}_n$, and leveraging the binary mask $\widetilde{\mathbf{c}}$ together with the vocabulary embeddings stored in the second layer to produce $\mathbf{u}_n$. Hence, we define the second layer as follows:

$$\mathbf{W}_{\mathtt{mlp}}^{(2)} = \begin{bmatrix} \mathbf{0}_{3\times 3} & \mathbf{0}_{3\times S} & \mathbf{0}_{3\times S} & \mathbf{0}_{3\times S} & \mathbf{0}_{3\times S} & \mathbf{0}_{3\times S} & \mathbf{0}_{3\times S} \\ \mathbf{0}_{S\times 3} & \mathbf{0}_{S\times S} & \mathbf{0}_{S\times S} & \mathbf{0}_{S\times S} & \mathbf{0}_{S\times S} & \mathbf{0}_{S\times S} & \mathbf{0}_{S\times S} \\ \mathbf{0}_{S\times 3} & \mathbf{0}_{S\times S} & \mathbf{0}_{S\times S} & \mathbf{0}_{S\times S} & \mathbf{0}_{S\times S} & \mathbf{0}_{S\times S} & \mathbf{0}_{S\times S} \\ \mathbf{0}_{S\times 3} & \frac{1}{C_{\mathtt{attn}}}\cdot I_{S\times S} & -\frac{3^k}{C_{\mathtt{attn}}}\cdot \mathbf{A}_{S\times S}^T & \mathbf{0}_{S\times S} & 3\cdot I_{S\times S} & \mathbf{0}_{S\times S} & \mathbf{0}_{S\times S} \\ \mathbf{0}_{S\times 3} & \mathbf{0}_{S\times S} & \mathbf{0}_{S\times S} & \mathbf{0}_{S\times S} & \mathbf{0}_{S\times S} & \mathbf{0}_{S\times S} & \mathbf{0}_{S\times S} \\ \mathbf{0}_{S\times 3} & \mathbf{0}_{S\times S} & \mathbf{0}_{S\times S} & \mathbf{0}_{S\times S} & \mathbf{0}_{S\times S} & \mathbf{0}_{S\times S} & \mathbf{0}_{S\times S} \\ \mathbf{0}_{S\times 3} & \mathbf{0}_{S\times S} & \mathbf{0}_{S\times S} & \mathbf{0}_{S\times S} & \mathbf{0}_{S\times S} & \mathbf{0}_{S\times S} & \mathbf{0}_{S\times S} \end{bmatrix}$$

$$\mathbf{b}_{\mathtt{mlp}}^{(2)} = \mathbf{0}_{(3+6S)\times 1}$$

Hence, on computing the output, we obtain:

$$\hat{\mathbf{x}}_{n,\mathtt{mlp}}^{(2)} = \mathbf{W}_{\mathtt{mlp}}^{(2)}\tilde{\mathbf{x}}_{n,\mathtt{mlp}}^{(1)} + \mathbf{b}_{\mathtt{mlp}}^{(2)}$$

$$= \begin{bmatrix} 0 & | & \mathbf{0}_{1\times 2} & | & \mathbf{0}_{1\times S} & | & \mathbf{0}_{1\times S} & | & \mathbf{u}_n & | & \mathbf{0}_{1\times S} & | & \mathbf{0}_{1\times 2S} \end{bmatrix}^T$$

$$\text{Where, } \mathbf{u}_n = \sum_{i=0}^{k-1} \frac{3^i}{C_{\mathtt{attn}}}\cdot e_{x_{n-i}}^S = \sum_{i=0}^{k-1} \mathtt{att}_{n,n-i}\cdot e_{x_{n-i}}^S$$

To clarify how $\mathbf{u}_n$ is constructed, we show that it can be obtained directly from $\mathbf{v}_n$, $e_{x_n}^S$, and $e_{x_{n-k}}^S$. The key observation is that $e_{x_{n-k}}^S$ can be recovered using the binary mask $\tilde{\mathbf{c}}$ as follows, $e_{x_{n-k}}^S = \mathbf{A}_{s\times s}^T\tilde{\mathbf{c}}$, where $\mathbf{A}_{s\times s}$ is the vocabulary embedding matrix. This holds because $\tilde{\mathbf{c}}$ is a one-hot vector that selects the embedding corresponding to the symbol at position $n-k$. Using this, the final representation $\mathbf{u}_n$ is computed as:

$$\mathbf{u}_n = \frac{1}{C_{\mathtt{attn}}}e_{x_n^S} + 3\mathbf{v}_n - \frac{3^k}{C_{\mathtt{attn}}}e_{x_{n-k}}^S$$

$$= \frac{1}{C_{\mathtt{attn}}}e_{x_n}^S + \sum_{i=1}^{k} \frac{3^i}{C_{\mathtt{attn}}}e_{x_{n-i}}^S - \frac{3^k}{C_{\mathtt{attn}}}e_{x_{n-k}}^S$$

$$= \sum_{i=0}^{k-1} \frac{3^i}{C_{\mathtt{attn}}}\cdot e_{x_{n-i}}^S$$

Hence, passing this through the normalization and with the skip connections we get:

$$\tilde{\mathbf{x}}_{n,\mathtt{mlp}}^{(2)} = \tilde{\mathbf{x}}_n^{(1)} + \mathrm{LN}\left(\mathrm{ReLU}\left(\mathbf{W}_{\mathtt{mlp}}^{(2)}\tilde{\mathbf{x}}_{n,\mathtt{mlp}}^{(1)} + \mathbf{b}_{\mathtt{mlp}}^{(2)}\right)\right)$$

$$= \begin{bmatrix} Z_n & | & \mathbf{0}_{1\times 2} & | & e_{x_n}^S & | & \mathbf{0}_n & | & \frac{\mathbf{u}_n}{\|\mathbf{u}_n\|_2} & | & \mathbf{v}_n & | & \mathbf{0}_{1\times 2S} \end{bmatrix}^T$$

Finally, on defining the third layer as follows:

$$\mathbf{W}_{\mathtt{mlp}}^{(3)} = \begin{bmatrix} \mathbf{0}_{3\times 3} & \mathbf{0}_{S\times S} & \mathbf{0}_{3\times 4S} \\ \mathbf{0}_{3S\times 3} & \mathbf{0}_{S\times S} & \mathbf{0}_{3S\times 4S} \\ \mathbf{0}_{S\times 3} & I_{S\times S} & \mathbf{0}_{S\times 4S} \\ \mathbf{0}_{2S\times 3} & \mathbf{0}_{S\times S} & \mathbf{0}_{2S\times 4S} \end{bmatrix}$$

$$\mathbf{b}_{\mathtt{mlp}}^{(3)} = \mathbf{0}_{(3+6S)\times 1}$$

Passing the input to the layer through the non-linearities along with the skip connections, we get:

$$\tilde{\mathbf{x}}_{n,\mathtt{mlp}}^{(3)} = \tilde{\mathbf{x}}_{n,\mathtt{mlp}}^{(2)} + \mathrm{LN}\left(\mathrm{ReLU}\left(\mathbf{W}_{\mathtt{mlp}}^{(2)}\tilde{\mathbf{x}}_{n,\mathtt{mlp}}^{(1)} + \mathbf{b}_{\mathtt{mlp}}^{(2)}\right)\right)$$

$$= \begin{bmatrix} Z_n & | & \mathbf{0}_{1\times 2} & | & e_{x_n}^S & | & 0_{1\times S} & | & \frac{\mathbf{u}_n}{\|\mathbf{u}_n\|_2} & | & \mathbf{v}_n & | & \frac{\mathbf{v}_n}{\|\mathbf{v}_n\|_2} & | & \mathbf{0}_{1\times S} \end{bmatrix}^T$$

**Layer 2.** In this layer, all relative position encodings are set to $\mathbf{0}$, and we use the following query and key matrices for the attention mechanism in the second layer:

$$
\mathbf{W}_Q^{(2)} = \sqrt{\kappa} \begin{bmatrix} 1 & \mathbf{0} & \mathbf{0} & \mathbf{0} \\ \mathbf{0} & \mathbf{0}_{S \times (2+3S)} & I_{S \times S} & \mathbf{0} \\ \mathbf{0} & \mathbf{0} & \mathbf{0} & \mathbf{0} \end{bmatrix}
$$

$$
\mathbf{W}_K^{(2)} = \sqrt{\kappa} \begin{bmatrix} 1 & \mathbf{0} & \mathbf{0} & \mathbf{0} \\ \mathbf{0} & \mathbf{0}_{S \times (2+4S)} & I_{S \times S} & \mathbf{0} \\ \mathbf{0} & \mathbf{0} & \mathbf{0} & \mathbf{0} \end{bmatrix}
$$

With these choices in place, we obtain:

$$
\left\langle (\mathbf{W}_K^{(2)} \widetilde{\mathbf{x}}_{i,\texttt{mlp}}^{(3)}), (\mathbf{W}_Q^{(2)} \widetilde{\mathbf{x}}_{n,\texttt{mlp}}^{(3)}) \right\rangle = \kappa \cdot Z_i Z_n + \frac{\kappa \cdot \langle \mathbf{v}_i, \mathbf{u}_n \rangle}{\|\mathbf{v}_i\|_2 \cdot \|\mathbf{u}_n\|_2} = \kappa \cdot Z_i Z_n + 2 - \kappa \cdot \left\| \frac{\mathbf{v}_i}{\|\mathbf{v}_i\|_2} - \frac{\mathbf{u}_n}{\|\mathbf{u}_n\|_2} \right\|^2
$$

Applying the softmax function, we obtain:

$$
\text{att}_{n,i}^{(2)} = \frac{\exp\left(\kappa Z_i Z_n + 2\kappa - \kappa \left\| \frac{\mathbf{v}_i}{\|\mathbf{v}_i\|_2} - \frac{\mathbf{u}_n}{\|\mathbf{u}_n\|_2} \right\|^2 \right)}{\sum_{j=0}^{n} \exp\left(\kappa Z_j Z_n + 2\kappa - \kappa \left\| \frac{\mathbf{v}_j}{\|\mathbf{v}_j\|_2} - \frac{\mathbf{u}_n}{\|\mathbf{u}_n\|_2} \right\|^2 \right)}
$$

We can see that for all $k+1 \leqslant i \leqslant n$, $\mathbf{v}_i = \mathbf{u}_n$ if and only if the local context around position $i$ matches that of position $n$, i.e., $\{x_{i-1-j} = x_{n-j} \text{ for all } j = 0, \ldots, k\}$. Moreover, for any $k+1 \leqslant i \leqslant n$, if $\mathbf{v}_i \neq \mathbf{u}_n$, then the normalized vectors are separated by a non-negligible margin:

$$
\left\| \frac{\mathbf{v}_i}{\|\mathbf{v}_i\|_2} - \frac{\mathbf{u}_n}{\|\mathbf{u}_n\|_2} \right\|_2 \geqslant \frac{1}{3^k}.
$$

Note that while this gap is small, it is strictly non-zero. Recall that $Z_i = \frac{3^{k+1}}{5}$ for $i \geqslant k+1$, and $Z_i \leqslant \frac{3^k}{5}$ otherwise. As a result, the attention mechanism favors positions $i$ such that $\mathbf{v}_i = \mathbf{u}_n$ and $i \geqslant k+1$. In the limit as $\kappa \to \infty$, this results in the following attention pattern:

$$
\text{att}_{n,\cdot}^{(2)} = \text{Unif}(\mathcal{I}_n),
$$

where $\mathcal{I}_n$ denotes the set of all indices $i \in \{k+1, \ldots, n\}$ whose local context matches that of position $n$, formally defined as:

$$
\mathcal{I}_n := \{i \in \{k, \ldots, n\} \mid x_{i-1-j} = x_{n-j} \text{ for all } j = 0, \ldots, k\}.
$$

Moreover, $\text{Unif}(\mathcal{I}_n)$ denotes the uniform distribution over the indices in $\mathcal{I}_n$; that is, each position $i \in \mathcal{I}_n$ receives equal attention weight, and all other positions receive zero. We now choose the value projection matrix in the second attention layer as:

$$
\mathbf{W}_V^{(2)} = \begin{bmatrix} \mathbf{0} & \mathbf{0} & \mathbf{0} \\ \mathbf{0}_{S \times 3} & I_{S \times S} & \mathbf{0} \end{bmatrix}.
$$

With this choice, the output of the second attention layer becomes:

$$
\widetilde{\mathbf{x}}_n^{(2)} = \widetilde{\mathbf{x}}_{n,\texttt{mlp}}^{(2)} + \sum_{i=0}^{n} \text{att}_{n,i}^{(2)} \begin{bmatrix} \mathbf{0} \\ e_{x_i}^S \end{bmatrix}
$$

$$= \widetilde{\mathbf{x}}_{n,\mathtt{mlp}}^{(2)} + \frac{1}{|\mathcal{I}_n|} \sum_{i \in \mathcal{I}_n} \begin{bmatrix} \mathbf{0} \\ e_{x_i}^S \end{bmatrix}.$$

For the output linear layer, we choose:

$$\mathbf{W}_o = \begin{bmatrix} \mathbf{0}_{S \times (5S+3)} & I_{S \times S} \end{bmatrix}, \qquad \mathbf{b}_o = \mathbf{0}.$$

This yields the output logits:

$$\mathrm{logit}_n = \frac{1}{|\mathcal{I}_n|} \sum_{i \in \mathcal{I}_n} e_{x_i}^S = \sum_{i=k}^{n} \frac{\mathbb{I}\left(\forall\, 1 \leqslant j \leqslant k,\ x_{i-j} = x_{n-j+1}\right)}{\sum_{i'=k}^{n} \mathbb{I}\left(\forall\, 1 \leqslant j \leqslant k,\ x_{i'-j} = x_{n-j+1}\right)} \cdot e_{x_i}^S.$$

Finally, by adjusting the notation and setting $n = T$ to reflect prediction at the final time step, we obtain:

$$\mathrm{logit}_T(x_{T+1}) = \frac{\sum_{n=k}^{T} \mathbb{I}\left(\forall\, 0 \leqslant i \leqslant k,\ x_{n-i} = x_{T-i+1}\right)}{\sum_{n=k}^{T} \mathbb{I}\left(\forall\, 1 \leqslant i \leqslant k,\ x_{n-i} = x_{T-i+1}\right)},$$

which precisely computes the conditional $k$-gram probability.

## B.2 Two Layer MLP followed by the Second Attention Layer

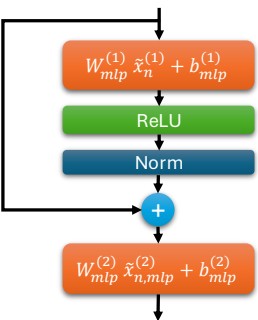

Figure 4: The MLP Architecture

We restate the output of the first attention layer:

$$\widetilde{\mathbf{x}}_n^{(1)} = \begin{bmatrix} Z_n & \mid & \mathbf{0}_{1 \times 2} & \mid & e_{x_n}^S & \mid & \mathbf{0}_{1 \times 2S} & \mid & \mathbf{v}_n & \mid & \mathbf{0}_{1 \times 2S} \end{bmatrix}^T,$$

$$\text{where, } \mathbf{v}_n = \sum_{i=1}^{\min\{n,k\}} \mathrm{att}_{n,n-i}\, e_{x_{n-i}}^S$$

$$Z_n = \sum_{i=1}^{\min\{k,n-1\}} \mathrm{att}_{n,n-i}\, 3^i,$$

The architecture of the MLP is illustrated in Fig. 4. We now explain how the first layer of the MLP extracts the symbol $e_{x_{n-k}}^S$ from $\widetilde{\mathbf{x}}_n^{(1)}$. To achieve this, the first layer of the MLP is designed as follows:

$$\mathbf{W}_{\mathtt{mlp}}^{(1)} = \begin{bmatrix} \mathbf{0}_{(3+S)\times(3+3S)} & \mathbf{0}_{(3+S)\times(S)} & \mathbf{0}_{(3+S)\times(2S)} \\ \mathbf{0}_{S\times(3+3S)} & \mathbf{A}_{S\times S} & \mathbf{0}_{S\times2S} \\ \mathbf{0}_{S\times(3+3S)} & \mathbf{0}_{S\times S} & \mathbf{0}_{S\times2S} \\ \mathbf{0}_{3S\times(3+3S)} & \mathbf{0}_{3S\times S} & \mathbf{0}_{3S\times2S} \end{bmatrix}$$

$$\mathbf{b}_{\mathtt{mlp}}^{(1)} = \begin{bmatrix} \mathbf{0}_{1\times(3+S)} & -\frac{(3^{k-1}-1)}{2\cdot C_{\mathtt{attn}}}\cdot\mathbf{1}_{1\times S} & \mathbf{0}_{1\times S} & \mathbf{0}_{1\times 2S} \end{bmatrix}^{T}$$

$$\text{where, } \mathbf{A} = \begin{bmatrix} e_1^T \\ e_2^T \\ \vdots \\ e_S^T \end{bmatrix}$$

Note that on computing, $\hat{\mathbf{x}}_{n,\mathtt{mlp}}^{(1)} = \mathrm{LN}\left(\mathrm{ReLU}\left(\mathbf{W}_{\mathtt{mlp}}^{(1)}\,\widetilde{\mathbf{x}}_n^{(1)} + \mathbf{b}_{\mathtt{mlp}}^{(1)}\right)\right)$, effectively isolates the component associated with the symbol at position $n - k$ in the sequence. This, in turn, allows us to identify the value of $e_{x_{n-k}}^S$ for a given $x_n$. Before demonstrating this explicitly, we first state a useful fact: in a triadic sum representation, the coefficient corresponding to position $n - k$ is always greater than the sum of all coefficients at earlier positions. To illustrate this, consider the scenario in which several consecutive symbols, specifically those from positions $n - 1$ to $n - k + 1$, are identical. Observe that:

$$\mathrm{att}_{n,n-k} = \frac{3^{k-1}}{C_{\mathtt{attn}}}$$

$$\sum_{i=1}^{k-1} \mathrm{att}_{n,n-i} = \sum_{i=1}^{k-1} \frac{3^{i-1}}{C_{\mathtt{attn}}} = \frac{3^{k-1}-1}{2\cdot C_{\mathtt{attn}}}$$

$$\text{Hence, } \mathrm{att}_{n,n-k} > \sum_{i=1}^{k-1} \mathrm{att}_{n,n-i}$$

The operation $\mathbf{W}_{\mathtt{mlp}}^{(1)}\,\widetilde{\mathbf{x}}_n^{(1)} + \mathbf{b}_{\mathtt{mlp}}^{(1)}$ computes the dot product of each symbol with $\mathbf{v}_n$ and subtracts $\frac{3^{k-1}}{2\cdot C_{\mathtt{attn}}}$ from each coefficient. This ensures that only the $n - k^{\text{th}}$ symbol retains a positive coefficient, while the rest become negative. Passing this result through a ReLU function eliminates the negative coefficients, leaving only the one corresponding to the $n - k^{\text{th}}$ symbol. Normalizing the output sets the coefficient of the $n - k^{\text{th}}$ symbol to one. Concretely:

$$\hat{\mathbf{x}}_{n,\mathtt{mlp}}^{(1)} = \mathbf{W}_{\mathtt{mlp}}^{(1)}\,\widetilde{\mathbf{x}}_n^{(1)} + \mathbf{b}_{\mathtt{mlp}}^{(1)} = \begin{bmatrix} \mathbf{0}_{1\times(3+S)} & \mathbf{c}_{1\times S} & \mathbf{0}_{1\times S} & \mathbf{0}_{1\times 2S} \end{bmatrix}^{T}$$

$$\mathrm{LN}\left(\mathrm{ReLU}(\hat{\mathbf{x}}_{n,\mathtt{mlp}}^{(1)})\right) = \begin{bmatrix} \mathbf{0}_{1\times(3+S)} & \widetilde{\mathbf{c}}_{1\times S} & \mathbf{0}_{1\times S} & \mathbf{0}_{1\times 2S} \end{bmatrix}^{T}$$

$$\text{Where, } \widetilde{\mathbf{c}} = \begin{bmatrix} \mathbb{I}(c_1 > \frac{3^{k-1}-1}{2\cdot C_{\mathtt{attn}}}) & \mathbb{I}(c_2 > \frac{3^{k-1}-1}{2\cdot C_{\mathtt{attn}}}) & \cdots & \mathbb{I}(c_S > \frac{3^{k-1}-1}{2\cdot C_{\mathtt{attn}}}) \end{bmatrix}$$

Note that in $\widetilde{\mathbf{c}}$, exactly one entry is equal to 1—specifically, the entry associated with the symbol at position $n - k$; all other entries are zero. After applying the skip connection, the final result becomes:

$$\widetilde{\mathbf{x}}_{n,\mathtt{mlp}}^{(1)} = \widetilde{\mathbf{x}}_n^{(1)} + \mathrm{LN}\left(\mathrm{ReLU}(\hat{\mathbf{x}}_{n,\mathtt{mlp}}^{(1)})\right)$$

$$= \begin{bmatrix} Z_n & | & \mathbf{0}_{1\times 2} & | & e_{x_n}^S & | & \widetilde{\mathbf{c}}_{1\times S} & | & \mathbf{0}_{1\times S} & | & \mathbf{v}_n & | & \mathbf{0}_{1\times 2S} \end{bmatrix}^{T}$$

The second layer of the MLP is designed to compute $\mathbf{u}_n$ from $\widetilde{\mathbf{x}}_{n,\mathtt{mlp}}^{(1)}$. It does so by extracting $e_{x_n}^S$ and $\mathbf{v}_n$, and leveraging the binary mask $\widetilde{\mathbf{c}}$ together with the vocabulary embeddings stored in the second layer to produce $\mathbf{u}_n$. Hence, we define the second layer as follows:

$$\mathbf{W}_{\texttt{mlp}}^{(2)} = \begin{bmatrix} I_{3\times 3} & \mathbf{0}_{3\times S} & \mathbf{0}_{3\times S} & \mathbf{0}_{3\times S} & \mathbf{0}_{3\times S} & \mathbf{0}_{3\times S} & \mathbf{0}_{3\times S} \\ \mathbf{0}_{S\times 3} & I_{S\times S} & \mathbf{0}_{S\times S} & \mathbf{0}_{S\times S} & \mathbf{0}_{S\times S} & \mathbf{0}_{S\times S} & \mathbf{0}_{S\times S} \\ \mathbf{0}_{S\times 3} & \frac{1}{C_{\texttt{attn}}}\cdot I_{S\times S} & -\frac{3^k}{C_{\texttt{attn}}}\cdot \mathbf{A}_{S\times S}^T & \mathbf{0}_{S\times S} & 3\cdot I_{S\times S} & \mathbf{0}_{S\times S} & \mathbf{0}_{S\times S} \\ \mathbf{0}_{S\times 3} & \mathbf{0}_{S\times S} & \mathbf{0}_{S\times S} & \mathbf{0}_{S\times S} & \mathbf{0}_{S\times S} & \mathbf{0}_{S\times S} & \mathbf{0}_{S\times S} \\ \mathbf{0}_{S\times 3} & \mathbf{0}_{S\times S} & \mathbf{0}_{S\times S} & \mathbf{0}_{S\times S} & I_{S\times S} & \mathbf{0}_{S\times S} & \mathbf{0}_{S\times S} \\ \mathbf{0}_{S\times 3} & \mathbf{0}_{S\times S} & \mathbf{0}_{S\times S} & \mathbf{0}_{S\times S} & \mathbf{0}_{S\times S} & \mathbf{0}_{S\times S} & \mathbf{0}_{S\times S} \\ \mathbf{0}_{S\times 3} & \mathbf{0}_{S\times S} & \mathbf{0}_{S\times S} & \mathbf{0}_{S\times S} & \mathbf{0}_{S\times S} & \mathbf{0}_{S\times S} & \mathbf{0}_{S\times S} \end{bmatrix}$$

$$\mathbf{b}_{\texttt{mlp}}^{(2)} = \mathbf{0}_{(3+6S)\times 1}$$

Hence, on computing the output, we obtain:

$$\widetilde{\mathbf{x}}_{n,\texttt{mlp}}^{(2)} = \mathbf{W}_{\texttt{mlp}}^{(2)}\widetilde{\mathbf{x}}_{n,\texttt{mlp}}^{(1)} + \mathbf{b}_{\texttt{mlp}}^{(2)}$$

$$= \begin{bmatrix} Z_n & | & \mathbf{0}_{1\times 2} & | & e_{x_n}^S & | & \mathbf{u}_n & | & \mathbf{0}_{1\times S} & | & \mathbf{v}_n & | & \mathbf{0}_{1\times 2S} \end{bmatrix}^T$$

$$\text{Where, } \mathbf{u}_n = \sum_{i=0}^{k-1}\frac{3^i}{C_{\texttt{attn}}}\cdot e_{x_{n-i}}^S = \sum_{i=0}^{k-1}\text{att}_{n,n-i}\cdot e_{x_{n-i}}^S$$

To clarify how $\mathbf{u}_n$ is constructed, we show that it can be obtained directly from $\mathbf{v}_n$, $e_{x_n}^S$, and $e_{x_{n-k}}^S$. The key observation is that $e_{x_{n-k}}^S$ can be recovered using the binary mask $\widetilde{\mathbf{c}}$ as follows, $e_{x_{n-k}}^S = \mathbf{A}_{s\times s}^T\widetilde{\mathbf{c}}$, where $\mathbf{A}_{s\times s}$ is the vocabulary embedding matrix. This holds because $\widetilde{\mathbf{c}}$ is a one-hot vector that selects the embedding corresponding to the symbol at position $n-k$. Using this, the final representation $\mathbf{u}_n$ is computed as:

$$\mathbf{u}_n = \frac{1}{C_{\texttt{attn}}}e_{x_n^S} + 3\mathbf{v}_n - \frac{3^k}{C_{\texttt{attn}}}e_{x_{n-k}}^S$$

$$= \frac{1}{C_{\texttt{attn}}}e_{x_n}^S + \sum_{i=1}^{k}\frac{3^i}{C_{\texttt{attn}}}e_{x_{n-i}}^S - \frac{3^k}{C_{\texttt{attn}}}e_{x_{n-k}}^S$$

$$= \sum_{i=0}^{k-1}\frac{3^i}{C_{\texttt{attn}}}\cdot e_{x_{n-i}}^S$$

**Layer 2.** In this layer, all the relative position encodings are set as $\mathbf{0}$ and instead,

$$\mathbf{W}_Q^{(2)} = \begin{bmatrix} 0 & \mathbf{0} & \mathbf{0} & \mathbf{0} \\ \mathbf{0} & \mathbf{0}_{S\times(2+3S)} & I_{S\times S} & \mathbf{0} \\ \mathbf{0} & \mathbf{0} & \mathbf{0} & \mathbf{0} \end{bmatrix}$$

$$\mathbf{W}_K^{(2)} = \begin{bmatrix} 0 & \mathbf{0} & \mathbf{0} & \mathbf{0} \\ \mathbf{0} & \mathbf{0}_{S\times(2+4S)} & I_{S\times S} & \mathbf{0} \\ \mathbf{0} & \mathbf{0} & \mathbf{0} & \mathbf{0} \end{bmatrix}$$

With these choices, and adding a layer normalization after the output of the key linear layer and the query linear layer

$$\left\langle \text{LN}(\mathbf{W}_K^{(2)}\widetilde{\mathbf{x}}_{i,\texttt{mlp}}^{(2)}), \text{LN}(\mathbf{W}_Q^{(2)}\widetilde{\mathbf{x}}_{n,\texttt{mlp}}^{(2)}) \right\rangle = \frac{2\langle \mathbf{v}_i, \mathbf{u}_n\rangle}{\|\mathbf{v}_i\|_2\cdot\|\mathbf{u}_n\|_2}$$

$$= 2 - \left\| \frac{\mathbf{v}_i}{\|\mathbf{v}_i\|_2} - \frac{\mathbf{u}_n}{\|\mathbf{u}_n\|_2} \right\|^2$$

Taking the softmax with a temperature parameter $\kappa$,

$$\text{att}_{n,i}^{(2)} = \frac{\exp\left(2\kappa - \kappa \left\| \frac{\mathbf{v}_i}{\|\mathbf{v}_i\|_2} - \frac{\mathbf{u}_n}{\|\mathbf{u}_n\|_2} \right\|^2\right)}{\sum_{j=0}^{n} \exp\left(2\kappa - \kappa \left\| \frac{\mathbf{v}_j}{\|\mathbf{v}_j\|_2} - \frac{\mathbf{u}_n}{\|\mathbf{u}_n\|_2} \right\|^2\right)}$$

Note that although this gap is small, it remains non-zero. In particular, as $\kappa \to \infty$, the resulting attention pattern approaches

$$\text{att}_{n,\cdot}^{(2)} = \text{Unif}(\mathcal{I}_n \cup \hat{\mathcal{I}}_k),$$

where $\mathcal{I}_n$ denotes the set of indices $i \in \{k, \dots, n\}$ whose local context matches that of position $n$. Formally,

$$\mathcal{I}_n := \{i \in \{k, \dots, n\} \mid x_{i-1-j} = x_{n-j} \text{ for all } j = 0, \dots, k\}.$$

In contrast, $\hat{\mathcal{I}}_k$ captures earlier positions $i \in \{0, \dots, k-1\}$ whose (normalized) key vectors are aligned with the normalized query at position $n$:

$$\hat{\mathcal{I}}_k := \left\{ i \in \{0, \dots, k-1\} \,\middle|\, \frac{\mathbf{v}_i}{\|\mathbf{v}_i\|_2} = \frac{\mathbf{k}_n}{\|\mathbf{u}_n\|_2} \right\}.$$

On choosing,

$$\mathbf{W}_V^{(2)} = \begin{bmatrix} \mathbf{0} & \mathbf{0} & \mathbf{0} \\ \mathbf{0}_{S \times 3} & I_{S \times S} & \mathbf{0} \end{bmatrix}.$$

We obtain,

$$\tilde{\mathbf{x}}_n^{(2)} = \mathbf{x}_n^{(2)} + \sum_{i=1}^{n} \text{att}_{n,i}^{(2)} \begin{bmatrix} \mathbf{0} \\ e_{x_i}^S \end{bmatrix} = \mathbf{x}_n^{(3)} + \frac{1}{|\mathcal{I}_n| + |\hat{\mathcal{I}}_k|} \sum_{i \in \mathcal{I}_n \cup \hat{\mathcal{I}}_k} \begin{bmatrix} \mathbf{0} \\ e_{x_i}^S \end{bmatrix}.$$

For the output linear layer, we choose:

$$\mathbf{W}_o = \begin{bmatrix} \mathbf{0}_{S \times (5S+3)} & I_{S \times S} \end{bmatrix}, \qquad \mathbf{b}_o = \mathbf{0}.$$

which results in,

$$\begin{aligned}
\text{logit}_n &= \frac{1}{|\mathcal{I}_n| + |\hat{\mathcal{I}}_k|} \sum_{i \in \mathcal{I}_n \cup \hat{\mathcal{I}}_k} e_{x_i}^S = \sum_{i=k}^{n} \frac{\mathbb{I}(\forall 1 \le j \le k, \; x_{i-j} = x_{n-j+1})}{\sum_{i'=k}^{n} \mathbb{I}(\forall 1 \le j \le k, \; x_{i'-j} = x_{n-j+1}) + \hat{\mathcal{I}}_k} \cdot e_{x_i}^S \\
&\quad + \sum_{i \in \hat{\mathcal{I}}_k} \frac{\mathbb{I}(\forall 1 \le j \le k, \; x_{i-j} = x_{n-j+1})}{\sum_{i'=k}^{n} \mathbb{I}(\forall 1 \le j \le k, \; x_{i'-j} = x_{n-j+1}) + \hat{\mathcal{I}}_k} \cdot e_{x_i}^S
\end{aligned}$$

Now, by substituting $n = T$, and aiming to predict the next symbol given the current context, we obtain:

$$\text{logit}_T(x_{T+1}) = \frac{\sum_{n=k}^{T} \mathbb{I}(\forall 0 \le i \le k, \; x_{n-i} = x_{T-i+1})}{\sum_{n=k}^{T} \mathbb{I}(\forall 1 \le i \le k, \; x_{n-i} = x_{T-i+1}) + \hat{\mathcal{I}}_k}$$

$$+ \frac{\sum_{i \in \hat{\mathcal{I}}_k} \langle e^S_{T+1}, e^S_i \rangle}{\sum_{n=k}^{T} \mathbb{I}(\forall 1 \leqslant i \leqslant k, \ x_{n-i} = x_{T-i+1}) + \hat{\mathcal{I}}_k}$$

This corresponds to a *biased conditional* $k$-gram model, where the bias originates from the second term in the expression above. Notably, this bias vanishes in the limit as $T \to \infty$. The reason is that the numerator of the second term remains constant with respect to sequence length, since $\hat{\mathcal{I}}_k$ is fixed and does not scale with $T$. In contrast, the denominator, given by $\sum_{n=k}^{T} \mathbb{I}(\forall\, 1 \leqslant i \leqslant k, \ x_{n-i} = x_{T-i+1})$, grows with $T$, assuming the underlying Markov chain is irreducible (i.e., every state is reachable from every other state). Consequently, under this assumption of irreducibility, the influence of the second term diminishes as $T$ becomes large, yielding the approximation for a sufficiently large $T$):

$$\text{logit}_T(x_{T+1}) \approx \frac{\sum_{n=k}^{T} \mathbb{I}(\forall 0 \leqslant i \leqslant k, \ x_{n-i} = x_{T-i+1})}{\sum_{n=k}^{T} \mathbb{I}(\forall 1 \leqslant i \leqslant k, \ x_{n-i} = x_{T-i+1})}$$

# C   Two-Layer Single-Head Gradient Analysis

As discussed in Sec. 5, we analyze a simplified model that retains only the key components of the two-layer architecture described in Thm. 3—specifically, the positional encodings and the attention layer—while treating the remaining components as approximately optimal. This reduction enables a tractable yet faithful reproduction of the in-context learning (ICL) behavior exhibited by the full model. Our approach is closely related to the reparameterization techniques used by Makkuva et al. [18] and Nichani et al. [20] in the context of gradient flow and gradient descent analysis. Alternatively, it can be viewed as a form of *optimal parameter initialization*, in the spirit of Edelman et al. [10] and Nichani et al. [20]. We reiterate the assumptions and provide additional details whenever necessary.

**Data Assumptions.** Following Nichani et al. [20], we assume a prior distribution $\mathbb{P}_\pi(\cdot)$ over irreducible and aperiodic Markov transition kernels $\pi$, such that:

> **Assumption 1.** (Assumption on $\mathbb{P}_\pi$): We consider a prior distribution over transition matrices of first-order aperiodic and irreducible Markov chains such that there exists a constant $\gamma > 0$ for which, over transition kernels drawn from $\mathbb{P}_\pi(\cdot)$, the following conditions are satisfied:
>
> 1. Minimum transition probability: $\min\limits_{s,s'} \pi(s' \mid s) > \dfrac{\gamma}{S}$
>
> 2. Non-trivial mixing: $\sum\limits_s \|\pi(\cdot \mid s) - \mu_\pi(\cdot)\|_2^2 \geqslant \dfrac{\gamma^2}{S}$
>
> 3. Permutation invariance: For any permutation $\sigma$ on $[S]$, $\sigma^{-1}\pi\sigma \overset{d}{=} \pi$
>
> 4. Uniform expected transition matrix: $\mathbb{E}_\pi[\pi] = \dfrac{1}{S}\mathbf{1}_S\mathbf{1}_S^\top$

**Model Simplifications**   We simplify the model as follows. More details can be found in App. C.1.

(i) The positional embeddings used for the keys in the first attention layer are scalar-valued, i.e. $\mathbf{p}_n^{(1,K)} \propto p_n$, where $p_n$ is a scalar.

(ii) The query, key and value matrices ($\mathbf{W}_{Q,K,V}^{(1)}$) and positional encodings for the value ($\mathbf{p}^{(1,V)}$) for the first layer, are chosen so that the attention weight for position $n - i$ is given by $\text{att}_{n,n-i}^{(1)} = \exp(p_i)/\left(\sum_{j=0}^i \exp(p_j)\right) \in \{0,1\}$, and the corresponding value output is $\mathbf{v}_n = \sum_{i=0}^n \text{att}_{n,n-i}^{(1)} \cdot e_{x_{n-i}}^S$.

(iii) We assume the MLP is optimal—i.e., it outputs the correct $\mathbf{u}_n$ even from suboptimal $\mathbf{v}_n$; this holds for first-order Markov chains via skip connections.

(iv) The query and key matrices for the second layer ($\mathbf{W}_{Q,K}^{(2)}$) are chosen such that the attention in this layer is defined as $\text{att}_{T,i}^{(2)} = \exp\left(a_2 \cdot \langle \mathbf{v}_i, \mathbf{u}_T \rangle\right)/\left(\sum_{j=0}^T \exp(a_2 \cdot \langle \mathbf{v}_j, \mathbf{u}_n \rangle)\right)$, where $a_2 \in \mathbb{R}$ is the attention scalar.

(v) $\mathbf{W}_V^{(2)}$ is chosen so that the final logit $\text{logit}_T = \sum_{i=0}^T \text{att}_{n,i}^{(2)} e_{x_i} \in \mathbb{R}^S$,

**Loss function.** We minimize the cross-entropy loss for next-token prediction [20]:

$$\mathcal{L}(\theta) = -\mathbb{E}_{\pi \sim \mathbb{P}_\pi, \, x_{0:T} \sim \pi}\left[\sum_{s \in \mathcal{S}} \pi\left(x_{T+1} = s \mid x_{0:T}\right) \log\left(\text{logit}_T(s) + \varepsilon\right)\right], \tag{2}$$

where $\varepsilon > 0$ is a small additive constant, $\mathbb{P}_\pi(\cdot)$ denotes the prior over Markov transition kernels, and $\text{logit}_T(s)$ is the logit corresponding to the symbol $s$.

**Training algorithm.** We denote the set of trainable parameters as $\theta = (\mathbf{p}, a_2)$, where $\mathbf{p} = [p_0, p_1, \ldots, p_T]^T$ represents the positional scalars and $a_2$ is the attention scalar. Following the approach of [20], we employ a two-stage training procedure. (i) In the first stage, we optimize the positional scalars $\mathbf{p}$ using gradient descent for $T_1$ steps with a learning rate $\eta_1$; (ii) in the second

stage, we freeze $\mathbf{p}$ and train only $a_2$ for an additional $T_2$ steps using a distinct learning rate $\eta_2$. A key difference in our setup lies in the treatment of layer normalization: during the first stage, all layer norms in the MLP are omitted, whereas in the second stage, they are reintroduced. This stage-wise strategy facilitates a more tractable theoretical analysis. The training algorithm is:

---

**Algorithm 1.** (Training Algorithm).

    **Input:** learning rates $\eta_1$, $\eta_2$; steps $T_1$, $T_2$; small scalar $a_{2,0}$

    Initialize $p_i^{(0)} = 0$ for all $i \in \{0, \ldots, n\}$; $a_2 = a_{2,0}$;
    Freeze all other parameters of the network
    **Stage 1: Train $p_i$ without norms added to the MLP**
    **for** $t = 1, \ldots, T_1$ **do**
        $\mathbf{p}^{(t)} \leftarrow \mathbf{p}^{(t-1)} - \eta_1 \cdot \nabla_{\mathbf{p}} \mathcal{L}(\theta^{(t-1)})$
        $\theta^{(t)} \leftarrow \left( \{p_i^{(t)}\}_{i=0}^n, a_2 = a_{2,0} \right)$
    **end for**
    **Stage 2: Train $a_2$ with norms added to the MLP**
    **for** $t = T_1 + 1, \ldots, T_1 + T_2$ **do**
        $a_2^{(t)} \leftarrow a_2^{(t-1)} - \eta_2 \cdot \nabla_{a_2} \mathcal{L}(\theta^{(t-1)})$
        $\theta^{(t)} \leftarrow \left( \{p_i^{(T_1)}\}_{i=0}^n, a_2^{(t)} \right)$
    **end for**
    $\hat{\theta} \leftarrow \theta^{(T_1 + T_2)}$
    **Output:** $\hat{\theta}$

---

Under the assumptions outlined above, we restate the main theorem presented in Section 5. In addition, we introduce a new theorem that addresses the generalization properties of the model in the inductive learning setting.

---

**Theorem 7.** (Convergence of the training Algorithm). Suppose that Assumptions (i)- (vi) hold and that the Markov sequence length $T$ satisfies $T + 1 \geqslant \mathrm{poly}(\gamma^{-1}, S)$ for some constant $\gamma > 0$. Then there exist $\varepsilon > 0$, learning rates $\eta_1, \eta_2$, and step counts $T_1, T_2$, such that the output of the above two-stage training algorithm, $\hat{\theta} = (\{\hat{p}_i\}_{i=0}^T, \hat{a}_2)$, satisfies

$$\mathcal{L}(\hat{\theta}) - \mathcal{L}^* \lesssim \frac{\log(T+1)}{(T+1)^{c\gamma}},$$

for a constant $c$ independent of $(\gamma, S)$, and the optimal loss $\mathcal{L}^* := -\mathbb{E}\left[ (1/S) \sum_{s,s'} \pi(s' \mid s) \log \pi(s' \mid s) \right]$.

---

**Theorem 8.** (Inductive Generalization): Let $\widetilde{\pi}$ be a transition matrix satisfying $\min_{s,s'} \widetilde{\pi}(s' \mid s) \geqslant \gamma/S$, and let $\hat{\theta}$ denote the trained model from Thm. 7. Consider a sequence $s_0, \ldots, s_n$ generated according to $\widetilde{\pi}$. Then the model satisfies:

$$\sup_{s'} \left| f_{\hat{\theta}}(s_0, \ldots, s_n)_{s'} - \widetilde{\pi}(s' \mid s_n) \right| \lesssim \frac{\log(n+1)}{(n+1)^{c\gamma}}.$$

---

This section is organized as follows. We begin by presenting the simplified construction in App. C.1. Next, we demonstrate that during the first stage of training C.2, the model converges to the optimal attention map for the first layer—specifically, it learns to attend to the previous token, which is optimal under a first-order Markov chain. In the second stage of training C.3, the model sharpens its attention, ultimately converging asymptotically to a near-deterministic distribution over the desired token. Finally, we provide arguments for the result in the context of inductive generalization, utilizing results from Nichani et al. [20]. We would like to note that our analysis for first-order processes is similar to that of Nichani et al. [20], but the architecture is fundamentally different.

## C.1 Model Simplification

We adopt a low-parameter regime by isolating the core components of the Transformer architecture that are essential to our analysis, while keeping all remaining parameters fixed at their optimal values. Here, optimality is defined with respect to a first order Markov chain setting. Under this setup, we proceed by stating the following assumptions.

$$\mathbf{x}_n^{(1)} = \texttt{Emb}(x_n) = \begin{bmatrix} \mathbf{0}_{1\times 3} & e_{x_n}^S & \mathbf{0}_{1\times 5S} \end{bmatrix}^T \in \mathbb{R}^{6S+3}$$

In the first layer, we define the positional encodings corresponding to the key as follows, while assuming that all other positional encodings remain fixed and set to zero.

$$\mathbf{p}_i^{(1),K} = \left\{ p_i \cdot \begin{bmatrix} 0 & 1 & \mathbf{0}_{1\times(1+6S)} \end{bmatrix}^T, \quad \text{if } i \in \{0,1,2,\cdots,n\}, \right.$$

**Layer 1.** We paramatrize the query and key matrice as follows:

$$\mathbf{W}_K^{(1)} = \begin{bmatrix} 0_{1\times 1} & \mathbf{1}_{1\times 2} & \mathbf{0}_{1\times 6S} \\ \mathbf{0}_{(2+6S)\times 1} & \mathbf{0}_{(2+6S)\times 2} & \mathbf{0}_{(2+6S)\times(2+6S)} \end{bmatrix}$$

$$\mathbf{W}_Q^{(1)} = \begin{bmatrix} \mathbf{0}_{1\times 3} & \mathbf{1}_{1\times S} & \mathbf{0}_{1\times 5S} \\ \mathbf{0}_{(2+6S)\times 3} & \mathbf{0}_{(2+6S)\times S} & \mathbf{0}_{(2+6S)\times 5S} \end{bmatrix}$$

This then leads to:

$$\left\langle \mathbf{W}_K^{(1)}\big(\texttt{Emb}(x_{n-i}) + \mathbf{p}_i^{(1),K}\big), \mathbf{W}_Q^{(1)}\texttt{Emb}(x_n) \right\rangle = p_i$$

Hence, on passing it through the softmax function:

$$\texttt{att}_{n,n-i}^{(1)} = \frac{\exp(p_i)}{\sum_{i=0}^n \exp(p_i)}$$

Choosing the value matrix as,

$$\mathbf{W}_V^{(1)} = \begin{bmatrix} I_{3\times 3} & \mathbf{0}_{3\times S} & \mathbf{0}_{3\times 4S} \\ \mathbf{0}_{3S\times 3} & \mathbf{0}_{S\times S} & \mathbf{0}_{3S\times 4S} \\ \mathbf{0}_{S\times 3} & I_{S\times S} & \mathbf{0}_{S\times 4S} \\ \mathbf{0}_{2S\times 3} & \mathbf{0}_{S\times S} & \mathbf{0}_{2S\times 4S} \end{bmatrix}$$

Hence, we can obtain,

$$\widetilde{\mathbf{x}}_n^{(1)} = \begin{bmatrix} \mathbf{0}_{1\times 3} & | & e_{x_n}^S & | & \mathbf{0}_{1\times 2S} & | & \mathbf{v}_n & | & \mathbf{0}_{1\times 2S} \end{bmatrix}^T,$$

$$\text{where, } \mathbf{v}_n = \sum_{i=0}^n \texttt{att}_{n,n-i}\, e_{x_{n-i}}^S$$

**MLP - Layer 1.** The first layer of the MLP is defined as:

$$\mathbf{W}_{\texttt{mlp}}^{(1)} = \begin{bmatrix} \mathbf{0}_{(3+S)\times(3+3S)} & \mathbf{0}_{(3+S)\times(S)} & \mathbf{0}_{(3+S)\times(2S)} \\ \mathbf{0}_{S\times(3+3S)} & \mathbf{0}_{S\times S} & \mathbf{0}_{S\times 2S} \\ \mathbf{0}_{S\times(3+3S)} & \mathbf{0}_{S\times S} & \mathbf{0}_{S\times 2S} \\ \mathbf{0}_{3S\times(3+3S)} & \mathbf{0}_{3S\times S} & \mathbf{0}_{3S\times 2S} \end{bmatrix}$$

$$\mathbf{b}_{\texttt{mlp}}^{(1)} = \begin{bmatrix} \mathbf{0}_{1\times(3+S)} & \mathbf{0}_{1\times S} & \mathbf{0}_{1\times S} & \mathbf{0}_{1\times 2S} \end{bmatrix}^T$$

Hence, after applying the skip connections, we obtain:

$$\widetilde{\mathbf{x}}_{n,\mathtt{mlp}}^{(1)} = \widetilde{\mathbf{x}}_n^{(1)} + \mathrm{LN}\left(\mathrm{ReLU}\left(\mathbf{W}^{(1)}\widetilde{\mathbf{x}}_n^{(1)} + \mathbf{b}_{\mathtt{mlp}}^{(1)}\right)\right)$$

$$= \left[\ \mathbf{0}_{1\times 3}\ \middle|\ e_{x_n}^S\ \middle|\ \mathbf{0}_{1\times S}\ \middle|\ \mathbf{0}_{1\times S}\ \middle|\ \mathbf{v}_n\ \middle|\ \mathbf{0}_{1\times 2S}\ \right]^T$$

**MLP - Layer 2.**   The second layer of the MLP is defined as:

$$\mathbf{W}_{\mathtt{mlp}}^{(2)} = \begin{bmatrix} \mathbf{0}_{3\times 3} & \mathbf{0}_{3\times S} & \mathbf{0}_{3\times S} & \mathbf{0}_{3\times S} & \mathbf{0}_{3\times S} & \mathbf{0}_{3\times S} & \mathbf{0}_{3\times S} \\ \mathbf{0}_{S\times 3} & \mathbf{0}_{S\times S} & \mathbf{0}_{S\times S} & \mathbf{0}_{S\times S} & \mathbf{0}_{S\times S} & \mathbf{0}_{S\times S} & \mathbf{0}_{S\times S} \\ \mathbf{0}_{S\times 3} & \mathbf{0}_{S\times S} & \mathbf{0}_{S\times S} & \mathbf{0}_{S\times S} & \mathbf{0}_{S\times S} & \mathbf{0}_{S\times S} & \mathbf{0}_{S\times S} \\ \mathbf{0}_{S\times 3} & I_{S\times S} & \mathbf{0}_{S\times S} & \mathbf{0}_{S\times S} & \mathbf{0}_{S\times S} & \mathbf{0}_{S\times S} & \mathbf{0}_{S\times S} \\ \mathbf{0}_{S\times 3} & \mathbf{0}_{S\times S} & \mathbf{0}_{S\times S} & \mathbf{0}_{S\times S} & \mathbf{0}_{S\times S} & \mathbf{0}_{S\times S} & \mathbf{0}_{S\times S} \\ \mathbf{0}_{S\times 3} & \mathbf{0}_{S\times S} & \mathbf{0}_{S\times S} & \mathbf{0}_{S\times S} & \mathbf{0}_{S\times S} & \mathbf{0}_{S\times S} & \mathbf{0}_{S\times S} \\ \mathbf{0}_{S\times 3} & \mathbf{0}_{S\times S} & \mathbf{0}_{S\times S} & \mathbf{0}_{S\times S} & \mathbf{0}_{S\times S} & \mathbf{0}_{S\times S} & \mathbf{0}_{S\times S} \end{bmatrix}$$

$$\mathbf{b}_{\mathtt{mlp}}^{(2)} = \mathbf{0}_{(3+6S)\times 1}$$

Hence, on computing the output, we obtain:

$$\widetilde{\mathbf{x}}_{n,\mathtt{mlp}}^{(2)} = \widetilde{\mathbf{x}}_n^{(1)} + \mathrm{LN}\left(\mathrm{ReLU}\left(\mathbf{W}_{\mathtt{mlp}}^{(2)}\widetilde{\mathbf{x}}_{n,\mathtt{mlp}}^{(1)} + \mathbf{b}_{\mathtt{mlp}}^{(2)}\right)\right)$$

$$= \left[\ \mathbf{0}_{1\times 3}\ \middle|\ e_{x_n}^S\ \middle|\ \mathbf{0}_{1\times S}\ \middle|\ \mathbf{u}_n\ \middle|\ \mathbf{v}_n\ \middle|\ \mathbf{0}_{1\times 2S}\ \right]^T$$

Where, $\mathbf{u}_n = e_{x_n}^S$

Note that, in this setting, the optimality condition is well defined due to the first order Markov property. In particular, $\mathbf{u}_n$ can be recovered directly via the skip connections, as it depends only on the current token.

**MLP - Layer 3.**   The third layer of the MLP is defined as:

$$\mathbf{W}_{\mathtt{mlp}}^{(3)} = \begin{bmatrix} \mathbf{0}_{3\times 3} & \mathbf{0}_{S\times S} & \mathbf{0}_{3\times 4S} \\ \mathbf{0}_{3S\times 3} & \mathbf{0}_{S\times S} & \mathbf{0}_{3S\times 4S} \\ \mathbf{0}_{S\times 3} & I_{S\times S} & \mathbf{0}_{S\times 4S} \\ \mathbf{0}_{2S\times 3} & \mathbf{0}_{S\times S} & \mathbf{0}_{2S\times 4S} \end{bmatrix}$$

$$\mathbf{b}_{\mathtt{mlp}}^{(3)} = \mathbf{0}_{(3+6S)\times 1}$$

Passing the input to the layer through the non-linearities along with the skip connections, we get:

$$\widetilde{\mathbf{x}}_{n,\mathtt{mlp}}^{(3)} = \widetilde{\mathbf{x}}_{n,\mathtt{mlp}}^{(2)} + \mathrm{LN}\left(\mathrm{ReLU}\left(\mathbf{W}_{\mathtt{mlp}}^{(2)}\widetilde{\mathbf{x}}_{n,\mathtt{mlp}}^{(1)} + \mathbf{b}_{\mathtt{mlp}}^{(2)}\right)\right)$$

$$= \left[\ 0\ \middle|\ \mathbf{0}_{1\times 2}\ \middle|\ e_{x_n}^S\ \middle|\ 0_{1\times S}\ \middle|\ \frac{\mathbf{u}_n}{\|\mathbf{u}_n\|_2}\ \middle|\ \mathbf{v}_n\ \middle|\ \frac{\mathbf{v}_n}{\|\mathbf{v}_n\|_2}\ \middle|\ \mathbf{0}_{1\times S}\ \right]^T$$

**Layer 2.**   In this layer, all the relative position encodings are set as $\mathbf{0}$ and instead,

$$\mathbf{W}_Q^{(2)} = a_2 \cdot \begin{bmatrix} 1 & \mathbf{0} & \mathbf{0} & \mathbf{0} \\ 0 & \mathbf{0}_{S\times(2+3S)} & I_{S\times S} & \mathbf{0} \\ 0 & \mathbf{0} & \mathbf{0} & \mathbf{0} \end{bmatrix}$$

$$\mathbf{W}_K^{(2)} = \begin{bmatrix} 1 & \mathbf{0} & \mathbf{0} & \mathbf{0} \\ 0 & \mathbf{0}_{S\times(2+4S)} & I_{S\times S} & \mathbf{0} \\ 0 & \mathbf{0} & \mathbf{0} & \mathbf{0} \end{bmatrix}$$

With these choices, and adding a normalization after the output of the key linear layer

$$\left\langle \mathbf{W}_K^{(2)} \tilde{\mathbf{x}}_{n,\mathtt{mlp}}^{(3)}, \mathbf{W}_Q^{(2)} \tilde{\mathbf{x}}_{n,\mathtt{mlp}}^{(3)} \right\rangle = a_2 \cdot \langle \mathbf{v}_i, \mathbf{u}_n \rangle$$

Taking the softmax, we obtain,

$$\mathrm{att}_{n,i}^{(2)} = \frac{\exp\left(a_2 \cdot \langle \mathbf{v}_i, \mathbf{u}_n \rangle\right)}{\sum_{j=0}^{n} \exp(a_2 \cdot \langle \mathbf{v}_j, \mathbf{u}_n \rangle)}$$

Finally the logit is computed as follows (selecting the optimal value and projection matrix):

$$\mathrm{logit}_n = \sum_{i=0}^{n} \mathrm{att}_{n,i}^{(2)} \cdot e_{x_i}$$

Motivated by this parameterization, we construct a simplified model that abstracts away the complex transformations occurring within the intermediate layers of the network. During stage one of training, we remove layer normalization; accordingly, we omit layer norms from the simplified model as well. These can be reintroduced later during stage two of training. However, in the case of a first order Markov chain, layer normalization does not affect the model's behavior, since both $\mathbf{u}_n$ and $\mathbf{v}_n$ converge to one-hot embedding vectors. With these considerations, the model can be concretely reduced as follows. Let $Z_n \in \mathbb{R}^{n+1}$, where $Z_i = \langle e_{x_n}, v_i \rangle$. Morever, let:

$$X_n = \begin{bmatrix} e_{x_0} & e_{x_1} & e_{x_2} & \cdots & e_{x_n} \end{bmatrix}$$

Hence,

$$\mathrm{logit}_{s'_s,n} = e_{s'}^T \left[ X_n \, \mathrm{softmax}\left(a_2 Z_n\right) \right]$$

Note that the logit computed above corresponds to the case where $x_n = s$. Using a slightly different notation to account for causal masking, we obtain the following:

$$A^{(1)} = \begin{bmatrix} p_0 & -\infty & -\infty & -\infty & -\infty \\ p_1 & p_0 & -\infty & -\infty & -\infty \\ \vdots & \vdots & \vdots & \vdots & \vdots \\ p_n & p_{n-1} & p_{n-1} & \cdots & p_0 \end{bmatrix}$$

$$A^{(2)} = a_2 \cdot I$$

Hence, $\mathrm{logit}_{s'_s,n} = e_{s'}^T \left[ X_n \, \mathrm{softmax}\left(a_2 Z_n\right) \right]$

$$= e_{s'}^T \left[ X_n \underbrace{\mathrm{softmax}\left(\mathrm{softmax}\left(A^{(1)}\right) X_n^T A^{(2)} e_{x_n}\right)}_{\mathcal{A}_\theta^{(2)}(X_n;1)} \right]$$

We now use this simplified model for the gradient analysis.

## C.2   Stage 1 Analysis

We use the simplified model described in the previous section. We begin by quantifying the loss as follows:

$$L(\theta) = -\frac{1}{S} \mathbb{E}_{\pi \sim \mathbb{P}(\pi), X_n} \left[ \sum_{s', s \in [S]} \pi \left( s_{n+1} = s' | s \right) \log \left( \text{logit}_{s'_s, n} + \epsilon \right) \right]$$

Let $\mathbf{1}_n = [1, 1, \cdots, 1]^T \in \mathbb{R}^n$. Hence, we can then define:

$$\text{logit}_{s'_s, n} = e_{s'}^T \left[ X_n \, \text{softmax} \left( a_{2,0} Z_n \right) \right]$$

Assuming, that $a_{2,0} = a_{2,0}$ is very small, we can use a first order taylor approximation. We also assume that $x_n = s$:

$$\text{logit}_{s'_s, n} \approx e_{s'}^T \cdot \left[ X_n \cdot \left\{ \frac{\mathbf{1_n}}{n} + a_{2,0} \cdot \left( \frac{I_n}{n} - \frac{1}{n^2} \cdot \mathbf{1}_n \mathbf{1}_n^T \right) \cdot Z_n \right\} \right]$$

$$\approx \frac{e_{s'}^T X_n \mathbf{1}_n}{n} + a_{2,0} \cdot \left[ \frac{e_{s'}^T X_n Z_n}{n} - \frac{e_{s'}^T X_n \mathbf{1}_n}{n} \cdot \frac{\mathbf{1_n}^T Z_n}{n} \right]$$

$$\approx \frac{e_{s'}^T X_n \mathbf{1}_n}{n} + \frac{a_{2,0}}{n} \cdot \left[ e_{s'}^T X_n Z_n - \frac{e_{s'}^T X_n \mathbf{1}_n}{n} \cdot \mathbf{1_n}^T Z_n \right]$$

Note that, on computing the quantities separately:

$$\frac{e_{s'}^T X_n \mathbf{1}_n}{n} = \hat{\mu}_{\pi, n}(s')$$

$$\mathbf{1_n}^T Z_n = \sum_{i=0}^{n} \langle e_{x_n}, v_i \rangle$$

$$= \sum_{i=0}^{n} \langle e_{x_n}, \sum_{j=0}^{i} \frac{\exp(p_j)}{\sum_{j=0}^{i} \exp(p_j)} e_{x_{i-j}} \rangle$$

$$= \sum_{i=0}^{n} \sum_{j=0}^{i} \frac{\exp(p_j)}{\sum_{j=0}^{i} \exp(p_j)} \langle e_{x_n}, e_{x_{i-j}} \rangle$$

$$e_{s'}^T X_n Z_n = \sum_{i=0}^{n} \sum_{j=0}^{i} \frac{\exp(p_j)}{\sum_{j=0}^{i} \exp(p_j)} \langle e_{x_n}, e_{x_{i-j}} \rangle \langle e_{x_i}, e_s \rangle$$

Hence, on substituting the above quantities:

$$\text{logit}_{s'_s, n} \approx \hat{\mu}_{\pi, n}(s') + \frac{a_{2,0}}{n} \cdot \left[ \sum_{i=0}^{n} \sum_{j=0}^{i} \frac{\exp(p_j)}{\sum_{j=0}^{i} \exp(p_j)} \left\{ \langle e_{x_n}, e_{x_{i-j}} \rangle \langle e_{x_i}, e_{s'} \rangle - \langle e_{x_n}, e_{x_{i-j}} \rangle \hat{\mu}_{\pi, n}(s') \right\} \right]$$

$$\approx \hat{\mu}_{\pi, n}(s) + \frac{a_{2,0}}{n} \cdot \left[ \sum_{i=0}^{n} \sum_{j=0}^{i} \frac{\exp(p_j)}{\sum_{j=0}^{i} \exp(p_j)} \left\{ \mathbb{1}_{x_{i-j}=s} \cdot \mathbb{1}_{x_i=s} - \mathbb{1}_{x_{i-j}=s} \cdot \hat{\mu}_{\pi, n}(s') \right\} \right]$$

Hence, on computing the derivatives of $p_m$, keeping all the other variables constant, we obtain

$$\frac{dL(\theta)}{dp_m} = -\frac{1}{S} \mathbb{E}_{\pi \sim \mathbb{P}(\pi), X_n} \left[ \sum_{s', s} \pi \left( s_{n+1} = s' | s \right) \frac{d \log \left( \text{logit}_{s'_s, n} \right)}{dp_m} \right]$$

Note that for sufficiently long sequences, $\hat{\mu}_{\pi,n}(s') \approx \mu_\pi(s')$. Thus:

$$\frac{\mathrm{d}\log\left(\mathrm{logit}_{s'_s,n}\right)}{\mathrm{d}p_m} = \frac{a_{2,0}}{S \cdot n \cdot \mu_\pi(s')} \cdot \sum_{i=m}^{n} \frac{\exp(p_m)}{\sum_{j=0}^{i}\exp(p_j)} \cdot \left[ q_{i,m} - \sum_{j=0}^{i} \frac{\exp(p_j \cdot a_1)}{\sum_{j=0}^{i}\exp(p_j \cdot a_1)} q_{i,j} \right]$$

Where, $q_{i,j} = \mathbb{1}_{x_{i-j}=s} \cdot \mathbb{1}_{x_i=s'} - \mathbb{1}_{x_{i-j}=s} \cdot \mu_\pi(s'$

Using the tower property of expectation,

$$\frac{\mathrm{d}L(\theta)}{\mathrm{d}p_m} = -\frac{a_{2,0}}{S \cdot kn} \cdot \mathbb{E}_{\pi \sim \mathbb{P}(\pi)}\left[ \sum_{i \geqslant m}^{n} \sum_{s',s} \frac{\pi(s'|s)}{\mu_\pi(s')} \cdot \frac{\exp(p_m)}{\sum_{j=0}^{i}\exp(p_j)} \right.$$

$$\left. \cdot \left( \mathbb{P}(x_i = s', x_{i-m} = s) - \mathbb{P}(x_{i-m} = s)\mu_\pi(s') \right) \right]$$

$$+ \frac{a_{2,0}}{S \cdot kn} \cdot \mathbb{E}_{\pi \sim \mathbb{P}(\pi)}\left[ \sum_{i \geqslant m}^{n} \sum_{s',s} \sum_{j=0}^{i} \frac{\pi(s'|s)}{\mu_\pi(s')} \cdot \frac{\exp(p_j)\exp(p_m)}{\left(\sum_{j=0}^{i}\exp(p_j)\right)^2} \right.$$

$$\left. \cdot \left( \mathbb{P}(x_i = s', x_{i-j} = s) - \mathbb{P}(x_{i-j} = s)\mu_\pi(s') \right) \right]$$

We define a new quantity,

$$g_{i,j}(\pi) = \sum_{s',s} \frac{\pi(s'|s)}{\mu_\pi(s')}\left( \mathbb{P}(x_i = s', x_{i-j} = s) - \mathbb{P}(x_{i-j} = s)\mu_\pi(s') \right)$$

$$= \sum_{s',s} \frac{\pi(s'|s)}{\mu_\pi(s')}\left( \mathbb{P}(x_i = s', x_{i-j} = s) - \mu_\pi(s)\mu_\pi(s') \right) \quad \text{(from stationarity)}$$

$$= \sum_{s',s} \frac{\pi(s'|s)}{\mu_\pi(s')}\mathbb{P}(x_i = s', x_{i-j} = s) - \sum_{s',s} \frac{\pi(s'|s)}{\mu_\pi(s')}\mu_\pi(s)\mu_\pi(s')$$

$$= \sum_{s',s} \frac{\pi(s'|s)}{\mu_\pi(s')}\mathbb{P}(x_i = s', x_{i-j} = s) - 1 \quad \text{(by marginalization)}$$

$$g_{i,j} = \mathbb{E}_{\pi \sim \mathbb{P}(\pi)}\left[ g_{i,j}(\pi) \right]$$

Hence, on substituting this back, we obtain,

$$\frac{\mathrm{d}L(\theta)}{\mathrm{d}p_m} = -\frac{a_{2,0}}{S \cdot kn} \cdot \mathbb{E}_{\pi \sim \mathbb{P}(\pi)}\left[ \sum_{i \geqslant m}^{n} \frac{\exp(p_m)}{\sum_{j=0}^{i}\exp(p_j)}g_{i,m}(\pi) - \sum_{i \geqslant m}^{n} \sum_{j=0}^{i} \frac{\exp(p_j)\exp(p_m)}{\left(\sum_{j=0}^{i}\exp(p_j)\right)^2}g_{i,j}(\pi) \right]$$

Therefore, rather than considering the entire expression, we focus on a specific row (i.e., for a given $i$), and obtain:

$$\frac{\mathrm{d}L(\theta)}{\mathrm{d}p_m}\bigg|_i = -\frac{a_{2,0}}{S \cdot kn} \cdot \mathbb{E}_{\pi \sim \mathbb{P}(\pi)}\left[ \left( \frac{\exp(p_m)}{\sum_{j=0}^{i}\exp(p_j)}g_{i,m}(\pi) - \sum_{j=0}^{i} \frac{\exp(p_j)\exp(p_m)}{\left(\sum_{j=0}^{i}\exp(p_j)\right)^2}g_{i,j}(\pi) \right) \right]$$

To obtain the optimal attention map for a first-order Markov chain, the goal is to demonstrate that, for every row, the gradient with respect to $p_1$ is greater than that of any other token and is also positive. Due to the properties of the softmax function, if this condition holds over the course of training, then the model will asymptotically attend to the previous token, thereby converging to the optimal attention map. Hence, the requirement is essentially, for every row:

$$\frac{\mathrm{d}L(\theta)}{\mathrm{d}p_1}\bigg|_i > \frac{\mathrm{d}L(\theta)}{\mathrm{d}p_k}\bigg|_i \quad \forall\, k \in \{0, 2, \ldots, n\}$$
$$\frac{\mathrm{d}L(\theta)}{\mathrm{d}p_1}\bigg|_i > 0$$

In this section, we will show that given the prior assumptions, this is indeed true. Towards this, we denote $d_{g_m,i} = \frac{\mathrm{d}L(\theta)}{\mathrm{d}p_m}\big|_i$. With this notation, we can express:

$$d_{g_m,i} = -\frac{a_{2,0}}{S \cdot kn} \cdot \left[ \frac{\exp(p_m)}{\sum_{j=0}^{i} \exp(p_j)} \cdot \left( g_{i,m} - \sum_{j=0}^{i} \frac{\exp(p_j)}{\sum_{j=0}^{i} \exp(p_j)} \cdot g_{i,j} \right) \right]$$

Now, suppose we initialize all positional encoding scalars to zero as defined in the training algorithm. Then, at initialization time $t = 0$, we have:

$$d_{g_m,i,t=0} = -\frac{a_{2,0}}{S \cdot kn} \cdot \frac{1}{i+1} \left( g_{i,m} - \sum_{j=1}^{i} \frac{1}{i+1} \cdot g_{i,j} \right), \quad \forall\, i \geqslant 0$$

Due to Lemma 7, we observe that at initialization time $t = 0$ (when all positional encoding scalars are initialized to zero), the gradient of the attention logits is maximized in the direction corresponding to position 1. This implies that the model is initially biased to attend more strongly to the immediately preceding token, aligning with the structure of a first-order Markov process. Concretely, we observe that:

$$-d_{g_1,i,0} > -d_{g_k,i,0} \quad \forall\, k \in \{0, 2, \ldots, n\}, \quad \forall\, i \in \{k, \ldots, n\}$$
$$\text{and} \quad -d_{g_1,i,0} \geqslant 0 \,\,\forall\, i \in \{k, \ldots, n\}$$

Hence, under the given initialization and model assumptions, performing gradient descent yields the update rule:

$$\mathbf{p}_{t+1} = \mathbf{p}_t - \eta_1 \nabla_{\mathbf{p}_t} L(\theta),$$
$$\text{where} \quad \mathbf{p}_t = \begin{bmatrix} p_{0,t} & p_{1,t} & \cdots & p_{n,t} \end{bmatrix}^\top \in \mathbb{R}^{n+1}.$$

Here, $p_{i,t}$ denotes the $i$-th positional encoding scalar at iteration $t$. Therefore, after the first update step (i.e., at $t = 1$), we have:

$$p_{1,t=1} > p_{k,t=1} \quad \forall k \in \{0, 2, \ldots, n\},$$

reflecting that the gradient is largest in the direction corresponding to position 1.

We need to show that this property holds for all iterates. By Lemma 3, we can show that for each iterate of the preconditioned descent, the following holds:

$$p_{1,t} > p_{k,t} \quad \forall\, k \in \{0, 2, \ldots, n\}$$

Furthermore, by Lemma 3 and Lemma 4, we observe that $p_1$ tends to infinity as it increases monotonically over time. In contrast, the other positional encodings do not diverge, as they do not exhibit similar growth. Due to the nature of the softmax function, the rapid increase of $p_1$ suppresses the relative influence of the other components, effectively preventing their growth. In addition, Lemma 4 provides an estimate of the number of iterations required for the iterates to converge to the optimal attention configuration. Therefore, assuming a sufficiently long training period during the first stage, the model successfully recovers the ideal attention map corresponding to a second-order Markov chain:

$$\mathbf{v}_i \approx e_{x_{i-1}}$$

Hence, in the second stage of training, the inclusion of layer normalization does not significantly affect the outcome, provided the Markov chain is sufficiently long and the training duration is adequate, and hence, we will assume that $\mathbf{v}_i/\|\mathbf{v}_i\| \approx e_{x_{i-1}}$, which is the optimal attention map. We note that the gradient analysis for the first stage of training closely mirrors that of the first stage in Nichani et al. [20], and is directly inspired by their approach.

### C.3 Stage 2 Analysis

We observe that, at this stage—assuming the attention map learned in the first stage is optimal by having the appropriate sequence length and training time—our simplified model aligns with the one presented in Nichani et al. [20]. This alignment arises because, after the first stage of training, both models converge to the same optimal attention map in the first layer and subsequently train a single scalar parameter in the second attention layer. Therefore, we can directly invoke the theorems from Nichani et al. [20] to analyze the convergence behavior of our model during the second stage of training. Accordingly, we state the relevant theorems below without proofs. Using the same set of assumptions outlined in Appendix C, we restate the key lemmas and results from Nichani et al. [20] as follows.

**Lemma 1** (Lemma D.8 in Nichani et al. [20]). *Let* $\theta = \left( \{p_i^{(T_1)}\}_{i=0}^n, a_2^{(t)} \right)$, *where* $\{p_i^{(T_1)}\}_{i=0}^n$ *denotes the output after the first stage of training. If $a_2$ satisfies* $\exp(a_2) \leqslant \exp(a_2^*) := C_{\gamma,S}\, n^{1/12} \log^{-1/6} n$, *and $a_{2,0} > 0$ (at initialization), then*

$$1 \geqslant -\frac{\mathrm{d}L(\theta)}{\mathrm{d}a_2} \geqslant \frac{1}{4}\gamma^8 S^{-6} e^{-2a_2} > 0.$$

This clearly indicates that the negative gradient remains positive, causing $a_2$ to increase indefinitely under gradient descent. As a result, the attention mechanism progressively sharpens, effectively converging to a hard attention regime. Over time, this leads to convergence to the optimal 1-gram estimator as the number of training iterations grows. For a thorough discussion of the time taken to converge, we refer readers to Nichani et al. [20] (see Lemma D.10). Furthermore, given the equivalence between the models, once training is complete—assuming a sufficiently long sequence and adequate training duration—they are essentially identical. Therefore, we defer the proof of convergence of our model's loss to the optimal loss (Theorem 4) to the corresponding argument in Nichani et al. [20] (see D.7. Proof of Theorem 4.4). Finally, due to this model equivalence, and after the completion of both training stages, we again refer to Nichani et al. [20] (see H.4. Proof of Theorem 4.5) for the proof of Theorem 8.

## D  Two Heads in Layer One, One in Layer Two Perturbative Analysis

Understanding the convergence behavior of a Transformer on Markov chains of arbitrary order in the prior setting (Appendix C) presents significant challenges. A central difficulty lies in the dependence of accurately estimating $\mathbf{u}_n$ on the precise recovery of $\mathbf{v}_n$. In earlier constructions, this estimation was enabled by complex non-linearities such as layer normalization, which makes direct gradient-based analysis intractable. To facilitate analysis, we focus on a simplified architecture: a two-layer model where the first layer includes two attention heads, as described in Section 3. Due to the intractability of analyzing the full model, we adopt a perturbative approach in which only a subset of parameters is trained, while the remainder of the network is fixed at optimal values. This aligns with our goal of exploring the behavior of a low-parameterized version of the model. Within this setup, we consider arbitrary-order Markov chains under the assumption that the first head is optimally initialized to recover $\mathbf{u}_n$. Consistent with the simplification in Appendix C, we limit learning to the key positional vectors of the second head in the first layer, parameterized by a single scalar. All other network components remain fixed. The second layer of attention is not assumed to be learnable; instead, we begin with a softmax function at very low temperature, held constant during the initial training phase. This temperature is later increased to infinity. Even under these constraints, analyzing convergence of the positional encodings to their optimal values remains analytically intractable without strong assumptions about initialization, training procedures, and the data distribution. Nonetheless, we regard this work as a first step toward understanding convergence in higher-order Markov models within more realistic Transformer architectures.

**Training Algorithm**  We introduce a training algorithm that updates only the positional scalars $p_i$ for the key vectors of the second attention head in the first layer, for all $i \in \{0, \ldots, n\}$. Here, $D_{(t-1)}^{-1}$ denotes a data-dependent diagonal preconditioner.

---

**Algorithm 2.** (Training Algorithm).
**Input:** learning rates $\eta_1$, steps $T_1$,
Initialize $p_i^{(0)} = 0$ for all $i \in \{1, \ldots, n\}$ and $p_0^{(0)} = -\infty$;
Freeze all other parameters of the network
Set the softmax temperature of the second attention head to a very low value.
**Stage 1: Train  $p_i$  without norms added to the MLP**
**for** $t = 1, \ldots, T_1$ **do**
$\quad \mathbf{p}^{(t)} \leftarrow \mathbf{p}^{(t-1)} - \eta_1 \cdot \left(D_{(t-1)}\right)^{-1} \nabla_{\mathbf{p}} \mathcal{L}(\theta^{(t-1)})$
$\quad \theta^{(t)} \leftarrow \left(\{p_i^{(t)}\}_{i=0}^n\right)$
**end for**
$\hat{\theta} \leftarrow \theta^{(T_1)}$
**Output:** $\hat{\theta}$
Add norms back to the MLP
Set the Set the softmax temperature of the second attention head to $\infty$

---

**Data Assumptions**  We introduce the following data assumptions on the prior over Markov transition kernels, first for second-order Markov chains and subsequently for higher-order Markov chains.

---

**Assumption 2.** (Assumption on $\mathbb{P}_\pi$ for Second-Order Markov Chains): We consider a prior distribution $\mathbb{P}_\pi$ over transition kernels of **second-order** Markov chains defined over a binary state space $\mathcal{S} = \{0, 1\}$. Each transition kernel $\pi \sim \mathbb{P}_\pi$ specifies conditional probabilities $\pi(s' \mid s_1, s_2)$, and satisfies the following assumptions along with stationarity:

1. **Time Reversibility:** There exists a stationary distribution $\mu(s_1, s_2)$ such that $\mu(s_1, s_2)\, \pi(s' \mid s_1, s_2) = \mu(s_2, s')\, \pi(s_1 \mid s_2, s')$.

2. **Transition Preference:** $\pi(0 \mid 0, 0) > \pi(0 \mid 1, 0)$ and $\pi(1 \mid 1, 1) > \pi(1 \mid 0, 1)$.

3. **Second-Hop Likelihood:** $\sum_{s \in \{0,1\}} \pi^2(s \mid s, s) \geqslant 1$.

---

**Assumption 3.** (Assumption on $\mathbb{P}_\pi$ for $k^{\text{th}}$-Order Reversible Markov Chains):We consider a prior distribution $\mathbb{P}_\pi$ over transition kernels for $k$th-order Markov chains on a finite state space $S$, with lifted space $\mathcal{A} = S^k$, such that each transition kernel $\pi \sim \mathbb{P}_\pi$ satisfies:

1. **Irreducibility and Aperiodicity:** The chain $(X_t)$ is irreducible and aperiodic;

2. **Reversibility:** The chain is reversible with respect to a stationary distribution $\pi$;

3. **Lifted Representation:** The chain admits a lifted first-order representation with transition kernel $P(a \to b) = \Pr((X_t, \ldots, X_{t-k+1}) = b \mid (X_{t-1}, \ldots, X_{t-k}) = a)$;

4. **Spectral Assumptions:** Under detailed balance with stationary distribution $\pi$, the transition matrix $P$ is self-adjoint on $\ell^2(\mathcal{A}, \pi)$, with eigenvalues $1 = \lambda_1 > \lambda_2 \geq \cdots \geq \lambda_N > 0$ and orthonormal eigenfunctions $\{\phi_m\}$;

5. **Non-degenerate Projection:** The eigenfunction projections $\beta_m$, defined by

$$\beta_m = \sum_{a,b \in \mathcal{A}} \phi_m(a)\, \phi_m(b)\, \pi(b)\, f(a),$$

satisfy $\beta_m > 0$ for all $m$, where $f : \mathcal{A} \to \mathbb{R}$ is an observable function.

Under the assumptions outlined above, we state the following theorems.

**Theorem 9.** (Convergence of the Training Algorithm - Second Order): For all second-order Markov chains satisfying Assumption 2, there exist, a learning rate $\eta_1$, and a step count $T_1$, such that the output of the training algorithm (Alg. 2), $\hat{\theta} = \{\hat{p}_i\}_{i=0}^n$, satisfies the following as $T_1 \to \infty$:

$$\frac{\exp(p_1)}{\sum_{i=0}^n \exp(p_i)} = \frac{\exp(p_2)}{\sum_{i=0}^n \exp(p_i)} \approx \frac{1}{2}.$$

Moreover, as the softmax temperature is increased to infinity, the model reduces to a conditional 2-gram estimator.

**Theorem 10.** (Convergence of the Training Algorithm - $k^{\text{th}}$-order): For all $k^{\text{th}}$-order Markov chains satisfying Assumption 3, there exist a learning rate $\eta_1$, and a step count $T_1$, such that the output of the training algorithm (Alg. 2), $\hat{\theta} = \{\hat{p}_i\}_{i=0}^n$, satisfies the following as $T_1 \to \infty$:

$$\frac{\exp(p_1)}{\sum_{i=0}^n \exp(p_i)} = \frac{\exp(p_2)}{\sum_{i=0}^n \exp(p_i)} = \cdots \frac{\exp(p_k)}{\sum_{i=0}^n \exp(p_i)} \approx \frac{1}{k}.$$

Moreover, as the softmax temperature is increased to infinity, the model reduces to a conditional $k$-gram estimator.

We begin by describing the model simplification and subsequently present the convergence analysis.

### D.1 Construction

Note that we assume one of the heads ($\mathbf{u}_n$ to be optimal, we only need to focus on the other head. Since, we are interested in understanding the perturbative analysis of the second head with regards to the first, we define the layers as follows:

$$\mathbf{x}_n^{(1)} = \texttt{Emb}(x_n) = \begin{bmatrix} \mathbf{0}_{1 \times 3} & e_{x_n}^S & \mathbf{0}_{1 \times 5S} \end{bmatrix}^T \in \mathbb{R}^{6S+3}$$

In the first layer, for both the heads we use the following relative value embeddings

$$\mathbf{p}_i^{(\texttt{head:1}),V} = \mathbf{p}_i^{(\texttt{head:2}),V} = \begin{cases} \mathbf{0} \cdot \begin{bmatrix} 1 & \mathbf{0} \end{bmatrix}^T & \text{for } i \leqslant k \\ \mathbf{0} & i > k. \end{cases}$$

**Layer 1, Head 1.** We assume that the first head is optimal and hence, have an architecture that outputs the optimal attention map. Specifically for a $k^{\text{th}}$-order markov chain, we get the following attention score:

$$\mathbf{u}_n = \frac{1}{k} \left( e_{x_n} + e_{x_{n-1}} + \ldots + e_{x_{n-k+1}} \right)$$

Hence, the output of the attention head is:

$$\widetilde{\mathbf{x}}_n^{(\texttt{head:1})} = \begin{bmatrix} 0 & | & \mathbf{0}_{1\times2} & | & \mathbf{0}_{1\times S} & | & \mathbf{u}_n & | & \mathbf{0}_{1\times2S} & | & \mathbf{0}_{1\times2S} \end{bmatrix}^T,$$

**Layer 1, Head 2.** We will use the same relative position embeddings, in particular,

$$\mathbf{p}_i^{(\texttt{head:2}),K} = \left\{ p_i \cdot \begin{bmatrix} 0 & 1 & \mathbf{0}_{1\times(1+6S)} \end{bmatrix}^T, \quad \text{if } i \in \{0, 1, 2, \cdots, n\}, \right.,$$

We paramatrize the query and key matrices such that:

$$\left\langle \mathbf{W}_K^{(\texttt{head:2})} \left( \texttt{Emb}(x_{n-i}) + \mathbf{p}_i^{(\texttt{head:2}),K} \right), \mathbf{W}_Q^{(\texttt{head:2})} \texttt{Emb}(x_n) \right\rangle = p_i$$

Hence, on passing it through the softmax function:

$$\text{att}_{n,n-i}^{(1)} = \frac{\exp(p_i)}{\sum_{i=0}^{n} \exp(p_i)}$$

Hence, once we keep the value matrix fixed (not learnable):

$$\mathbf{W}_V^{(1)} = \begin{bmatrix} I_{3\times3} & \mathbf{0}_{3\times S} & \mathbf{0}_{3\times4S} \\ \mathbf{0}_{3S\times3} & \mathbf{0}_{S\times S} & \mathbf{0}_{3S\times4S} \\ \mathbf{0}_{S\times3} & I_{S\times S} & \mathbf{0}_{S\times4S} \\ \mathbf{0}_{2S\times3} & \mathbf{0}_{S\times S} & \mathbf{0}_{2S\times4S} \end{bmatrix}$$

The output of the second attention head is,

$$\widetilde{\mathbf{x}}_n^{(\texttt{head:2})} = \begin{bmatrix} \mathbf{0} & | & \mathbf{0}_{1\times2} & | & \mathbf{0}_{1\times S} & | & \mathbf{0}_{1\times2S} & | & \mathbf{v}_n & | & \mathbf{0}_{1\times2S} \end{bmatrix}^T,$$

$$\text{where, } \mathbf{v}_n = \sum_{i=1}^{\min\{n,k\}} \text{att}_{n,n-i} \, e_{x_{n-i}}^S$$

Therefore, with skip connections:

$$\begin{aligned} \widetilde{\mathbf{x}}_n^{(1)} &= \mathbf{x}_n^{(1)} + \widetilde{\mathbf{x}}_n^{(\texttt{head:1})} + \widetilde{\mathbf{x}}_n^{(\texttt{head:2})} \\ &= \begin{bmatrix} 0 & | & \mathbf{0}_{1\times2} & | & e_{x_n}^S & | & \mathbf{u}_n & | & \mathbf{0}_{1\times S} & | & \mathbf{v}_n & | & \mathbf{0}_{1\times2S} \end{bmatrix}^T \end{aligned}$$

**MLP.** The layers of the MLP are defined as follows:

The first layer for the MLP is :

$$\mathbf{W}_{\mathtt{mlp}}^{(1)} = \begin{bmatrix} \mathbf{0}_{(3+S)\times(3+3S)} & \mathbf{0}_{(3+S)\times(S)} & \mathbf{0}_{(3+S)\times(2S)} \\ \mathbf{0}_{S\times(3+3S)} & \mathbf{0}_{S\times S} & \mathbf{0}_{S\times 2S} \\ \mathbf{0}_{S\times(3+3S)} & I_{S\times S} & \mathbf{0}_{S\times 2S} \\ \mathbf{0}_{3S\times(3+3S)} & \mathbf{0}_{3S\times S} & \mathbf{0}_{3S\times 2S} \end{bmatrix}$$

$$\mathbf{b}_{\mathtt{mlp}}^{(1)} = \mathbf{0}_{(3+6S)\times 1}$$

The second layer for the MLP is :

$$\mathbf{W}_{\mathtt{mlp}}^{(2)} = \begin{bmatrix} \mathbf{0}_{(3+S)\times(3+3S)} & \mathbf{0}_{(3+S)\times(S)} & \mathbf{0}_{(3+S)\times(2S)} \\ \mathbf{0}_{S\times(3+3S)} & \mathbf{0}_{S\times S} & \mathbf{0}_{S\times 2S} \\ \mathbf{0}_{S\times(3+3S)} & \mathbf{0}_{S\times S} & \mathbf{0}_{S\times 2S} \\ \mathbf{0}_{S\times(3+3S)} & I_{S\times S} & \mathbf{0}_{S\times 2S} \\ \mathbf{0}_{2S\times(3+3S)} & \mathbf{0}_{2S\times S} & \mathbf{0}_{2S\times 2S} \end{bmatrix}$$

$$\mathbf{b}_{\mathtt{mlp}}^{(2)} = \mathbf{0}_{(3+6S)\times 1}$$

Hence, passing this through the normalization along with the skip connections, we obtain:

$$\widetilde{\mathbf{x}}_{n,\mathtt{mlp}}^{(2)} = \begin{bmatrix} 0 & \mathbf{0}_{1\times 2} & e_{x_n}^S & \mathbf{u}_n & \frac{\mathbf{u}_n}{\|\mathbf{u}_n\|_2} & \mathbf{v}_n & \frac{\mathbf{v}_n}{\|\mathbf{v}_n\|_2} & \mathbf{0}_{1\times S} \end{bmatrix}^T$$

**Layer 2.** In this layer, all the relative position encodings are set as $\mathbf{0}$ and instead, and paramaterize $W_Q^{(2)}, W_K^{(2)}$, such that

$$\left\langle (\mathbf{W}_K^{(2)}\widetilde{\mathbf{x}}_{i,\mathtt{mlp}}^{(2)}), (\mathbf{W}_Q^{(2)}\widetilde{\mathbf{x}}_{n,\mathtt{mlp}}^{(2)}) \right\rangle = \frac{2\langle \mathbf{v}_i, \mathbf{u}_n\rangle}{\|\mathbf{v}_i\|_2 \cdot \|\mathbf{u}_n\|_2}$$

$$= \left( 2 - \left\| \frac{\mathbf{v}_i}{\|\mathbf{v}_i\|_2} - \frac{\mathbf{u}_n}{\|\mathbf{u}_n\|_2} \right\|^2 \right)$$

Hence, the attention is computed as follows, using a softmax with temperature parameter $\kappa$:

$$\mathtt{att}_{n,i}^{(2)} = \frac{\exp\left( \frac{\kappa \cdot 2\langle \mathbf{v}_i, \mathbf{u}_n\rangle}{\|\mathbf{v}_i\|_2 \cdot \|\mathbf{u}_n\|_2} \right)}{\sum_{i=0}^{n} \exp(\kappa \cdot \frac{2\langle \mathbf{v}_i, \mathbf{u}_n\rangle}{\|\mathbf{v}_i\|_2 \cdot \|\mathbf{u}_n\|_2})}$$

Hence, we obtain,

$$\mathtt{logit}_n = \sum_{i=0}^{n} \mathtt{att}_{n,i}^{(2)} \cdot e_{x_i}$$

## D.2 Convergence Analysis

We can effectively reduce the model described in App, D.1 to a simplified form, analogous to that in App. C.1. Specifically, in this case, we obtain:

Let $Z_n \in \mathbb{R}^{n+1}$, where

$$Z_i = \left\langle \frac{1}{k}\left(e_{x_n} + e_{x_{n-1}} + \cdots + e_{x_{n-k}}\right), v_i \right\rangle.$$

Moreover, define:

$$X_n = \begin{bmatrix} e_{x_0} & e_{x_1} & e_{x_2} & \cdots & e_{x_n} \end{bmatrix}$$

Then,

$$\text{logit}_{s'_{s_k},n} = e_{s'}^T \left[ X_n \, \text{softmax}\left( \kappa Z_n \right) \right]$$

Using a slightly different notation, we can express the same as:

$$A^{(1)} = \begin{bmatrix} p_0 & -\infty & -\infty & -\infty & -\infty \\ p_1 & p_0 & -\infty & -\infty & -\infty \\ \vdots & \vdots & \vdots & \vdots & \vdots \\ p_n & p_{n-1} & p_{n-2} & \cdots & p_0 \end{bmatrix}$$

$$A^{(2)} = \kappa \cdot I$$

$$\text{logit}_{s'_{s_k},n} = e_{s'}^T \left[ X_n \, \text{softmax}\left( \kappa Z_n \right) \right]$$

$$= e_{s'}^T X_n \underbrace{\text{softmax}\left( \kappa \cdot \text{softmax}(A^{(1)}) X_n^T \cdot \frac{1}{k} \left( \sum_{i=0}^{k-1} e_{x_{n-i}} \right) \right)}_{\mathcal{A}_\theta^{(2)}(X_n;k)}$$

Clearly, the above expression closely mirrors the reduced model presented in Nichani et al. [20]. Since we have picked the softmax parameter $\kappa$ to be very small, we can apply a first-order Taylor approximation to simplify the expression. Let $\mathbf{1}_n = [1,1,\cdots,1] \in \mathbb{R}^{n+1}$. We also assume that $x_n = s_1, x_{n-1} = s_2, \ldots, x_{n-k} = s_k$.

$$\text{logit}_{s'_{s_k},n} \approx e_{s'}^T \cdot \left[ X_n \cdot \left( \frac{\mathbf{1}_n}{n} + \kappa \cdot \left( \frac{I_n}{n} - \frac{1}{n^2} \cdot \mathbf{1}_n \mathbf{1}_n^T \right) \cdot Z_n \right) \right]$$

$$\approx \frac{e_{s'}^T X_n \mathbf{1}_n}{n} + \kappa \cdot \left( \frac{e_{s'}^T X_n Z_n}{n} - \frac{e_{s'}^T X_n \mathbf{1}_n}{n} \cdot \frac{\mathbf{1}_n^T Z_n}{n} \right)$$

$$\approx \frac{e_{s'}^T X_n \mathbf{1}_n}{n} + \frac{\kappa}{n} \cdot \left( e_{s'}^T X_n Z_n - \frac{e_{s'}^T X_n \mathbf{1}_n}{n} \cdot \mathbf{1}_n^T Z_n \right)$$

Now, compute each term separately:

$$\frac{e_{s'}^T X_n \mathbf{1}_n}{n} = \hat{\mu}_{\pi,n}(s')$$

$$\mathbf{1}_n^T Z_n = \sum_{i=0}^{n} \left\langle \frac{1}{k} \sum_{l=0}^{k-1} e_{x_{n-l}}, v_i \right\rangle$$

$$= \sum_{i=0}^{n} \left\langle \frac{1}{k} \sum_{l=0}^{k-1} e_{x_{n-l}}, \sum_{j=0}^{i} \frac{\exp(p_j)}{\sum_{j=0}^{i} \exp(p_j)} e_{x_{i-j}} \right\rangle$$

$$= \sum_{i=0}^{n} \sum_{j=0}^{i} \frac{\exp(p_j)}{\sum_{j=0}^{i} \exp(p_j)} \left\langle \frac{1}{k} \sum_{l=0}^{k-1} e_{x_{n-l}}, e_{x_{i-j}} \right\rangle$$

$$e_{s'}^T X_n Z_n = \sum_{i=0}^{n} \sum_{j=0}^{i} \frac{\exp(p_j)}{\sum_{j=0}^{i} \exp(p_j)} \left\langle \frac{1}{k} \sum_{l=0}^{k-1} e_{x_{n-l}}, e_{x_{i-j}} \right\rangle \langle e_{x_i}, e_{s'} \rangle$$

Hence, substituting the expressions above,

$$\text{logit}_{s'_{s_k},n} \approx \hat{\mu}_{\pi,n}(s') + \frac{\kappa}{kn} \cdot \sum_{l=0}^{k-1} \left[ \sum_{i=0}^{n} \sum_{j=0}^{i} \frac{\exp(p_j)}{\sum_{j=0}^{i} \exp(p_j)} \cdot \right.$$

$$\left( \langle e_{x_{n-l}}, e_{x_{i-j}} \rangle \langle e_{x_i}, e_{s'} \rangle - \langle e_{x_{n-l}}, e_{x_{i-j}} \rangle \hat{\mu}_{\pi,n}(s') \right) \Bigg]$$

$$\approx \hat{\mu}_{\pi,n}(s') + \frac{\kappa}{kn} \cdot \sum_{l=1}^{k} \Bigg[ \sum_{i=0}^{n} \sum_{j=0}^{i} \frac{\exp(p_j)}{\sum_{j=0}^{i} \exp(p_j)} \cdot$$

$$\left( \mathbb{1}_{x_{i-j}=s_l} \cdot \mathbb{1}_{x_i=s'} - \mathbb{1}_{x_{i-j}=s_l} \cdot \hat{\mu}_{\pi,n}(s') \right) \Bigg]$$

Hence, by computing the derivatives with respect to $p_m$, while keeping all other variables constant, we obtain:

$$\frac{\mathrm{d}L(\theta)}{\mathrm{d}p_m} = -\frac{1}{|\mathcal{S}|^k} \mathbb{E}_{\pi \sim \mathbb{P}(\pi), \, s_n \sim \pi} \Bigg[ \sum_{s', s_1, \ldots, s_k} \pi(s_{n+1} = s' \mid s_1, \ldots, s_k) \cdot \frac{\mathrm{d}\log\left(\mathrm{logit}_{s'_s,n}\right)}{\mathrm{d}p_m} \Bigg]$$

Note that for sufficiently long sequences, $\hat{\mu}_{\pi,n}(s') \approx \mu_\pi(s')$. Thus:

$$\frac{\mathrm{d}\log\left(\mathrm{logit}_{s'_s,n}\right)}{\mathrm{d}p_m} = \frac{\kappa}{|\mathcal{S}|^k \cdot kn \cdot \mu_\pi(s')} \cdot \sum_{l=1}^{k} \sum_{i=m}^{n} \frac{\exp(p_m)}{\sum_{j=0}^{i} \exp(p_j)} \cdot \left[ q_{i,m,l} - \sum_{j=0}^{i} \frac{\exp(p_j)}{\sum_{j=0}^{i} \exp(p_j)} q_{i,j,l} \right]$$

where

$$q_{i,j,l} = \mathbb{1}_{x_{i-j}=s_l} \cdot \mathbb{1}_{x_i=s'} - \mathbb{1}_{x_{i-j}=s_l} \cdot \mu_\pi(s')$$

Using the tower property of expectation,

$$\frac{\mathrm{d}L(\theta)}{\mathrm{d}p_m} = -\frac{\kappa}{|\mathcal{S}|^k \cdot kn} \cdot \mathbb{E}_{\pi \sim \mathbb{P}(\pi)} \Bigg[ \sum_{i \geq m}^{n} \sum_{l=1}^{k} \sum_{s', s_1, \ldots, s_k} \frac{\pi(s' | s_1, \ldots, s_k)}{\mu_\pi(s')} \cdot \frac{\exp(p_m)}{\sum_{j=0}^{i} \exp(p_j)}$$

$$\cdot \left( \mathbb{P}(x_i = s', x_{i-m} = s_l) - \mathbb{P}(x_{i-m} = s_l)\mu_\pi(s') \right) \Bigg]$$

$$+ \frac{\kappa}{|\mathcal{S}|^k \cdot kn} \cdot \mathbb{E}_{\pi \sim \mathbb{P}(\pi)} \Bigg[ \sum_{i \geq m}^{n} \sum_{l=1}^{k} \sum_{s', s_1, \ldots, s_k} \sum_{j=0}^{i} \frac{\pi(s' | s_1, \ldots, s_k)}{\mu_\pi(s')} \cdot \frac{\exp(p_j)\exp(p_m)}{\left(\sum_{j=0}^{i} \exp(p_j)\right)^2}$$

$$\cdot \left( \mathbb{P}(x_i = s', x_{i-j} = s_l) - \mathbb{P}(x_{i-j} = s_l)\mu_\pi(s') \right) \Bigg] \tag{3}$$

Hence, instead of looking at this in totality, for a particular row (i.e. for a particular $i$, we obtain:)

$$\frac{\mathrm{d}L(\theta)}{\mathrm{d}p_m} \bigg|_i = -\frac{\kappa}{|\mathcal{S}|^k \cdot kn} \cdot \mathbb{E}_{\pi \sim \mathbb{P}(\pi)} \Bigg[ \sum_{l=1}^{k} \sum_{s', s_1, \ldots, s_k} \frac{\pi(s' | s_1, \ldots, s_k)}{\mu_\pi(s')} \cdot \frac{\exp(p_m)}{\sum_{j=0}^{i} \exp(p_j)}$$

$$\cdot \left( \mathbb{P}(x_i = s', x_{i-m} = s_l) - \mathbb{P}(x_{i-m} = s_l)\mu_\pi(s') \right) \Bigg]$$

$$+ \frac{\kappa}{|\mathcal{S}|^k \cdot kn} \cdot \mathbb{E}_{\pi \sim \mathbb{P}(\pi)} \Bigg[ \sum_{l=1}^{k} \sum_{s', s_1, \ldots, s_k} \sum_{j=0}^{i} \frac{\pi(s' | s_1, \ldots, s_k)}{\mu_\pi(s')} \cdot \frac{\exp(p_j)\exp(p_m)}{\left(\sum_{j=0}^{i} \exp(p_j)\right)^2}$$

$$\cdot \left( \mathbb{P}(x_i = s', x_{i-j} = s_l) - \mathbb{P}(x_{i-j} = s_l)\mu_\pi(s') \right) \Bigg] \tag{4}$$

We define a new quantity,

$$
\begin{aligned}
g_{i,j,l} &= \sum_{s',s_1,\cdots,s_k} \frac{\pi(s'|s_1,\cdots,s_k)}{\mu_\pi(s')} \left( \mathbb{P}(x_i = s', x_{i-j} = s_l) - \mathbb{P}(x_{i-j} = s_l)\mu_\pi(s') \right) \\
&= \sum_{s',s_1,\cdots,s_k} \frac{\pi(s'|s_1,\cdots,s_k)}{\mu_\pi(s')} \left( \mathbb{P}(x_i = s', x_{i-j} = s_l) - \mu_\pi(s_l)\mu_\pi(s') \right) \quad \text{(from stationarity)} \\
&= \sum_{s',s_1,\cdots,s_k} \frac{\pi(s'|s_1,\cdots,s_k)}{\mu_\pi(s')} \mathbb{P}(x_i = s', x_{i-j} = s_l) - \sum_{s',s_1,\cdots,s_k} \frac{\pi(s'|s_1,\cdots,s_k)}{\mu_\pi(s')} \mu_\pi(s_l)\mu_\pi(s') \\
&= \sum_{s',s_1,\cdots,s_k} \frac{\pi(s'|s_1,\cdots,s_k)}{\mu_\pi(s')} \mathbb{P}(x_i = s', x_{i-j} = s_l) - |\mathcal{S}|^{k-1} \quad \text{(by marginalization)}
\end{aligned}
$$

Moroever, we also define:

$$
g_{i,j} = \sum_{l=1}^{k} g_{i,j,l}
$$

Hence, on substituting this back to the original equation:

$$
\frac{\mathrm{d}L(\theta)}{\mathrm{d}p_m}\bigg|_i = -\frac{\kappa}{|\mathcal{S}|^k \cdot kn} \cdot \mathbb{E}_{\pi \sim \mathbb{P}(\pi)} \left[ \left( \frac{\exp(p_m)}{\sum_{j=0}^{i}\exp(p_j)} g_{i,m} - \sum_{j=0}^{i} \frac{\exp(p_j)\exp(p_m)}{\left(\sum_{j=0}^{i}\exp(p_j)\right)^2} g_{i,j} \right) \right]
$$

### D.2.1 Warmup: Second Order Markov Chain

We begin by presenting the dynamics for a second-order Markov chain as a warm-up. Since the vocabulary is binary, we have $|\mathcal{S}| = 2$. Therefore,

$$
\begin{aligned}
g_{i,j} &= \sum_{l=1}^{k} g_{i,j,l} \\
&= \sum_{s',s_1,s_2} \left( \frac{\pi(s'\mid s_1, s_2)}{\mu_\pi(s')} \mathbb{P}(x_i = s', x_{i-j} = s_1) + \frac{\pi(s'\mid s_1, s_2)}{\mu_\pi(s')} \mathbb{P}(x_i = s', x_{i-j} = s_2) \right) - 4
\end{aligned}
$$

Enumerating all possible values of $s_1, s_2 \in \{0,1\}$, we get:

$$
\begin{aligned}
g_{i,j} &= \sum_{s'} \left( \frac{\pi(s'\mid 0,0)}{\mu_\pi(s')} \mathbb{P}(x_i = s', x_{i-j} = 0) + \frac{\pi(s'\mid 0,0)}{\mu_\pi(s')} \mathbb{P}(x_i = s', x_{i-j} = 0) \right) \\
&\quad + \sum_{s'} \left( \frac{\pi(s'\mid 1,0)}{\mu_\pi(s')} \mathbb{P}(x_i = s', x_{i-j} = 1) + \frac{\pi(s'\mid 1,0)}{\mu_\pi(s')} \mathbb{P}(x_i = s', x_{i-j} = 0) \right) \\
&\quad + \sum_{s'} \left( \frac{\pi(s'\mid 0,1)}{\mu_\pi(s')} \mathbb{P}(x_i = s', x_{i-j} = 0) + \frac{\pi(s'\mid 0,1)}{\mu_\pi(s')} \mathbb{P}(x_i = s', x_{i-j} = 0) \right) \\
&\quad + \sum_{s'} \left( \frac{\pi(s'\mid 1,1)}{\mu_\pi(s')} \mathbb{P}(x_i = s', x_{i-j} = 1) + \frac{\pi(s'\mid 1,1)}{\mu_\pi(s')} \mathbb{P}(x_i = s', x_{i-j} = 1) \right) - 4
\end{aligned}
$$

Using marginalization, we can simplify terms such as:

$$\sum_{s'} \left( \frac{\pi(s' \mid 1, 0)}{\mu_\pi(s')} \mathbb{P}(x_i = s', x_{i-j} = 1) + \frac{\pi(s' \mid 1, 0)}{\mu_\pi(s')} \mathbb{P}(x_i = s', x_{i-j} = 0) \right)$$

$$= \sum_{s'} \left( \pi(s' \mid 1, 0) \cdot \frac{\mathbb{P}(x_i = s', x_{i-j} = 1) + \mathbb{P}(x_i = s', x_{i-j} = 0)}{\mu_\pi(s')} \right)$$

$$= \sum_{s'} \left( \pi(s' \mid 1, 0) \cdot \frac{\mu_\pi(s')}{\mu_\pi(s')} \right) \qquad \text{(by marginalization)}$$

$$= \sum_{s'} \pi(s' \mid 1, 0)$$

$$= 1$$

Similarly, we obtain:

$$\sum_{s'} \left( \frac{\pi(s' \mid 0, 1)}{\mu_\pi(s')} \mathbb{P}(x_i = s', x_{i-j} = 0) + \frac{\pi(s' \mid 0, 1)}{\mu_\pi(s')} \mathbb{P}(x_i = s', x_{i-j} = 1) \right) = 1$$

Therefore, substituting these identities back into the expression for $g_{i,j}$, we obtain:

$$g_{i,j} = 2 \cdot \left( \sum_{s'} \frac{\pi(s' \mid 0, 0)}{\mu_\pi(s')} \mathbb{P}(x_i = s', x_{i-j} = 0) + \sum_{s'} \frac{\pi(s' \mid 1, 1)}{\mu_\pi(s')} \mathbb{P}(x_i = s', x_{i-j} = 1) - 1 \right)$$

In an ideal second-order Markov chain scenario, we expect the positional scalars $p_1$ and $p_2$ to approach infinity, while the remaining positional weights remain relatively small. This reflects ideal attention behavior: the model should focus primarily on the two preceding tokens, consistent with the structure needed for an accurate $k$-gram (specifically, bigram) estimator. Hence, the requirement is essentially:

$$\left. \frac{\mathrm{d}L(\theta)}{\mathrm{d}p_1} \right|_i = \left. \frac{\mathrm{d}L(\theta)}{\mathrm{d}p_2} \right|_i > \left. \frac{\mathrm{d}L(\theta)}{\mathrm{d}p_k} \right|_i \qquad \forall\, k \in \{0, 3, \ldots, n\}$$

We denote $d_{g_{m,i}} = \left. \frac{\mathrm{d}L(\theta)}{\mathrm{d}p_m} \right|_i$. With this notation, we can express:

$$d_{g_{m,i}} = -\frac{a_{2,0}}{|\mathcal{S}|^k \cdot kn} \cdot \mathbb{E}_{\pi \sim \mathbb{P}(\pi)} \left[ \frac{\exp(p_m)}{\sum_{j=0}^i \exp(p_j)} \cdot \left( g_{i,m} - \sum_{j=0}^i \frac{\exp(p_j)}{\sum_{j=0}^i \exp(p_j)} \cdot g_{i,j} \right) \right]$$

According to assumptions (Assump. 2), the chain satisfies detailed balance. Additionally, by Lemma. 5, we know that the values $g_{i,j}$ decrease monotonically:

$$g_{i,0} > g_{i,1} > g_{i,2} > \cdots > g_{i,n}$$

Using this monotonicity, we can bound the weighted average:

$$\sum_{j=0}^i \frac{\exp(p_j)}{\sum_{j=0}^i \exp(p_j)} \cdot g_{i,j} \leqslant \sum_{j=0}^i \frac{\exp(p_j)}{\sum_{j=0}^i \exp(p_j)} \cdot \sup_j g_{i,j}$$

$$\leqslant \sup_j g_{i,j} = g_{i,0} \quad \text{(by monotonicity)}$$

Now, suppose we initialize all positional encoding scalars to zero except for $p_0 = -\infty$. Then, at initialization time $t = 0$, we have:

$$d_{g_m,i,t=0} = -\frac{a_{2,0}}{|\mathcal{S}|^k \cdot kn} \cdot \frac{1}{i-1} \left( \mathbb{E}_{\pi \sim \mathbb{P}(\pi)}[g_{i,m}] - \sum_{j=1}^{i} \frac{1}{i-1} \cdot \mathbb{E}_{\pi \sim \mathbb{P}(\pi)}[g_{i,j}] \right), \quad \forall\, i \geqslant 1$$

$$d_{g_m,0,t=0} = 0$$

Since we assume detailed balance (Assump. 2) we can now deduce:

$$g_{i,j} = 2 \cdot \left( \sum_{s'} \frac{\pi(s' \mid 0, 0)}{\mu_\pi(s')} \mathbb{P}(x_i = s', x_{i-j} = 0) + \sum_{s'} \frac{\pi(s' \mid 1, 1)}{\mu_\pi(s')} \mathbb{P}(x_i = s', x_{i-j} = 1) - 1 \right)$$

$$= 2 \cdot \left( \sum_{s'} \mathbb{P}(x_{i-j} = 0 \mid x_i = s') \cdot \pi(s' \mid 0, 0) + \sum_{s'} \mathbb{P}(x_{i-j} = 1 \mid x_i = s') \cdot \pi(s' \mid 1, 1) - 1 \right)$$

$$= 2 \cdot \left( \sum_s P^{|i-j|}(s \mid s, s) - 1 \right) \quad \text{(by time-homogeneity and detailed balance)}$$

Under the assumptions on the Markov chain (Assump. 2), we observe that at initialization time $t = 0$ (when all positional encoding scalars are initialized to zero, except for $p_0 = -\infty$), the monotonicity property $g_{i,1} > g_{i,2} > \cdots > g_{i,n}$ holds.

Moreover, by assumption (Assump. 2), we know $g_{i,1} > 1$, which implies:

$$-d_{g_1,i,0} > -d_{g_2,i,0} > \cdots > -d_{g_k,i,0} \quad \text{for all } k \geqslant 0$$

This contradicts the ideal case described earlier, where we would prefer:

$$-d_{g_1,i,0} = -d_{g_2,i,0} > -d_{g_k,i,0},$$
$$\forall\, k \in \{0, 3, \ldots, n\}, \quad \forall\, i \in \{k, \ldots, n\}, \tag{5}$$
$$\text{and} \quad -d_{g_1,i,0} \geqslant 0$$

Towards this, we first define a row-summed quantity:

$$d_{g_m,t} = \sum_{i \geqslant m}^{n} d_{g_m,i,t}$$

To satisfy the condition in the ideal case (Eq. 5), we now introduce a preconditioner-based optimizer:

$$\mathbf{p}_{t+1} = \mathbf{p}_t - \eta_t \mathbf{D}_t^{-1} \nabla_{\mathbf{p}_t} L(\theta)$$

where

$$\mathbf{p}_t = [p_{0,t} \quad p_{1,t} \quad \cdots \quad p_{n,t}]^\top \in \mathbb{R}^{n+1}$$
$$\mathbf{D}_t^{-1} = \text{diag}\left( 0, 1, \frac{d_{g_1,t}}{d_{g_2,t}}, 1, 1, \ldots, 1 \right) \in \mathbb{R}^{(n+1) \times (n+1)}$$

Here, $p_{i,t}$ denotes the $i$th positional encoding scalar at iteration $t$. Therefore, after the first update step (i.e., at $t = 0$), we obtain:

$$p_{1,1} = p_{2,1} > p_{k,1} \quad \forall k \in \{0, 3, \ldots, n\}$$

Note that while we assumed $p_0 = -\infty$ for analysis, in practice we can set it to a large negative constant to ensure that $-d_{g_m,i,t=0} > 0$. We need to show that this property holds for all iterates. By Lemma 3, we can show that for each iterate of the preconditioned descent, the following holds:

$$p_{1,t} = p_{2,t} > p_{k,t} \quad \forall\, k \in \{0, 3, \ldots, n\}$$

Furthermore, by Lemma 3 and Lemma 4, we observe that $p_1$ and $p_2$ tend to infinity, as they increase monotonically over time, whereas the remaining positional scalars grow at a much slower rate and thus either diverge to $-\infty$ or converge to a finite constant. In addition, Lemma 4 characterizes the approximate time required for the iterates to reach the optimal attention configuration. Therefore, by the end of the first stage of training (as $T_1 \to \infty$), we recover the ideal attention map corresponding to a second-order Markov chain:

$$\mathbf{v}_i \approx \frac{1}{2}(e_{x_{i-1}} + e_{x_{i-2}})$$

By reintroducing the layer normalization and taking the limit $k \to \infty$, we recover the conditional 2-gram estimator, as shown in Appendix A.

### D.2.2 Any-Order Markov Chain

Following up from the previous section, we follow the same philosophy as before. Similar to before, we expect that in an ideal scenario, we expect the positional scalars $p_1, p_2, \cdots p_k$ to approach infinity, while the remaining positional weights remain relatively small. This reflects ideal attention behavior consistent with the structure needed for an accurate $k$-gram estimator. Hence, the requirement is essentially:

$$\frac{\mathrm{d}L(\theta)}{\mathrm{d}p_1}\bigg|_i = \frac{\mathrm{d}L(\theta)}{\mathrm{d}p_2}\bigg|_i \cdots = \frac{\mathrm{d}L(\theta)}{\mathrm{d}p_k}\bigg|_i > \frac{\mathrm{d}L(\theta)}{\mathrm{d}p_q}\bigg|_i \quad \forall\, q \in \{0, k+1, \ldots, n\}$$

We denote $d_{g_m,i} = \frac{\mathrm{d}L(\theta)}{\mathrm{d}p_m}\big|_i$. With this notation, we can express:

$$d_{g_m,i} = -\frac{a_{2,0}}{|\mathcal{S}|^k \cdot kn} \cdot \mathbb{E}_{\pi \sim \mathbb{P}(\pi)} \left[ \frac{\exp(p_m)}{\sum_{j=0}^{i} \exp(p_j)} \cdot \left( g_{i,m} - \sum_{j=0}^{i} \frac{\exp(p_j)}{\sum_{j=0}^{i} \exp(p_j)} \cdot g_{i,j} \right) \right]$$

Now, suppose we initialize all positional encoding scalars to zero except for $p_0 = -\infty$. Then, at initialization time $t = 0$, we have:

$$d_{g_m,i,t=0} = -\frac{a_{2,0}}{|\mathcal{S}|^k \cdot kn} \cdot \frac{1}{i-1} \left( \mathbb{E}_{\pi \sim \mathbb{P}(\pi)}[g_{i,m}] - \sum_{j=1}^{i} \frac{1}{i-1} \cdot \mathbb{E}_{\pi \sim \mathbb{P}(\pi)}[g_{i,j}] \right), \quad \forall\, i \geq 1$$

$$d_{g_m,0,t=0} = 0$$

Under the assumptions on the Markov chain (Assump. 3) and Lemma. 6, we observe that at initialization time $t = 0$ (when all positional encoding scalars are initialized to zero, except for $p_0 = -\infty$), the monotonicity property $g_{i,1} > g_{i,2} > \cdots > g_{i,n}$ holds. Moreover, by Assump. 3, we know $g_{i,1} > 1$, which implies:

$$-d_{g_1,i,0} > -d_{g_2,i,0} > \cdots > -d_{g_k,i,0} \quad \forall \geq 0,\ \forall i \geq 0$$

This contradicts the ideal case described earlier, where we would prefer:

$$-d_{g_1,i,0} = -d_{g_2,i,0} = \cdots = -d_{g_k,i,0} > -d_{g_q,i,0},$$
$$\forall\, q \in \{0, k+1, \ldots, n\}, \quad \forall\, i \in \{k, \ldots, n\}, \tag{6}$$
$$\text{and} \quad -d_{g_1,i,0} \geq 0$$

Towards this, we first define a row-summed quantity:

$$d_{g_m,t} = \sum_{i \ge m}^{n} d_{g_m,i,t}$$

To satisfy the condition in the ideal case (Eq. 6), we now introduce a preconditioner-based optimizer:

$$\mathbf{p}_{t+1} = \mathbf{p}_t - \eta_t \, \mathbf{D}_t^{-1} \nabla_{\mathbf{p}_t} L(\theta)$$

where

$$\mathbf{p}_t = \begin{bmatrix} p_{0,t} & p_{1,t} & \cdots & p_{n,t} \end{bmatrix}^\top \in \mathbb{R}^{n+1}$$

$$\mathbf{D}_t^{-1} = \mathrm{diag}\left(0, 1, \frac{d_{g_1,t}}{d_{g_2,t}}, \frac{d_{g_1,t}}{d_{g_3,t}}, \cdots, \frac{d_{g_1,t}}{d_{g_k,t}}, 1, \ldots, 1\right) \in \mathbb{R}^{(n+1)\times(n+1)}$$

Here, $p_{i,t}$ denotes the $i$th positional encoding scalar at iteration $t$. Therefore, after the first update step (i.e., at $t = 0$), we obtain:

$$p_{1,1} = p_{2,1} = \cdots = p_{k,1} > p_{q,1} \quad \forall q \in \{0, k+1, \ldots, n\}$$

Note that while we assumed $p_0 = -\infty$ for analysis, in practice we can set it to a large negative constant to ensure that $-d_{g_m,i,t=0} > 0$. We need to show that this property holds for all iterates. By Lemma 3, we can show that for each iterate of the preconditioned descent, the following holds:

$$p_{1,t} = p_{2,t} = \cdots = p_{k,t} > p_{q,t} \quad \forall\, q \in \{0, k+1, \ldots, n\}$$

Furthermore, by Lemma 3 and Lemma 4, we observe that $p_1$ and $p_2$ tend to infinity as they increase monotonically over time, whereas the other positional scalars either tend to $-\infty$ or converge to a constant. In addition, Lemma 4 provides the approximate time required for the iterates to reach the optimal attention configuration. Therefore, by the end of the first stage of training (as $T_1 \to \infty$), we can recover the ideal attention map corresponding to a $k^{\text{th}}$-order Markov chain:

$$\mathbf{v}_i \approx \frac{1}{k}(e_{x_{i-1}} + e_{x_{i-2}} \cdots + e_{x_{i-k}})$$

By reintroducing the layer normalization and taking the limit $\kappa \to \infty$, we recover the conditional k-gram estimator, as shown in Appendix A.

# E Training Dynamics: Stage One Lemmas

We use the following notation:

$$\text{logit}_{s'_{s_k},n} = e_{s'}^T \left[ X_n \, \text{softmax} \left( a_2 Z_n \right) \right] \tag{7}$$

$$= e_{s'}^T \left[ X_n \, \text{softmax} \left( \text{softmax} \left( A^{(1)} \right) X_n^T A^{(2)} \frac{1}{k} \left( \sum_{i=0}^{k-1} e_{x_{n-i}} \right) \right) \right] \tag{8}$$

Initially, we know that $A^{(2)} = a_{2,0} \cdot I$

Hence, we can rewrite this as:

$$\text{logit}_{s_k,n} = X_n \, \text{softmax} \left( \underbrace{a_{2,0} \cdot \text{softmax} \left( A^{(1)} \right) X_n^T \frac{1}{k} \left( \sum_{i=0}^{k-1} e_{x_{n-i}} \right)}_{\mathcal{A}_\theta^{(2)}(X_n;k)} \right) \tag{9}$$

**Lemma 2** (Exension of Lemma G.1. in Nichani et al. [20] to $k^{\text{th}}$-order). *Let $\theta = (A^{(1)}, a_{2,0}I)$, $\hat{\theta} = (A^{(1)}, 0)$, for $a_{2,0} \leqslant 1$. Define $g_i^*, \hat{g}_i \in \mathbb{R}^i$ by*

$$g_i^* := n \sum_{s',s_1,\cdots s_k} \mathbb{E}_{\pi \sim \mathbb{P}(\pi)} \left[ \pi(s' \mid s_1, \cdots, s_k) \frac{e_{s'}^\top X_n D(\mathcal{A}_\theta^{(2)}(X_n;k))e_i \cdot X_n^T \left( \sum_{i=0}^{k-1} \frac{1}{k} e_{x_{n-i}=s_{i+1}} \right)}{\text{logit}_{s'_{s_k},n,\theta} + \epsilon} \right],$$

$$\hat{g}_i := n \sum_{s',s_1,\cdots s_k} \mathbb{E}_{\pi \sim \mathbb{P}(\pi)} \left[ \pi(s' \mid s_1, \cdots, s_k) \frac{e_{s'}^\top X_n D(\mathcal{A}_{\hat{\theta}}^{(2)}(X_n;k))e_i \cdot X_n^T \left( \sum_{i=0}^{k-1} \frac{1}{k} e_{x_{n-i}=s_{i+1}} \right)}{\text{logit}_{s'_{s_k},n,\hat{\theta}} + \epsilon} \right].$$

*Then, it holds that*

$$\|g_i^* - \hat{g}_i\|_\infty \leqslant 3S^{k+1}\epsilon^{-2}(e^{a_{2,0}} - 1) \leqslant 6S^{k+1}\epsilon^{-2}a_{2,0}$$

*Proof.* The proof follows similarly to the proof of Lemma G.1 in Nichani et al. [20]. We begin by bounding the difference inside the expectation.

We first apply the triangle inequality to separate the difference between the two fractions:

$$\left| \frac{e_{s'}^\top X_n D(\mathcal{A}_\theta^{(2)}(X_n;k))e_i}{\text{logit}_{s'_{s_k},n,\theta} + \epsilon} - \frac{e_{s'}^\top X_n D(\mathcal{A}_{\hat{\theta}}^{(2)}(X_n;k))e_i}{\text{logit}_{s'_{s_k},n,\hat{\theta}} + \epsilon} \right|$$

$$\leqslant \left| \frac{1}{\text{logit}_{s'_{s_k},n,\theta} + \epsilon} - \frac{1}{\text{logit}_{s'_{s_k},n,\hat{\theta}} + \epsilon} \right| \cdot \left| e_{s'}^\top X_n D(\mathcal{A}_\theta^{(2)}(X_n;k))e_i \right|$$

$$+ \left| \frac{1}{\text{logit}_{s'_{s_k},n,\hat{\theta}} + \epsilon} \right| \cdot \left| e_{s'}^\top X_n \left( D(\mathcal{A}_\theta^{(2)}(X_n;k)) - D(\mathcal{A}_{\hat{\theta}}^{(2)}(X_n;k)) \right) e_i \right|.$$

To bound this expression, we begin by bounding the logit difference in the numerator of the first term.

$$\left| \text{logit}_{s'_{s_k},n,\theta} - \text{logit}_{s'_{s_k},n,\hat{\theta}} \right| = \left| e_{s'}^\top X_n(\mathcal{A}_\theta^{(2)}(X_n;k) - \mathcal{A}_{\hat{\theta}}^{(2)}(X_n;k)) \right|$$

$$\leqslant \|e_{s'}^\top X_n\|_\infty \cdot \|\mathcal{A}_\theta^{(2)}(X_n;k) - \mathcal{A}_{\hat{\theta}}^{(2)}(X_n;k))\|$$

$$\leqslant \|\mathcal{A}_\theta^{(2)}(X_n;k) - \mathcal{A}_{\hat{\theta}}^{(2)}(X_n;k))\|_1,$$

where the last inequality uses the fact that $\|e_{s'}^\top X_n\|_\infty \leqslant 1$.

We now recall the definition of $\mathcal{A}_\theta^{(2)}$ and analyze its properties.

$$\mathcal{A}_\theta^{(2)}(X_n; k)) = \text{softmax}\left(a_{2,0} \cdot \text{softmax}(A^{(1)})X_n^T \cdot \frac{1}{k}\sum_{i=0}^{k-1} e_{x_{n-i}}\right)$$

In particular, if $a_{2,0} = 0$, the argument to softmax is zero, and we obtain the uniform distribution:

$$\mathcal{A}_\theta^{(2)}(X_n; k)) = \text{softmax}(\mathbf{0})$$

Since the entries of the softmax input are in $[0, 1]$, we can bound the minimum and maximum values of any softmax output entry as follows:

$$\frac{1}{(n-1)e^{a_{2,0}}+1} \leqslant \mathcal{A}_\theta^{(2)}(X_n; k))_i \leqslant \frac{e^{a_{2,0}}}{(n-1)+e^{a_{2,0}}}$$

These bounds let us now quantify the maximum possible deviation between corresponding softmax outputs.

$$\begin{aligned}
|\sup \mathcal{A}_\theta^{(2)}(X_n; k))_i - \mathcal{A}_{\hat\theta}^{(2)}(X_n; k))_i| &= \left|\frac{e^{a_{2,0}}}{(n-1)+e^{a_{2,0}}} - \frac{1}{n}\right| \\
&= \left|\frac{(n-1)(e^{a_{2,0}}-1)}{n((n-1)+e^{a_{2,0}})}\right| \\
&\leqslant \frac{e^{a_{2,0}}-1}{n}
\end{aligned}$$

Similarly, for the lower bound:

$$\begin{aligned}
|\inf \mathcal{A}_\theta^{(2)}(X_n; k))_i - \mathcal{A}_{\hat\theta}^{(2)}(X_n; k))_i| &= \left|\frac{1}{(n-1)e^{a_{2,0}}+1} - \frac{1}{n}\right| \\
&\leqslant \frac{e^{a_{2,0}}-1}{n}
\end{aligned}$$

Thus, we conclude:

$$|\mathcal{A}_\theta^{(2)}(X_n; k))_i - \mathcal{A}_{\hat\theta}^{(2)}(X_n; k))_i| \leqslant \frac{e^{a_{2,0}}-1}{n}$$

Next, we simplify the derivative expression using the identity for the softmax Jacobian.

$$e_{s'}^\top X_n D(\mathcal{A}_\theta^{(2)}(X_n; k))e_i = \mathcal{A}_\theta^{(2)}(X_n; k)_i(\mathbb{I}_{x_i=e_{s'}} - \text{logit}_{s'_{s_k},n,\theta})$$

We now bound the difference in these derivative expressions using triangle inequality:

$$\begin{aligned}
&\left|e_{s'}^\top X_n D(\mathcal{A}_\theta^{(2)}(X_n; k))e_i - e_{s'}^\top X_n D(\mathcal{A}_{\hat\theta}^{(2)}(X_n; k))e_i\right| \\
&\leqslant |\mathcal{A}_\theta^{(2)}(X_n; k)_i - \mathcal{A}_{\hat\theta}^{(2)}(X_n; k)_i| \cdot |\mathbb{I}_{x_i=e_{s'}} - \text{logit}_{s'_{s_k},n,\theta}|
\end{aligned}$$

$$+ \mathcal{A}_{\hat{\theta}}^{(2)}(X_n; k)_i \cdot | \text{logit}_{s'_{s_k}, n, \theta} - \text{logit}_{s'_{s_k}, n, \hat{\theta}} |$$

$$\leqslant 2 \cdot \frac{e^{a_{2,0}} - 1}{n}$$

Here we used that $|\mathbb{I}_{x_i = e_{s'}} - \text{logit}| \leqslant 1$ and $\mathcal{A}_{\hat{\theta}}^{(2)} \leqslant 1$.

Combining the bounds for the numerator and denominator differences, we obtain:

$$\left| \frac{e_{s'}^{\top} X_n D(\mathcal{A}_{\theta}^{(2)}(X_n; k)) e_i}{\text{logit}_{s'_{s_k}, n, \theta} + \epsilon} - \frac{e_{s'}^{\top} X_n D(\mathcal{A}_{\hat{\theta}}^{(2)}(X_n; k)) e_i}{\text{logit}_{s'_{s_k}, n, \hat{\theta}} + \epsilon} \right| \leqslant 3 \cdot \frac{e^{a_{2,0}} - 1}{\epsilon^2 n}$$

Finally, we bound the overall error $\|g_i^* - \hat{g}_i\|_{\infty}$ by summing over all possible state sequences:

$$\|g_i^* - \hat{g}_i\|_{\infty} \leqslant n \sum_{s', s_1, \cdots, s_k} \mathbb{E}_{\pi \sim \mathbb{P}(\pi)} \left[ \pi(s' \mid s_1, \cdots, s_k) \cdot \frac{3(e^{a_{2,0}} - 1)}{\epsilon^2 n} \right]$$

$$= 3 \cdot \frac{(e^{a_{2,0}} - 1)}{\epsilon^2} \sum_{s', s_1, \cdots, s_k} \mathbb{E}_{\pi} \left[ \pi(s' \mid s_1, \cdots, s_k) \right]$$

$$\leqslant 3 S^{k+1} \cdot \epsilon^{-2} \cdot (e^{a_{2,0}} - 1)$$

Using the inequality $e^{a_{2,0}} - 1 \leqslant 2 a_{2,0}$ for $a_{2,0} \in [0, 1]$, we conclude:

$$\|g_i^* - \hat{g}_i\|_{\infty} \leqslant 6 S^{k+1} \epsilon^{-2} a_{2,0}$$

$\square$

**Lemma 3** (Extension of Lemma D.3 in Nichani et al. [20]). *Let $A^{(2)} = a_{2,0} I$. There exist constants $c_{\gamma,S}, C_{\gamma,S}$ such that, if $a_{2,0} \leqslant c_{\gamma,S}(n(1 - \lambda))^{-3/2}$. If $r \in \mathcal{I}$, where $\mathcal{I} = \{1, \cdots, k\}$, where $k$ denotes the order of the markov chain:*

$$G^{(1)}(A^{(1)}, A^{(2)})_{i,r} \leqslant G^{(1)}(A^{(1)}, A^{(2)})_{i,j} - \text{softmax}(A_i^{(1)})_r (1 - \text{softmax}(A_i^{(1)})_r) \cdot \frac{C_{\gamma,S} a_{2,0}}{n}.$$

*Note that the subscript $i$ in this case denotes the $i^{th}$ row and the subscript $i, j$ denotes the $j^{th}$ element in the $i^{th}$ row.*

*Proof.* The proof follows similarly to Nichani et al. [20]. The gradient $G^{(1)}(A^{(1)}, A^{(2)})_i$ can be expanded as

$$G^{(1)}(A^{(1)}, A^{(2)})_i = -a_{2,0} D(\text{softmax}(A_i^{(1)})) \cdot \frac{1}{S} \sum_{s', s_1, \cdots, s_k} \mathbb{E}_{\pi, X} \left[ \pi(s' \mid s_1, \cdots, s_k) \, \xi_{s', s_1, \cdots, s_k}(X) \right],$$

where

$$\xi_{s', s_1, \cdots, s_k}(X) = \frac{1}{\text{logit}_{s'_{s_k}, n} + \epsilon} e_{s'}^{\top} X_n D(\mathcal{A}_{\theta}^{(2)}(X_n; k)) e_i \left( \sum_{i=0}^{k-1} \frac{1}{k} e_{x_{n-i} = s_{i+1}} \right).$$

Define

$$g_i^* := n \sum_{s', s_1, \cdots, s_k} \mathbb{E}_{\pi \sim \mathbb{P}(\pi)} \left[ \pi(s' \mid s_1, \cdots, s_k) \, \xi_{s', s_1, \cdots, s_k}(X) \right],$$

$$\hat{g}_i := n \sum_{s', s_1, \cdots, s_k} \mathbb{E}_{\pi \sim \mathbb{P}(\pi)} \left[ \pi(s' \mid s_1, \cdots, s_k) \, \hat{\xi}_{s', s_1, \cdots, s_k}(X) \right],$$

where $\hat{\xi}_{s,s'}(X)$ is evaluated at $\hat{\theta} = (A^{(1)}, 0)$.

Thus,

$$G^{(1)}(A^{(1)}, A^{(2)})_i = -\frac{a_{2,0}}{Sn} D(\text{softmax}(A_i^{(1)})) g_i^*.$$

Since $a_{2,0} \leqslant c_{\gamma,S}(n(1-\lambda))^{-3/2}$ and applying Lemma 2, we obtain

$$\|\hat{g}_i - g_i^*\|_\infty \leqslant \frac{C}{\sqrt{n(1-\lambda)}}.$$

For some arbitrary constant $C$. Thus, it suffices to analyze $\hat{g}_i$.

**Computation of $\hat{g}_i$ under $\hat{\theta}$.** Under $\hat{\theta}$,

$$\mathcal{A}_{\hat{\theta}}^{(2)}(X_n; k) = \frac{1}{n}\mathbf{1}_n,$$

thus,

$$\text{logit}_{s',n}^{\hat{\theta}} = \hat{\mu}_X(s')$$

Therefore,

$$e_{s'}^\top X_n D(\mathcal{A}_{\hat{\theta}}^{(2)}(X_n; k)) e_i = \frac{1}{n}(\mathbb{I}_{x_i = s'} - \hat{\mu}_X(s')).$$

Thus, the $j$-th coordinate satisfies

$$\hat{g}_{i,j} = \sum_{s', s_1, \cdots, s_k} \mathbb{E}_{\pi \sim \mathbb{P}(\pi)} \left[ \frac{\pi(s' \mid s_1, \cdots, s_k)}{(\hat{\mu}_X(s') + \epsilon)} (\mathbb{I}_{x_i = s'} - \hat{\mu}_X(s')) \sum_{i=0}^{k-1} \frac{1}{k} \mathbb{I}_{x_j = s_{i+1}} \right].$$

Applying Lemma 11

$$|\hat{g}_{i,j} - g_{i,j}| \leqslant \frac{C_{\gamma,S}}{\sqrt{n(1-\lambda)}}.$$

By Lemma 7 (for first order Markov chains) or by (App. D.2.1, Lemma. 5 and Assump. 2 for second-order Markov chains) or by (App. D.2.2, Lemma. 6 and Assump. 3 for any-order Markov chains), for all $j \neq r \in \mathcal{I}$,

$$g_{i,j}^* - g_{i,r}^* \leqslant g_{i,j} - g_{i,r} + |g_{i,j} - g_{i,j}^*| + |g_{i,p(i)} - g_{i,r}^*|$$
$$\leqslant -\delta + \frac{C'_{\gamma,S}}{\sqrt{n(1-\lambda)}}$$
$$\leqslant -\delta/2.$$

Note that, in the last inequality, we assume $n$ to be long enough for this to hold.

Now expanding:

$$G^{(1)}(A^{(1)}, A^{(2)})_{i,j} = -\frac{a_{2,0}}{Sn} \left( \text{softmax}(A_i^{(1)})_j g_{i,j}^* - (\text{softmax}(A_i^{(1)}))^\top g_i^* \, \text{softmax}(A_i^{(1)})_j \right),$$
$$G^{(1)}(A^{(1)}, A^{(2)})_{i,r} = -\frac{a_{2,0}}{Sn} \left( \text{softmax}(A_i^{(1)})_r g_{i,r}^* - (\text{softmax}(A_i^{(1)}))^\top g_i^* \, \text{softmax}(A_i^{(1)})_r \right).$$

Thus the difference is

$$G^{(1)}(A^{(1)}, A^{(2)})_{i,j} - G^{(1)}(A^{(1)}, A^{(2)})_{i,r}$$

$$= \frac{a_{2,0}}{Sn} \left[ \left( \mathrm{softmax}(A_i^{(1)})_r - \mathrm{softmax}(A_i^{(1)})_j \right) \left( g_{i,r}^* - (\mathrm{softmax}(A_i^{(1)}))^\top g_i^* \right) \right.$$

$$\left. + \mathrm{softmax}(A_i^{(1)})_j \left( g_{i,r}^* - g_{i,j}^* \right) \right] \quad \forall r \in \mathcal{I}$$

Applying bounds:

$$g_{i,r}^* - g_{i,j}^* \geqslant \frac{\delta}{2},$$

thus,

$$G^{(1)}(A^{(1)}, A^{(2)})_{i,j} - G^{(1)}(A^{(1)}, A^{(2)})_{i,r} \geqslant \frac{a_{2,0}}{Sn} \mathrm{softmax}(A_i^{(1)})_r (1 - \mathrm{softmax}(A_i^{(1)})_r \frac{\delta}{2}.$$

$\square$

**Lemma 4** (Extension of Lemma D.5 in [20]: Bounding the time of convergence). *Let $A^{(2)}(0) = a_{2,0} I_S$, where $a_{2,0} \leqslant c_{\gamma,S}\, n^{-3/2}(1-\lambda)^{-3/2}$. Let $\mathcal{I} = \{1, \cdots, k\}$. Then there exists*

$$\tau_1 \lesssim n\eta_1^{-1} a_{2,0}^{-1} \left( n\log(2n) + \alpha^{-1} \log\left( \frac{\left(\frac{1}{k} - \alpha\right)(n-k)}{1 - k\left(\frac{1}{k} - \alpha\right)} \right) \right)$$

*such that for any $t \geqslant \tau_1$,*

$$\mathrm{softmax}(A^{(1)}(t))_{i,r} \geqslant \frac{1}{k} - \alpha$$

*for all $i$ and all $r \in \mathcal{I}$.*

*Proof.* The proof follows similarly to the proof of Lemma D.5 in Nichani et al. [20]. Let $\mathcal{I} = \{1, \ldots, k\}$. Then, by Lemma 3, we have that throughout training,

$$A^{(1)}(t)_{i,r} \geqslant A^{(1)}(t)_{i,j} \quad \text{for all } j \notin \mathcal{I},$$

for all $r \in \mathcal{I}$.

Moreover, by Lemma 3, the quantity $\mathrm{softmax}(A^{(1)}(t))_{i,r}$ is monotonically increasing in $t$ for all $r \in \mathcal{I}$.

For a fixed row $i$, define the gap function:

$$\Delta(t) := A^{(1)}(t)_{i,r} - \max_{j \notin \mathcal{I}} A^{(1)}(t)_{i,j}.$$

By properties of the softmax function, we can compute the lower bound:

$$\mathrm{softmax}(A^{(1)}(t))_{i,r} \geqslant \frac{\exp(\Delta(t))}{n - k + k\exp(\Delta(t))}.$$

Now, consider the first time $\tau_+(1/2)$ such that

$$\mathrm{softmax}(A^{(1)}(\tau_+(1/2)))_{i,r} > \frac{1}{2k}.$$

For all $t < \tau_+(1/2)$, we have

$$1 - \mathrm{softmax}(A^{(1)}(t))_{i,r} \geqslant \frac{1}{2k}.$$

Thus, by Lemma 3, the gap $\Delta(t)$ evolves as

$$\Delta(t+1) \geqslant \Delta(t) + \frac{C_{\gamma,S}\, a_{2,0}}{n^2} \eta_1.$$

Consequently, by successive addition over time,

$$\Delta(\tau_+(1/2)) \gtrsim \frac{a_{2,0}\eta_1}{n^2}\tau_+(1/2).$$

Suppose for contradiction that

$$\Delta(\tau_+(1/2)) \geq \log(2n).$$

Then we have

$$\text{softmax}(A^{(1)}(\tau_+(1/2)))_{i,r} \geq \frac{\exp(\log(2n))}{n - k + k\exp(\log(2n))}$$

$$= \frac{2n}{n - k + 2kn}$$

$$= \frac{2}{2k + 1 - k/n}.$$

Since $k \geq 1$ and $n$ is large, we have

$$2k + 1 - \frac{k}{n} \leq 4k,$$

and thus

$$\frac{2}{2k + 1 - k/n} \geq \frac{1}{2k}.$$

This contradicts the definition of $\tau_+(1/2)$ as the first time when $\text{softmax}(A^{(1)}(t))_{i,r} > \frac{1}{2k}$. Therefore, we must have

$$\Delta(\tau_+(1/2)) \leq \log(2n),$$

and hence

$$\tau_+(1/2) \lesssim n^2\eta_1^{-1}a_{2,0}^{-1}\log(2n).$$

Now define $\tau_+(\alpha)$ as the first time $t$ such that

$$\text{softmax}(A^{(1)}(t))_{i,r} < \frac{1}{k} - \alpha, \quad \text{for all } r \in \mathcal{I}.$$

This implies that, for a sufficiently small positive $\alpha$, to reach a contradiction, we would require:

$$\frac{e^{\Delta(\tau_+(\alpha))}}{n - k + k \cdot e^{\Delta(\tau_+(\alpha))}} \geq \frac{1}{k} - \alpha,$$

$$\Delta(\tau_+(\alpha)) \geq \log\left(\frac{\left(\frac{1}{k} - \alpha\right)(n - k)}{1 - k\left(\frac{1}{k} - \alpha\right)}\right).$$

For $t \in [\tau_+(1/2), \tau_+(\alpha))$, the gap $\Delta(t)$ evolves according to

$$\Delta(t + 1) \geq \Delta(t) + \frac{C_{\gamma,S}a_{2,0}\alpha}{n}\eta_1.$$

Therefore, in order to achieve the required increase in $\Delta(t)$, we must satisfy

$$\tau_+(\alpha) - \tau_+(1/2) \lesssim n\alpha^{-1}a_{2,0}^{-1}\log\left(\frac{\left(\frac{1}{k} - \alpha\right)(n - k)}{1 - k\left(\frac{1}{k} - \alpha\right)}\right).$$

Combining with the earlier phase where

$$\tau_+(1/2) \lesssim n^2\eta_1^{-1}a_{2,0}^{-1}\log(2n),$$

we conclude

$$\tau_+(\alpha) \lesssim n^2 \eta_1^{-1} a_{2,0}^{-1} \log(2n) + n\alpha^{-1}\eta_1^{-1} a_{2,0}^{-1} \log\left(\frac{\left(\frac{1}{k} - \alpha\right)(n-k)}{1 - k\left(\frac{1}{k} - \alpha\right)}\right).$$

Grouping terms, we obtain

$$\tau_+(\alpha) \lesssim n\eta_1^{-1} a_{2,0}^{-1}\left(n\log(2n) + \alpha^{-1}\log\left(\frac{\left(\frac{1}{k} - \alpha\right)(n-k)}{1 - k\left(\frac{1}{k} - \alpha\right)}\right)\right).$$

$\square$

## F  Higher Order Markov Chain Lemmas

**Lemma 5** (Successive Markov Chain Lemmas). *For a second-order Markov Chain, if $\pi(0 \mid 0,0) - \pi(0 \mid 1,0) \geqslant \delta'$ and $\pi(1|1,1) - \pi(1|0,1) \geqslant \delta'$ then the probability $P^i(0 \mid 0,0)$ and $P^i(1 \mid 1,1)$ decreases with increasing $i$. Moreover,*

$$\mathbb{E}_{\pi \sim \mathbb{P}}\left[P^1(0 \mid 0,0) + P^1(1 \mid 1,1)\right] - \mathbb{E}_{\pi \sim \mathbb{P}}\left[P^k(0 \mid 0,0) + P^k(1 \mid 1,1)\right] \geqslant \delta \quad \forall k \geqslant 4$$

*Proof.* We begin by establishing the condition for two steps and then generalize by induction.

$$P^2(0 \mid 0,0) = \pi(0 \mid 0,0)\pi(0 \mid 0,0) + \pi(0 \mid 1,0)\pi(1 \mid 0,0)$$

Given the assumption $\pi(0 \mid 0,0) > \pi(0 \mid 1,0)$, we multiply both sides by $1 - \pi(0 \mid 0,0)$:

$$\pi(0 \mid 0,0)\left(1 - \pi(0 \mid 0,0)\right) > \pi(0 \mid 1,0)\left(1 - \pi(0 \mid 0,0)\right)$$
$$\pi(0 \mid 0,0) > \pi(0 \mid 0,0)^2 + \pi(0 \mid 1,0)\pi(1 \mid 0,0)$$
$$\pi(0 \mid 0,0) > P^2(0 \mid 0,0) \quad \text{(by definition)}$$

Now, we apply induction.

$$\begin{aligned}
P^3(0 \mid 0,0) &= \pi(0 \mid 0,0)P^2(0 \mid 0,0) + \pi(0 \mid 1,0)(1 - P^2(0 \mid 0,0)) \\
&= \pi(0 \mid 1,0) + (\pi(0 \mid 0,0) - \pi(0 \mid 1,0))\,P^2(0 \mid 0,0) \\
&< \pi(0 \mid 1,0) + (\pi(0 \mid 0,0) - \pi(0 \mid 1,0))\,\pi(0 \mid 0,0) \\
&< P^2(0 \mid 0,0)
\end{aligned}$$

Hence, by induction:

$$\begin{aligned}
P^{i+1}(0 \mid 0,0) &= \pi(0 \mid 1,0) + (\pi(0 \mid 0,0) - \pi(0 \mid 1,0))\,P^i(0 \mid 0,0) \\
&< \pi(0 \mid 1,0) + (\pi(0 \mid 0,0) - \pi(0 \mid 1,0))\,P^{i-1}(0 \mid 0,0) \\
&< P^i(0 \mid 0,0)
\end{aligned}$$

Now, using the recursive relation derived above, we can express $P^i(0 \mid 0,0)$ explicitly. The recurrence is:

$$P^{i+1}(0 \mid 0,0) = \pi(0 \mid 1,0) + (\pi(0 \mid 0,0) - \pi(0 \mid 1,0))\,P^i(0 \mid 0,0)$$

We separate this into a homogeneous and a particular solution:

**Homogeneous Solution:**

$$P_h^{i+1}(0 \mid 0,0) = (\pi(0 \mid 0,0) - \pi(0 \mid 1,0))\,P_h^i(0 \mid 0,0)$$
$$P_h^i(0 \mid 0,0) = C\left(\pi(0 \mid 0,0) - \pi(0 \mid 1,0)\right)^i$$

**Particular Solution:**

$$P_p = \pi(0 \mid 1,0) + (\pi(0 \mid 0,0) - \pi(0 \mid 1,0))\,P_p$$

$$P_p = \frac{\pi(0 \mid 1,0)}{1 - \pi(0 \mid 0,0) + \pi(0 \mid 1,0)}$$

Using the initial condition $P^1(0 \mid 0,0) = \pi(0 \mid 0,0)$, we solve for the constant:

$$C = \frac{\pi(0 \mid 0,0) - P_p}{\pi(0 \mid 0,0) - \pi(0 \mid 1,0)}$$

**Final Solution**

$$
\begin{aligned}
P^i(0 \mid 0,0) = {} & \frac{\pi(0 \mid 1,0)}{1 - \pi(0 \mid 0,0) + \pi(0 \mid 1,0)} \\
& + \left( \pi(0 \mid 0,0) - \frac{\pi(0 \mid 1,0)}{1 - \pi(0 \mid 0,0) + \pi(0 \mid 1,0)} \right) (\pi(0 \mid 0,0) - \pi(0 \mid 1,0))^{i-1}
\end{aligned}
$$

Similarly, we obtain:

$$P^{i+1}(1 \mid 1,1) < P^i(1 \mid 1,1) \quad \forall i \in \mathbb{N}$$

Now, we can first express

$$
\begin{aligned}
P^i(1 \mid 1,1) = {} & \frac{\pi(1 \mid 0,1)}{1 - \pi(1 \mid 1,1) + \pi(1 \mid 0,1)} \\
& + \left( \pi(1 \mid 1,1) - \frac{\pi(1 \mid 0,1)}{1 - \pi(1 \mid 1,1) + \pi(1 \mid 0,1)} \right) (\pi(1 \mid 1,1) - \pi(1 \mid 0,1))^{i-1}.
\end{aligned}
$$

Now, define

$$
\begin{aligned}
G_0 &= \left( \pi(0 \mid 0,0) - \frac{\pi(0 \mid 1,0)}{1 - \pi(0 \mid 0,0) + \pi(0 \mid 1,0)} \right), \\
G_1 &= \left( \pi(1 \mid 1,1) - \frac{\pi(1 \mid 0,1)}{1 - \pi(1 \mid 1,1) + \pi(1 \mid 0,1)} \right).
\end{aligned}
$$

Thus, we can write

$$
\begin{aligned}
P^3(0 \mid 0,0) &= \frac{\pi(0 \mid 1,0)}{1 - \pi(0 \mid 0,0) + \pi(0 \mid 1,0)} + G_0 \cdot (\pi(0 \mid 0,0) - \pi(0 \mid 1,0))^2, \\
P^i(0 \mid 0,0) &= \frac{\pi(0 \mid 1,0)}{1 - \pi(0 \mid 0,0) + \pi(0 \mid 1,0)} + G_0 \cdot (\pi(0 \mid 0,0) - \pi(0 \mid 1,0))^{i-1},
\end{aligned}
$$

and therefore

$$P^3(0 \mid 0,0) - P^i(0 \mid 0,0) = G_0 \left( \pi(0 \mid 0,0) - \pi(0 \mid 1,0) \right) \left( 1 - \left( \pi(0 \mid 0,0) - \pi(0 \mid 1,0) \right)^{i-2} \right).$$

We can further simplify $G_0$:

$$
\begin{aligned}
G_0 &= \pi(0 \mid 0,0) - \frac{\pi(0 \mid 1,0)}{1 - \pi(0 \mid 0,0) + \pi(0 \mid 1,0)} \\
&= \frac{(1 - \pi(0 \mid 0,0) + \pi(0 \mid 1,0)) (\pi(0 \mid 0,0) - \pi(0 \mid 1,0))}{1 - \pi(0 \mid 0,0) + \pi(0 \mid 1,0)} \\
&= \frac{(\pi(0 \mid 0,0) - \pi(0 \mid 1,0)) (1 - \pi(0 \mid 0,0))}{1 - \pi(0 \mid 0,0) + \pi(0 \mid 1,0)}.
\end{aligned}
$$

By assumption, $\pi(0 \mid 0, 0) - \pi(0 \mid 1, 0) \geqslant \delta'$ and $1 - \pi(0 \mid 0, 0) \geqslant \epsilon'$, thus

$$G_0 \geqslant \epsilon''$$

for some positive constant $\epsilon'$.

Hence, substituting back, we obtain

$$P^3(0 \mid 0, 0) - P^i(0 \mid 0, 0) \geqslant \delta''' \quad \text{(Where } \delta''' \text{ is a positive constant)}$$

Taking expectation over $\pi \sim \mathbb{P}$, we have

$$\mathbb{E}_{\pi \sim \mathbb{P}} \left[ P^3(0 \mid 0, 0) - P^i(0 \mid 0, 0) \right] \geqslant \delta,$$

where $\delta''$ is defined formally as follows:

The constant $\delta$ represents a uniform lower bound on the expected gap between the third-step and $\forall i$-th-step transition probabilities across all Markov chains sampled from the prior distribution $\mathbb{P}$. It is defined as

$$\delta'' = \inf_{\pi \sim \mathbb{P}} \mathbb{E} \left[ P^3(0 \mid 0, 0) - P^i(0 \mid 0, 0) \right],$$

and satisfies $\delta > 0$. This ensures that, even after taking expectations, the separation between transition probabilities remains bounded away from zero across all sampled chains.

Similarly, for the other case, we obtain

$$P^3(1 \mid 1, 1) - P^i(1 \mid 1, 1) \geqslant \delta'''',$$
$$\mathbb{E}_{\pi \sim \mathbb{P}} \left[ P^3(1 \mid 1, 1) - P^i(1 \mid 1, 1) \right] \geqslant \delta.$$

Therefore, using linearity of expectation, we conclude

$$\mathbb{E}_{\pi \sim \mathbb{P}} \left[ P^3(0 \mid 0, 0) + P^3(1 \mid 1, 1) \right] - \mathbb{E}_{\pi \sim \mathbb{P}} \left[ P^k(0 \mid 0, 0) + P^k(1 \mid 1, 1) \right] \geqslant \delta \quad \forall k \geqslant 4.$$

$$\square$$

**Lemma 6** (Monotonicity via spectral decomposition). *Let $(X_t)$ be a reversible, irreducible, aperiodic $k$th-order Markov chain on state-space $S$, and let $\mathcal{A} = S^k$. For integers $i, \ell \geqslant 1$, define the return count*

$$g_{i,\ell} = \sum_{(s_1, \ldots, s_k) \in S^k} \sum_{j=1}^{k} \Pr\left( X_{i+\ell} = s_j \mid X_{i-1} = s_1, \ldots, X_{i-k} = s_k \right).$$

*Equivalently, one can view the chain as a lifted first-order markov chain on $\mathcal{A}$, where $a = (a_1, \ldots, a_k)$, $b = (b_1, \ldots, b_k) \in \mathcal{A}$ and*

$$P(a \to b) = \Pr\left( (X_t, \ldots, X_{t-k+1}) = b \mid (X_{t-1}, \ldots, X_{t-k}) = a \right),$$
$$f(a) = |a| \text{ (where } | \cdot | \text{ denotes the cardinality)},$$
$$g_{i,\ell} = \sum_{a,b \in \mathcal{A}} P^\ell(a \to b) \, f(a).$$

*Under detailed balance with stationary distribution $\pi$. If $P$ is self-adjoint on $\ell^2(\mathcal{A}, \pi)$ with eigenvalues $1 = \lambda_1 > \lambda_2 \geqslant \cdots \geqslant \lambda_N > 0$ and if $\beta_m > 0$ where orthonormal eigenfunctions are defined as $\{\phi_m\}$ and $\beta_m$ is defined as:*

$$\beta_m = \sum_{a,b \in \mathcal{A}} \phi_m(a) \, \phi_m(b) \, \pi(b) \, f(a).$$

*Then for all $\ell \geqslant 0$:*

$$g_{i,\ell} - g_{i,\ell+1} = \sum_{m=2}^{N} \lambda_m^\ell (1 - \lambda_m) \beta_m \geqslant 0,$$

$$g_{i,1} - g_{i,\ell} = \sum_{m=2}^{N} (\lambda_m^2 - \lambda_m^\ell)\beta_m > 0.$$

*Hence,* $g_{i,1} > g_{i,2} > g_{i,3} > \cdots$.

Moreover, $g_{i,1} \geqslant 2^{k-1}$ if $\sum_{m=1}^{N} \lambda_m^2 \beta_m \geqslant 2^{k-1}$

*Proof.* From spectral expansion, we know that:

$$P^\ell(a \to b) = \sum_m \lambda_m^\ell \phi_m(a)\phi_m(b)\pi(b)$$

Therefore, we obtain (on substitution)

$$g_{i,\ell} = \sum_{a,b} P^\ell(a \to b)f(a,b) = \sum_m \lambda_m^{\ell+1}\beta_m$$

Hence, finally we obtain monotonicity, due to the assumptions:

$$g_{i,\ell} - g_{i,\ell+1} = \sum_{m \geqslant 2} \lambda_m^{\ell+1}(1 - \lambda_m)\beta_m \geqslant 0$$

The last condition is obtained by direct substitution.

$\square$

# G    Additional Lemmas

**Lemma 7** (Lemma D.2 in Nichani et al. [20]). *For all $j \geqslant 0$, $j \neq 1$, we have*

$$g_{i,1} \geqslant g_{i,j} + \frac{\gamma^3}{2S}.$$

*Where:*

$$g_{i,j}(\pi) = \sum_{s',s} \frac{\pi(s' \mid s)}{\mu_\pi(s')} \mathbb{P}(x_i = s', x_{i-j} = s) - 1,$$

$$g_{i,j} = \mathbb{E}_{\pi \sim \mathbb{P}(\pi)}\left[g_{i,j}(\pi)\right].$$

**Lemma 8** (Levin and Peres [13]). *Let $(z_t)_{t \geqslant 0}$ be a stationary, reversible, ergodic Markov chain with stationary distribution $\pi$, and let $P$ denote its Markov operator on $L^2(\pi)$. For any function $f : \mathcal{Z} \to \mathbb{R}$ with $\mathbb{E}_\pi[f^2] < \infty$ and all $t, t' \geqslant 0$,*

$$\mathrm{Cov}(f(z_t), f(z_{t'})) = \left\langle \widetilde{f}, P^{|t-t'|}\widetilde{f} \right\rangle_\pi,$$

*where $\widetilde{f} = f - \mathbb{E}_\pi[f]$ and $\langle g, h \rangle_\pi := \mathbb{E}_\pi[g(z)h(z)]$.*

*Proof.* By stationarity and the Markov property,

$$\mathbb{E}[f(z_{t'}) \mid z_t] = (P^{t'-t}f)(z_t), \tag{10}$$

$$\mathbb{E}_\pi[f(z_t)f(z_{t'})] = \langle f, P^{t'-t}f \rangle_\pi. \tag{11}$$

Thus,

$$\mathrm{Cov}(f(z_t), f(z_{t'})) = \langle f, P^{|t-t'|}f \rangle_\pi - \left(\mathbb{E}_\pi[f]\right)^2. \tag{12}$$

Expanding $f = \widetilde{f} + \mathbb{E}_\pi[f]$ and using that $P$ preserves constants,

$$\langle f, P^{|t-t'|}f \rangle_\pi = \langle \widetilde{f}, P^{|t-t'|}\widetilde{f} \rangle_\pi + \left(\mathbb{E}_\pi[f]\right)^2, \tag{13}$$

which gives

$$\mathrm{Cov}(f(z_t), f(z_{t'})) = \langle \widetilde{f}, P^{|t-t'|}\widetilde{f} \rangle_\pi. \tag{14}$$

Applying Cauchy–Schwarz inequality,

$$|\mathrm{Cov}(f(z_t), f(z_{t'}))| \leqslant \|\widetilde{f}\|_{L^2(\pi)} \|P^{|t-t'|}\widetilde{f}\|_{L^2(\pi)}. \tag{15}$$

If the spectral gap satisfies $\|P\|_{\mathrm{op}} \leqslant \lambda$ on centered functions, then

$$\|P^{|t-t'|}\widetilde{f}\|_{L^2(\pi)} \leqslant \lambda^{|t-t'|}\|\widetilde{f}\|_{L^2(\pi)}, \tag{16}$$

thus

$$|\mathrm{Cov}(f(z_t), f(z_{t'}))| \leqslant \lambda^{|t-t'|} \mathrm{Var}_\pi(f). \tag{17}$$

$\square$

**Lemma 9** (Variance Lemma similar to Cor. F.6 in Nichani et al. [20] ). *Suppose:*

1. *$(s_t)_{t \geqslant 0}$ is a stationary, irreducible, aperiodic $k$-th order Markov chain on a finite state space $\mathcal{S}$ of size $S$,*

2. *Define the lifted process $z_t = (s_t, s_{t-1}, \ldots, s_{t-k+1}) \in \mathcal{S}^k$,*

3. *Assume the transition matrix $P$ of $(z_t)$ has second-largest eigenvalue modulus $\lambda < 1$.*

*Then, if*

$$\hat{\mu}_X(s') = \frac{1}{n} \sum_{t=1}^n \mathbb{I}_{z_i = s'},$$

*is the empirical frequency of observing $s'$, we have the variance bound*

$$\mathbb{E}\left[\left(\hat{\mu}_X(s') - \mu_\pi(s')\right)^2\right] \lesssim \frac{1}{n(1-\lambda)},$$

*where:*

- $\mu_\pi(s') = \pi_Z(\{z : z_1 = s'\})$ *is the stationary probability of $s'$,*

- *and the hidden constant is universal (independent of $T$).*

*Proof.* Let $z_n = (x_n, x_{n-1}, \cdots, )$.

Therefore, we can see that:

$$\hat{\mu}_{\pi,X}(s') = \frac{1}{n} \sum_{i=1}^{n} \mathbb{I}_{z_i = s'} \tag{18}$$

Therefore:

$$\text{Var}(\hat{\mu}_{\pi,X}(s')) = \text{Var}\left(\frac{1}{n} \sum_{t=1}^{n} \mathbb{I}_{z_t = s'}\right) \tag{19}$$

$$= \frac{1}{n^2} \sum_{t=1}^{n} \sum_{t'=1}^{n} \text{Cov}\left(\mathbb{I}_{z_t = s'}, \mathbb{I}_{z_{t'} = s'}\right) \tag{20}$$

From Lemma 8, we can see that,

$$\text{Var}(\hat{\mu}_{\pi,X}(s')) \lesssim \frac{1}{n^2} \sum_{t,t'} \lambda^{|t-t'|} \tag{21}$$

$$\lesssim \frac{1}{n(1-\lambda)} \tag{22}$$

$\square$

**Lemma 10** (Extension-1 to $k^{\text{th}}$-order of Lemma G.2 in Nichani et al. [20])**.** *For any $s, s' \in \mathcal{S}$ and any $k^{\text{th}}$-order transition kernel, $\pi$ with spectral gap $1 - \lambda(\pi) \geq 1 - \lambda$, and with $\mu_\pi(s') \geq 0$, and $i > j$ there exists a sufficiently large constant $C_{\gamma,S}$ such that if $\epsilon \geq C_{\gamma,S}(n(1-\lambda))^{-1/2}$, then:*

$$\left| \mathbb{E}_X\left[ \frac{(\mathbb{I}_{x_i=s'} - \hat{\mu}_X(s'))\mathbb{I}_{x_j=s}}{\hat{\mu}_X(s') + \epsilon} \right] - \left( \frac{\mathbb{P}_X[s_i = s', s_j = s]}{\mu_\pi(s')} - \mathbb{P}_X[s_j = s] \right) \right| \lesssim \frac{1}{\sqrt{n(1-\lambda)}}.$$

*Proof.* The proof is the same as the proof of Lemma G.2 in Nichani et al. [20], where to control the variance term, we use 9. For completeness, we desribe the proof below:

Define

$$E_\pi(s, s') := \mathbb{E}_X\left[ \frac{(\mathbb{I}_{x_i=s'} - \hat{\mu}_X(s'))\mathbb{I}_{x_j=s}}{\hat{\mu}_X(s') + \epsilon} \right] - \frac{\mathbb{P}_X[x_i = s', x_j = s]}{\mu_\pi(s')} + \mathbb{P}_X[x_j = s].$$

Expanding:

$$(\mathbb{I}_{x_i=s'} - \hat{\mu}_X(s'))\mathbb{I}_{x_j=s} = \mathbb{I}_{x_i=s'}\mathbb{I}_{x_j=s} - \hat{\mu}_X(s')\mathbb{I}_{x_j=s}.$$

Thus:

$$E_\pi(s, s') = \mathbb{E}_X\left[ \frac{\mathbb{I}_{x_i=s'}\mathbb{I}_{x_j=s}}{\hat{\mu}_X(s') + \epsilon} \right] - \frac{\mathbb{P}_X[x_i = s', x_j = s]}{\mu_\pi(s')} - \mathbb{E}_X\left[ \frac{\hat{\mu}_X(s')\mathbb{I}_{x_j=s}}{\hat{\mu}_X(s') + \epsilon} \right] + \mathbb{P}_X[x_j = s].$$

Grouping terms:

$$E_\pi(s, s') = \mathbb{E}_X\left[ \mathbb{I}_{x_i=s'}\mathbb{I}_{x_j=s}\left( \frac{1}{\hat{\mu}_X(s') + \epsilon} - \frac{1}{\mu_\pi(s')} \right) \right] + \mathbb{E}_X\left[ \mathbb{I}_{x_j=s}\left( 1 - \frac{\hat{\mu}_X(s')}{\hat{\mu}_X(s') + \epsilon} \right) \right].$$

Grouping terms:

$$\frac{1}{\hat{\mu}_X(s') + \epsilon} - \frac{1}{\mu_\pi(s')} \approx \frac{\mu_\pi(s') - \hat{\mu}_X(s') - \epsilon}{(\hat{\mu}_X(s') + \epsilon)\mu_\pi(s')},$$

and

$$1 - \frac{\hat{\mu}_X(s')}{\hat{\mu}_X(s') + \epsilon} = \frac{\epsilon}{\hat{\mu}_X(s') + \epsilon}.$$

Thus by the triangle inequality and boundedness of indicator variables $\mathbb{I}_{x_i=s'}, \mathbb{I}_{x_j=s} \in \{0,1\}$, we get

$$|E_\pi(s,s')| \lesssim \mathbb{E}_X\left[\frac{|\mathbb{I}_{x_i=s'}\mathbb{I}_{x_j=s}||\hat{\mu}_X(s') - \mu_\pi(s')| + \epsilon(|\mathbb{I}_{x_i=s'}\mathbb{I}_{x_j=s}| + \mu_\pi(s')|\mathbb{I}_{x_j=s}|)}{(\hat{\mu}_X(s') + \epsilon)\mu_\pi(s')}\right]$$

$$\lesssim \mathbb{E}_X\left[\frac{|\hat{\mu}_X(s') - \mu_\pi(s')| + \epsilon}{\mu_\pi(s')^2}\right] + \epsilon^{-1}\mathbb{P}_X\left[\hat{\mu}_X(s') \leq \frac{\mu_\pi(s')}{2}\right].$$

Using concentration inequalities (Lemma 9):

$$\mathbb{E}_X\left[(\hat{\mu}_X(s') - \mu_\pi(s'))^2\right] \lesssim \frac{1}{n(1-\lambda)}, \quad \mathbb{P}_X\left[\hat{\mu}_X(s') \leq \frac{\mu_\pi(s')}{2}\right] \lesssim \frac{1}{n(1-\lambda)},$$

and $\epsilon \sim (n(1-\lambda))^{-1/2}$, we conclude:

$$|E_\pi(s,s')| \lesssim \frac{1}{n(1-\lambda)}.$$

$\square$

**Lemma 11** (Extension-2 to k$^{th}$-order of Lemma G.2 in Nichani et al. [20]). *For any $s', s_1, s_2, \cdots s_k \in \mathcal{S}$ and any $k^{th}$-order transition kernel, $\pi$ with spectral gap $1 - \lambda(\pi) \geq 1 - \lambda$, and with $\mu_\pi(s') \geq 0$, and $i > j$, there exists a sufficiently large constant $C_{\gamma,S}$ such that if $\epsilon \geq C_{\gamma,S}(n(1-\lambda))^{-1/2}$, then:*

$$\sum_{l=1}^k \frac{1}{k}\left\{\mathbb{E}_X\left[\frac{(\mathbb{I}_{x_i=s'} - \hat{\mu}_X(s'))\mathbb{I}_{x_j=s_l}}{\hat{\mu}_X(s') + \epsilon}\right] - \left(\frac{\mathbb{P}_X[s_i=s', s_j=s_l]}{\mu_\pi(s')} - \mathbb{P}_X[s_j=s_l]\right)\right\} \lesssim \frac{1}{\sqrt{n(1-\lambda)}}.$$

*Proof.* The proof follows immediatly from Lemma 10. Let:

$$\sum_{l=1}^k \frac{1}{k}\underbrace{\left\{\mathbb{E}_X\left[\frac{(\mathbb{I}_{x_i=s'} - \hat{\mu}_X(s'))\mathbb{I}_{x_j=s_l}}{\hat{\mu}_X(s') + \epsilon}\right] - \left(\frac{\mathbb{P}_X[s_i=s', s_j=s_l]}{\mu_\pi(s')} - \mathbb{P}_X[s_j=s_l]\right)\right\}}_{Q_l} \lesssim \frac{1}{\sqrt{n(1-\lambda)}}.$$

We know from Lemma 10 that $Q_l \lesssim \frac{1}{\sqrt{n(1-\lambda)}}$ and hence it immediatly follows that $\sum_{l=1}^k \frac{1}{k}Q_l \lesssim \frac{1}{\sqrt{n(1-\lambda)}}$.

$\square$

# H   Experiments and Experimental Details

## H.1   Additional Experiments

In this section, we present additional attention map visualizations obtained by training a two-layer, single-head transformer on first, second, and third-order Markov chains, respectively. Attention maps from both the first and second layers are shown. For the first layer, we compute the average attention map over multiple sequences to understand the expected attention pattern. For the second layer, we display the attention map for a specific sequence. Additionally, we include a *Pseudo Attention Map*, which highlights the tokens that would ideally be attended to under a $k$-gram estimator. To assess alignment, we also plot the absolute difference between the actual attention map and the pseudo attention map. Note that for the first layer, we do not compute a pseudo attention map; hence, the pseudo attention and the corresponding absolute difference are not meaningful in that context.

### H.1.1   First-Order Markov Chains

The following plots show the attention maps learned from first-order Markov chains.

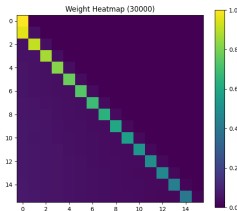

Figure 5: Layer 1: Average attention map computed over multiple sequences.



Figure 6: Layer 2: Attention map corresponding to a randomly sampled sequence (denoted as sequence–1)



Figure 7: Layer 2: Attention map corresponding to a randomly sampled sequence (denoted as sequence–2)



Figure 8: Layer 2: Attention map corresponding to a randomly sampled sequence (denoted as sequence–3)

### H.1.2 Second-Order Markov Chains

The following plots show the attention maps learned from second-order Markov chains.

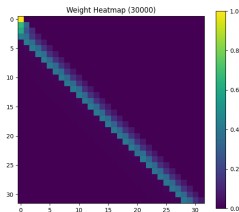

Figure 9: Layer 1: Average attention map computed over multiple sequences.



Figure 10: Layer 2: Attention map corresponding to a randomly sampled sequence (denoted as sequence–1)



Figure 11: Layer 2: Attention map corresponding to a randomly sampled sequence (denoted as sequence–2)



Figure 12: Layer 2: Attention map corresponding to a randomly sampled sequence (denoted as sequence–3)

### H.1.3 Third-Order Markov Chains

The following plots show the attention maps learned from third-order Markov chains.

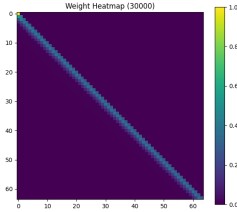

Figure 13: Layer 1: Average attention map computed over multiple sequences.



Figure 14: Layer 2: Attention map corresponding to a randomly sampled sequence (denoted as sequence–1)



Figure 15: Layer 2: Attention map corresponding to a randomly sampled sequence (denoted as sequence–2)



Figure 16: Layer 2: Attention map corresponding to a randomly sampled sequence (denoted as sequence–3)

## H.2 Experiments on One Layer Transformers

To assess whether single-layer transformers can implement induction heads, we trained one-layer models on sequences generated from second-order Markov chains, using the same hyperparameter ranges as in our two-layer baselines.

**Quantitative results.** As shown in Tab. 2, the gap between the true loss (computed from the ground-truth transition probabilities) and the loss achieved by the trained model remains significantly larger for one-layer transformers than for their two-layer counterparts. This difference persists across hyperparameter settings and indicates that deeper models capture the underlying structure more effectively.

| Number of Layers | Excess Cross-Entropy Loss |
|---|---|
| 1 | $0.131 \pm 0.000$ |
| 2 | $0.100 \pm 0.003$ |

Table 2: Excess cross-entropy loss (loss obtained by the model subtracted with the true loss) for one and two-layer transformers trained on sequences from a second-order Markov chain.

**Qualitative results.** Examination of the attention maps (Fig. 17, Fig. 18, Fig. 19) confirms this finding: one-layer transformers converge to a qualitatively different structure than the pseudo-attention map required for a $k$-gram estimator (see Fig. 1b). Specifically, attention mass concentrates along the two to three lower diagonals, rather than following the induction-like pattern.

These results are consistent with prior observations [22] that one-layer transformers are insufficient for representing induction heads, while two-layer models succeed in doing so (albeit on different datasets).



Figure 17: Attention map corresponding to a randomly sampled sequence (denoted as sequence–1)



Figure 18: Attention map corresponding to a randomly sampled sequence (denoted as sequence–2)



Figure 19: Attention map corresponding to a randomly sampled sequence (denoted as sequence–3)



Figure 20: Attention map corresponding to a randomly sampled sequence (denoted as sequence–4)

## H.3 Experiments on Noisy Sequences

We evaluate the robustness of our $k$-gram framework under non-ideal conditions by introducing stochastic noise into synthetic sequences generated from second-order Markov chains. We consider two noise processes: (i) random token substitution and (ii) perturbation of the transition dynamics. Models are two-layer, single-head transformers trained with three random seeds. Performance is reported as the difference between the model's cross-entropy loss and the cross-entropy computed under the true transition probabilities over a large evaluation batch.

### H.3.1 Random Token Substitution

We corrupt sequences at the token level with probability $p_n$. For each token, with probability $p_n$ we replace it with a uniformly sampled alternative; with probability $1 - p_n$ it is left unchanged. The results can be found in Tab. 3. In the noiseless case ($p_n = 0$), the learned second-layer attention map closely aligns with the *pseudo-attention map* (the optimal attention pattern for an ideal $k$-gram estimator). As $p_n$ increases, the alignment degrades and the loss gap widens.

### H.3.2 Perturbation of Transition Dynamics

We next perturb the sequence dynamics directly. Let $P_d$ denote the true transition probability matrix. Before each prediction step we interpolate between $P_d$ and a randomly generated transition matrix $P_r$, defining

$$P = (1 - \alpha) P_d + \alpha P_r, \qquad \alpha \in [0, 1], \tag{23}$$

| $p_n$ | Difference in Cross-Entropy Loss |
|-----|-----|
| 0.0 | $0.10 \pm 0.003$ |
| 0.1 | $0.15 \pm 0.010$ |
| 0.25 | $0.18 \pm 0.002$ |
| 0.5 | $0.21 \pm 0.005$ |

Table 3: Cross-entropy loss difference under token substitution noise. Values are mean $\pm$ std. across three seeds.

| $\alpha$ | Difference in Cross-Entropy Loss |
|-----|-----|
| 0.0 | $0.100 \pm 0.003$ |
| 0.1 | $0.101 \pm 0.003$ |
| 0.25 | $0.141 \pm 0.005$ |
| 0.5 | $0.181 \pm 0.004$ |

Table 4: Cross-entropy loss difference under transition perturbations defined in Eq. (23). Values are mean $\pm$ std. across three seeds.

where a new transition matrix $P_r$ is resampled at every time step. As in the previous setting, we measure the difference between the model's cross-entropy and the true loss. Note that the results can be found in Tab. 4. The loss gap increases with $\alpha$. For $\alpha \leqslant 0.1$, attention maps remain close to the pseudo-attention pattern; at $\alpha = 0.25$ and beyond, alignment deteriorates substantially. The model is more robust to this transition noise than to token-level corruption.

### H.3.3 Discussion

These results complement our theory: when the data-generating process is close to Markovian, the learned attention structure approximates the pseudo-attention map; as noise increases, alignment degrades in tandem with cross-entropy performance. Overall, transformer-based estimators are sensitive to deviations from ideal $k$-gram dependencies, with greater robustness observed for transition perturbations than for token substitution.

### H.4 Experiments on Bit Precision

Our bit-precision results are the same as those of [26], which show that $\Omega(\log T + k)$ bits per parameter suffice for an additive error of $\mathcal{O}(1/T)$, where $T$ is the sequence length and $k$ the Markov order. To validate this, we quantize weights and activations of a model trained on a second-order Markov chain with sequence length 32. We then compute the Frobenius norm between the pseudo (optimal) attention map and the second-layer attention map of the quantized model across 10 random sequences (each from a different Markov transition kernel), and report the mean and variance:

| Quantization (Bits) | Frobenius Norm (Mean $\pm$ Std) |
|-----|-----|
| 2 | $4.6108 \pm 0.7190$ |
| 4 | $4.5842 \pm 0.9826$ |
| 8 | $3.8615 \pm 0.4590$ |
| 32 (No quantization) | $3.7700 \pm 0.4725$ |

These results suggest that 8-bit quantization preserves the attention structure (as measured by the Frobenius norm) almost as well as full precision (32-bit), and aligns with the theoretical bounds.

### H.5 Experimental Details

The table below provides details of the hyperparameters used across all experiments in the paper. Each experiment was conducted on a single NVIDIA A100 GPU, with runtimes ranging from 30 minutes to 2 hours.

| | |
|---|---|
| Dataset | $k$-th order binary Markov source |
| Architecture | Based on the GPT-2 architecture as implemented in [24] |
| Batch size | Grid-searched in $\{16, 32, 64\}$ |
| Accumulation steps | 1 |
| Optimizer | AdamW ($\beta_1 \in \{0.85, 0.9\}, \beta_2 \in \{0.9, 0.95\}$) |
| Learning rate | Grid-searched in $\{0.0001, 0.001, 0.01\}$ |
| Scheduler | Cosine |
| # Iterations | 30000 |
| Weight decay | Grid-searched in $\{0, 10^{-4}, 10^{-3}\}$ |
| Dropout | 0 or 0.1 |
| Sequence length | 8, 32, 64 |
| Embedding dimension | Grid-searched in $\{32, 64, 128\}$ |
| Transformer layers | 2 |
| Attention heads | 1 or 2 depending on the experiment |
| Repetitions | 5 |

Table 5: Settings and parameters for the transformer model used in the experiments.

# I Parameter Breakdown and Comparison

In this section, we present a comprehensive comparison of parameter counts that explicitly accounts for all components of the models under consideration. Our analysis shows that our construction achieves greater parameter efficiency than the design of Rajaraman et al. [26] and also improves upon the approach proposed by Nichani et al. [20]. To make this concrete, we provide a full parameter breakdown for both our model and that of Rajaraman et al. [26], detailing the contributions of embeddings, attention layers, feed-forward modules, and output projection.

## Our Construction

The total parameter count of our construction is: $9(6S + 3)^2 + (6S + 3)(2T + 2S + 9)$

| Component | Parameters |
|---|---|
| Token Embedding | $(6S + 3) \times S$ |
| **Attention Layer 1** | |
| Positional Encoding | $(6S + 3) \times T$ |
| Query, Key, Value Projections | $3(6S + 3)^2$ |
| **MLP — (Three Layers)** | |
| Weights | $3(6S + 3)^2$ |
| Biases | $3(6S + 3)$ |
| Layer Norms | $6(6S + 3)$ |
| **Attention Layer 2** | |
| Positional Encoding | $(6S + 3) \times T$ |
| Query, Key, Value Projections | $3(6S + 3)^2$ |
| Output Projection | $(6S + 3) \times S$ |
| **Total** | $9(6S + 3)^2 + (6S + 3)(2T + 2S + 9)$ |

Table 6: Parameter breakdown for our construction.

## Construction in Rajaraman et al. [26]

The total parameter count of the construction in Rajaraman et al. [26] is: $15(6S + 3)^2 + (6S + 3)(3T + 2S + 10)$

| Component | Parameters |
|---|---|
| Token Embedding | $(6S + 3) \times S$ |
| **Transformer Layer 1** | |
| Positional Encoding | $(6S + 3) \times T$ |
| Query, Key, Value Projections | $3(6S + 3)^2$ |
| MLP (2 layers) Weights | $2(6S + 3)^2$ |
| MLP (2 layers) Biases | $2(6S + 3)$ |
| Layer Norms | $2(6S + 3)$ |
| **Transformer Layer 2** | |
| *Same as Transformer Layer 1:* $(6S + 3) \times T + 5(6S + 3)^2 + 4(6S + 3)$ | |
| **Transformer Layer 3** | |
| Positional Encoding | $(6S + 3) \times T$ |
| Query, Key, Value Projections | $3(6S + 3)^2$ |
| MLP (2 layers) Weights | $2(6S + 3)^2$ |
| MLP (2 layers) Biases | $2(6S + 3)$ |
| Layer Norms | $2(6S + 3)$ |
| Output Projection | $(6S + 3) \times S$ |
| **Total** | $15(6S + 3)^2 + (6S + 3)(3T + 2S + 10)$ |

Table 7: Parameter breakdown for the construction in Rajaraman et al. [26].

