# OpenReview forum: "What One Cannot, Two Can: Two-Layer Transformers Provably Represent Induction Heads on Any-Order Markov Chains"
_NeurIPS.cc/2025/Conference — NeurIPS 2025 spotlight_

### Official Review · Reviewer_VB7N · 2025-06-27

**Clarity:** 4
**Significance:** 2
**Originality:** 2
**Rating:** 3
**Confidence:** 4

**Summary:**

This is a theory paper on multi-layer Transformer models. The paper studies inductions heads for the conditional k-grams problem: given a sequence of k tokens, can the model predict the next token correctly?  This problem is modeled as a Markov predictions in a k-gram and has connections with k-th order Markov chains and In Context Learning.
Authors establish that a 2-layer Transformer model is sufficient for building an induction head for this problem, regardless of the value of k. This is an improvement on previously known result which had established that 3 layers are sufficient. Authors also share a result on the dynamics of gradient descent for converging to the solution.

**Questions:**

1. optimization: strong assumptions are made. What happens if this is not the case that all layers have their optimal values? how far are we from this setup when data is noisy?

2. are 2 layers the fewest needed layers? how hard is it to establish a lower-bound?

**Ethical Concerns:**

["NO or VERY MINOR ethics concerns only"]

**Limitations:**

more numerical experiments on noisy / real data could help bridge the gap between theory results and practical architecture selection

**Quality:**

2

**Strengths And Weaknesses:**

Strengths:

1. the paper is well written and easy to follow
2. the result is well exposed: reduction of 3 to 2 layers
3. literature review is complete and nicely exposed

Weaknesses

1. optimization result makes very strong assumptions. I understand that the problem is really hard and unless we make strong assumptions it is not possible to prove anything useful. However we also need to understand the behavior outside the nice regime proposed here. Authors are assuming all layers but one already have the optimal value.

2. lack of lower bound for number of layers

3. optimization is disconnected from realistic situations because very strong assumptions are needed.

4. limited numerical experiments: gap between theory and practice

---

> ### Author Rebuttal · Authors · 2025-07-30
>
> We would like to sincerely thank the reviewer for their thoughtful feedback and insightful comments. We address the individual comments and questions below.
>
> ---
>
> ## Comments about weaknesses
>
> **W1.  optimization result makes very strong assumptions. I understand that the problem is really hard and unless we make strong assumptions it is not possible to prove anything useful. However we also need to understand the behavior outside the nice regime proposed here. Authors are assuming all layers but one already have the optimal value.**
>
> We believe this question can be interpreted in two ways: one theoretical and the other practical. We address both perspectives separately.
>
> *Theoretical assumptions*: Prior work has shown that full optimization analysis in this setting remains a challenging problem. Nichani et. al. [2], for example, provides such analysis only for first-order Markov chains and a specialized disentangled transformer. Rajaraman et. al. [1] offers a representation result for more realistic architectures (including MLPs and LayerNorms), but does not address optimization. In contrast, our work presents the tightest known construction relating transformer depth to Markov order under realistic architectural assumptions and initiates an optimization analysis in this setting. It is also worth noting that Nichani et. al. [2] makes comparable assumptions by analyzing optimization on a reduced model of a disentangled transformer, where the second attention layer relies on a single scalar, an assumption similar to ours. We agree with the reviewer that a full optimization analysis remains an important open question. Our primary focus was on the representational results, while also making partial progress on learning dynamics. A complete characterization is left for future work.
>
> *Practical Experiments:* We would like to emphasize that all experiments in our paper were conducted using a two-layer, single-head transformer model with standard initialization schemes. No component was assumed to be optimal. Further details are provided in our responses to Weaknesses 3 and 4.
>
> **W2. lack of lower bound for number of layers**
>
> We would like to clarify that, as discussed in lines 39–47 of the paper, Sanford et al. [3] have provided a formal lower bound in their work titled **"One-layer transformers fail to solve the induction heads task."** Using a communication complexity argument, they demonstrate that no one-layer transformer can solve the induction heads task unless its size is exponentially larger than that of a two-layer transformer capable of solving the task. This result establishes a meaningful lower bound on the number of layers required.
>
> **W3. optimization is disconnected from realistic situations because very strong assumptions are needed.**
>
> We kindly refer the reviewer to our response to the first weakness for additional context. Importantly, all experimental results presented in the paper (Fig. 2 and Figs. 5–16) were obtained using standard initialization schemes, without assuming any component to be optimal. These experiments show that, under standard initialization, a two-layer, single-head transformer reliably converges to a $k$-gram estimator. Moreover, our theoretical construction aligns closely with these empirical findings, as detailed in lines 205–209 of the paper.
>
> **W4. limited numerical experiments: gap between theory and practice**
>
> We would like to emphasize that all experimental results in the paper (Fig. 2 and Figs. 5–16) were obtained using standard initialization schemes, without assuming any component to be optimal. These findings demonstrate that, under such initialization, a two-layer, single-head transformer converges to a $k$-gram estimator. Additionally, our theoretical construction closely aligns with the empirical observations. We also perform experiments on noisy data and we present our results below (please see the reply to question -1).
>
> ---
>
> ## Questions
>
> **Q1. optimization: strong assumptions are made. What happens if this is not the case that all layers have their optimal values? how far are we from this setup when data is noisy?**
>
> We believe this question can be interpreted in two ways: one theoretical and the other practical. We believe that we have attempted to address both aspects in our responses to Weaknesses 1, 3, and 4. Additionally, below we provide experimental results for our model trained on noisy data.
>
> **Noisy Data:** To evaluate robustness to noise, we sampled sequences from second-order Markov chains and introduced noise with probability $p_n$: for each token, with probability $p_n$, we replaced it with a random token; otherwise, it was left unchanged. Each model was run three times, and we compared their cross-entropy loss to the cross-entropy computed using the true transition probabilities over a large batch. We observe that when $p_n = 0$ (no noise), the ideal attention map closely aligns with the learned attention map in the second layer. However, as $p_n$ increases, this alignment degrades. Notably, even at $p_n = 0.1$, the pseudo attention map already starts to differ significantly. Additionally, the gap between the model's cross-entropy loss and the cross entropy loss computed using the true transition probabilities steadily increases with higher noise levels:
>
> | $p_n$ | Difference in Cross-Entropy Loss|
> |-|-|
> | 0| $0.10 \pm 0.003$|
> | 0.1| $0.15 \pm 0.010$ |
> | 0.25| $0.18 \pm 0.002$|
> | 0.5|$0.21 \pm 0.005$|
>
> We emphasize that our theoretical construction is designed for estimating $k$-th order Markov chains in the absence of noise. We agree with the reviewer that understanding the impact of noise on such constructions is an interesting direction, and we leave a thorough theoretical treatment under noisy conditions to future work. We will add these results to the revised version of the paper.
>
>
> **Q2. are 2 layers the fewest needed layers? how hard is it to establish a lower-bound?**
>
> As described in the response to Weakness-2, Sanford et. al. [3] already establishes a lower bound. Therefore, we believe that two layers are the minimal number required.
>
> ---
>
> **References:**
>
> [1] Rajaraman, Nived, et al. "Transformers on markov data: Constant depth suffices." NeurIPS 37
>
> [2] Nichani, Eshaan et. al. "How transformers learn causal structure with gradient descent."ICML 2024.
>
> [3] Sanford, Clayton, et. al. "One-layer transformers fail to solve the induction heads task." arXiv preprint arXiv:2408.14332 (2024).

---

> ### Author Response · Authors · 2025-08-04
>
> We thank the reviewer for their constructive comments. Below we have some more experiments and clarifications. Please let us know if there are any further questions.
>
> ---
>
> **Q1. (Followup): optimization: strong assumptions are made. What happens if this is not the case that all layers have their optimal values? how far are we from this setup when data is noisy?**
>
> We conducted an additional experiment to analyze the effects of a different type of noise. Specifically, we sampled sequences from second-order Markov chains, where, prior to sampling, we interpolated between the true transition probability matrix and a randomly generated one. Let $P_d$ denote the transition matrix corresponding to the true dynamics for a given sequence. At each prediction step, we use a perturbed matrix defined as $P = (1 - \alpha) P_d + \alpha P_r$, where $P_r$ is a randomly sampled transition matrix. Importantly, $P_r$ is resampled at every time step, introducing stochastic perturbations before each token is generated. We evaluate performance across various values of $\alpha$. As in prior experiments, we report the difference between the cross-entropy loss of the model (a two-layer, single-head transformer with standard initialization, assuming no prior optimization) and the loss computed using the true transition probabilities over a large evaluation batch. We find that this loss difference increases with larger values of $\alpha$, reflecting greater degradation in model performance due to increased randomness in the sequence dynamics. Furthermore, for $\alpha = 0$ and $\alpha = 0.1$, the second-layer attention maps remain closely aligned with the pseudo-attention map (i.e., the optimal attention pattern for a $k$-gram estimator). In contrast, for $\alpha = 0.25$, the alignment is significantly degraded—this is also reflected in the corresponding increase in cross-entropy loss. (As in the previous experiment, results are averaged over three independent random seeds.) Notably, the model demonstrates greater robustness to this form of noise compared to the perturbations used in earlier experiments.
>
> | α | Difference in Cross Entropy Loss |
> |---|----------------------------------|
> | 0.0 | $0.100 ± 0.003$ |
> | 0.1 | $0.101 ± 0.003$ |
> | 0.25 | $0.141 ± 0.005$ |
> | 0.5 | $0.181 ± 0.004$ |
>
> We would like to reiterate that our theoretical construction is specifically designed for estimating $k$-th order Markov chains in the absence of noise. We agree with the reviewer that understanding the impact of noise on such constructions is an important and interesting direction. A thorough theoretical analysis under noisy conditions is beyond the current scope and is left for future work. We will include the new experimental results discussed above in the revised version of the paper.
>
>
> **Q2. (Followup):  are 2 layers the fewest needed layers? how hard is it to establish a lower-bound?**
>
> Following up on our previous response, we also conducted experiments to demonstrate that one-layer transformers fail to represent induction heads. Specifically, we trained one-layer transformers on sequences sampled from second-order Markov chains, using a wide range of hyperparameters (identical to those used for training the two-layer models). We observed that the difference between the model’s cross-entropy loss and the loss computed using the true transition probabilities over a large evaluation batch is significantly higher for the one-layer transformers compared to their two-layer counterparts. Notably, inspection of the attention maps reveals a clear deviation from the pseudo-attention map (i.e., the optimal attention pattern for a $k$-gram estimator). In fact, the one-layer transformer consistently converges to a distinct attention pattern: for the lowest-loss configurations, the attention is concentrated on the lower two or three diagonals. This is qualitatively different from the pattern required for a $k$-gram estimator and indicates that the model fails to implement induction-like behavior. We also note that similar observations have been made in prior work [1], which showed that one-layer transformers fail to represent induction heads, whereas two-layer transformers are capable of doing so (albeit on a different dataset).
>
>
> | Number of Layers | Difference in Cross Entropy Loss |
> |---|----------------------------------|
> | 1 | $0.131 ± 0.000$ |
> | 2 | $0.100 ± 0.003$ |
>
> Taken together, our experimental findings and prior theoretical results support the conclusion that two-layer transformers represent the minimal architectural requirement for solving this task effectively. We will include the new experimental results discussed in the revised version of the paper.
>
> ---
>
> **References**
>
> [1] Olsson, Catherine, Nelson Elhage, Neel Nanda, Nicholas Joseph, Nova DasSarma, Tom Henighan, Ben Mann et al. "In-context learning and induction heads." arXiv preprint arXiv:2209.11895 (2022).

---

> ### Author Response · Authors · 2025-08-05
> **Discussion period closing soon**
>
> Dear Reviewer VB7N,
>
> We sincerely appreciate the time you have taken to provide valuable feedback for our work. As we are getting closer to the end of the discussion period, could you let us know if our responses above have adequately addressed your concerns? We remain at your disposal for any further questions.
>
> If you agree that our responses to your reviews have addressed the concerns you listed, we kindly ask that you consider whether raising your score would more accurately reflect your updated evaluation of our paper. Thank you again for your time and thoughtful comments!
>
> Sincerely, The Authors

---

> > ### Author Response · Authors · 2025-08-07
> > **Discussion Window Closing – Final Check-In**
> >
> > Dear Reviewer VB7N,
> >
> > We appreciate the time you've taken to provide feedback on our work. As the discussion period comes to a close, we would value your thoughts on whether our responses have addressed your concerns. We remain available to clarify any remaining points if needed.
> >
> > If you feel that our rebuttal has addressed the issues raised in your initial review, we respectfully ask that you consider whether an updated score would better reflect your current assessment of the paper.
> >
> > Thank you again for your time.
> >
> > Sincerely,
> >
> > *The Authors*

---

> ### Comment · Area_Chair_JVKk · 2025-08-07
>
> Dear reviewer GjjU,
>
> This is a kind reminder that reviewers should participate in discussions with authors. Please provide explanations about your updated (or not-updated) score and the “Mandatory Acknowledgement”. Please try to engage with the authors based on their rebuttal before the deadline. Your feedback is essential for the decision on this paper.
>
> Thank you!
> AC

---

### Official Review · Reviewer_GjjU · 2025-06-30

**Clarity:** 3
**Significance:** 2
**Originality:** 3
**Rating:** 4
**Confidence:** 4

**Summary:**

The paper shows that a two-layer transformer with a single attention head per layer can represent the conditional k-gram model for arbitrary Markov order k (Thm 4.1) by using the universal approximation of MLP. It further provides
- (i) a depth-width trade-off construction with two heads in the first layer (Thm 3.2)
-  (ii) a gradient-descent analysis showing that a simplified two-layer network learns first-order induction heads (Thm 5.1).

Empirical evidence is limited to visualizing attention heatmaps on synthetic data (Fig. 2).


While the authors successfully construct a representation of the  $k$-gram model using a two-layer Transformer with a single attention head per layer, the construction heavily relies on the universal approximation capability of the MLP, which likely introduces a substantial number of parameters. This reliance renders the comparison with alternative architectures—such as deeper models or those with more attention heads but less dependence on the MLP—potentially unfair. Addressing these empirical gaps, along with providing a fair and comprehensive comparison of the total parameter counts across models, would substantially enhance the contribution.

**Questions:**

- Could the authors provide a simplified explanation of the key technical challenges that arise when extending the optimization analysis to the case $k>1$.
- The reviewer is curious whether the hand-crafted attention pattern proposed in the construction actually emerges in practice. Could the authors present attention scores from real-world tasks to examine whether such patterns are observed empirically?

**Ethical Concerns:**

["NO or VERY MINOR ethics concerns only"]

**Final Justification:**

The authors have addressed some of my concerns. As a result, I am raising my score to 4.

**Limitations:**

yes

**Quality:**

3

**Strengths And Weaknesses:**

**Strengths**

-  Prior work has demonstrated that representing the conditional $k$-gram model typically requires either a 3-layer Transformer with a single attention head per layer or a 2-layer Transformer with two heads per layer. The proposed new construction only use a 2-layer Transformer with a single attention head per layer. Therefore, they answer the question "can a 2-layer, 1-head transformer represent the conditional k-gram" successfully, bridging the gap between $1$-layer and $3$-layer.
- The paper constructs the representation using the non-non-linearities of MLP. The proof sketch highlights how ReLU + LayerNorm can simulate an additional “virtual head”, a perspective likely useful beyond this setting.


**Weakness**

-  The proposed construction necessitates that the MLP in the first layer effectively encodes the entire one-hot vector within its weight matrix, which likely entails a substantial number of parameters. However, the authors appear to account only for the parameter counts in the embedding layer and the query, key, and value matrices, while omitting those in the MLP. As a result, the parameter comparison with alternative architectures may be misleading. For a fair and comprehensive evaluation, the total parameter count should include all components—namely, the embedding layer, attention mechanism, LayerNorm, and MLP—for each architecture under consideration.
-  The assumptions underlying the gradient descent analysis appear highly artificial—for instance, Assumption (iv) involves a carefully designed initialization coupled with frozen, "optimal" MLP blocks. Consequently, it remains unclear whether standard stochastic gradient descent applied to a genuine two-layer, single-head transformer would converge to the proposed representation of the $k$-gram model. To substantiate the theoretical claims, it is recommended to conduct empirical experiments using the actual two-layer single-head architecture and to examine whether the trained MLP replicates the functional behavior of encoding the entire one-hot vector, as prescribed in the hand-crafted construction.
- The paper lacks empirical comparisons between the two-layer single-head architecture and the three-layer single-head counterpart on realistic datasets such as natural language, music, or other real-world sequences that can be effectively modeled by the $k$-gram framework.

---

> ### Author Rebuttal · Authors · 2025-07-30
>
> We thank the reviewer for the helpful feedback and insightful comments. We address the individual comments and questions below. We respectfully believe that there maybe some misunderstandings about the contributions of our work and also try to clarify them.
>
> ---
> ## Clarifications on potential misunderstandings
>
> **C1. The paper shows ... universal approximation of MLP. (Universal approximation of MLP)**
>
> We would like to clarify that **our representation result does not rely on the universal approximation theorem of an MLP**. Instead, we provide an **explicit and constructive design** that uses an MLP of depth 3, where each linear layer is a finite-dimensional matrix of size $(6S + 3) \times (6S + 3)$, with $S$ denoting the vocabulary size. We include a high-level sketch of the MLP's function here (please see Section 4, lines 210–222 in the paper, and the appendix for full details). The first attention layer computes a geometrically weighted average over the previous $k$ tokens, yielding a vector $v_n$. The MLP then reconstructs a second vector $u_n$, resembling a hard attention over the current and preceding $k-1$ tokens, without using an additional attention head. This is achieved via careful weight encoding, bias selection, ReLU activations, and normalization to algebraically extract and combine embeddings (notably $x_n$, $x_{n-k}$, and $v_n$) using orthogonal subspaces and skip connections.
>
> **C2. While the authors ... would substantially enhance the contribution. (Parameter Count)**
>
> We would like to reiterate that our representation results **do not rely on the universal approximation theorem**. Instead, we provide an explicit construction that uses a 3-layer MLP, where each linear layer is a finite-dimensional matrix of size $(6S + 3) \times (6S + 3)$, with $S$ denoting the vocabulary size. Importantly, the **parameter count reported in Table 1 in the paper includes the full MLP used in our construction**. This ensures a fair and accurate comparison with prior works. We would also like to point out that Rajaraman et al. [1] use three separate 2-layer MLPs, each with linear layers of the same dimensions ($(6S + 3) \times (6S + 3)$). This design choice is similarly reflected in the parameter count for their model as shown in Table-1 in the paper. We would like to note that the parameter counts in the paper (Table-1) account only for the most important components in both our construction and that of Rajaraman et al. [1], namely the query, key, and value layers across different attention layers, the weights of the MLPs, and the relevant positional encodings. Below, we provide an explicit comparison that takes all components into account. We show that our model is not only **still more parameter-efficient than Rajaraman et al. [1], but the margin is even larger than initially reported**. Note that even in this setting, our construction is still more efficient when compared to Nichani et al. [2] as well. To clearly illustrate this, we now provide the full parameter breakdown for both our method and Rajaraman et al. [1], along with a comparative table summarizing prior constructions for a $k$-th order Markov chain. Here, $S$ denotes the vocabulary size and $T$ the sequence length.
>
> **Our Construction (Total: $9(6S + 3)^2 + (6S + 3)(2T + 2S + 9)$)**
>
> 1. Token Embedding: $(6S + 3) \times S $
> 2. Attention Layer 1:
>    a. Positional Encoding: $(6S + 3) \times T$
>    b. Q, K, V: $3(6S + 3)^2$
> 3. MLP (after Attn 1):
>    a. Weights: $3(6S + 3)^2$
>    b. Biases: $3(6S + 3)$
>    c. Layer Norms: $6(6S + 3)$
> 4. Attention Layer 2:
>    a. Positional Encoding: $(6S + 3) \times T$
>    b. Q, K, V: $3(6S + 3)^2$
> 5. Output Projection: $(6S + 3) \times S$
>
> **Rajaraman et al. [1] Construction (Total: $15(6S + 3)^2 + (6S + 3)(3T + 2S + 10)$**
>
> 1. Token Embedding: $(6S + 3) \times S$
> 2. Transformer Layer 1:
>    a. Positional Encoding: $(6S + 3) \times T$
>    b. Q, K, V: $3(6S + 3)^2$
>    c. MLP: $2(6S + 3)^2 + 2(6S + 3)$
>    d. Layer Norm: $2(6S + 3)$
> 3. Transformer Layer 2: (same as Layer 1)
> 4. Transformer Layer 3:
>    a. Positional Encoding: $(6S + 3) \times T$
>    b. Q, K, V: $3(6S + 3)^2$
>    c. MLP: $2(6S + 3)^2 + 2(6S + 3)$
> 5. Output Projection: $(6S + 3) \times S$
>
> Hence finally the table for all the kth order construction is:
>
>
> | **Previous works**| **Markov order** | **Architecture**| **# Parameters**|
> |-|-|-|-
> | Nichani et al. (2024) [2] | $k$| $2$-layer, $k$-head| $(k+(k+1)^2)(S+T)^2 + T(S+T)$|
> | Rajaraman et al. (2024) [1] | $k$| $3$-layer, $1$-head| $15(6S + 3)^2 +  (6S + 3)(3T + 2S +10)$|
> | **Ours**|$k$| $2$-layer, $1$-head| $9(6S + 3)^2 + (6S + 3)(2T + 2S + 9)$|
>
> We will include these results in the revised version of the paper.
>
> ---
> ## Comments about weaknesses
>
> **W1. The proposed construction...for each architecture under consideration.**
>
> Please refer to the second clarification.
>
> **W2. The assumptions underlying  ... in the hand-crafted construction..**
>
> We believe there may have been a misunderstanding. **All experimental results in the paper (Fig. 2 and Figs. 5–16) were obtained using standard initialization schemes, without assuming any component to be optimal**. These results demonstrate that, under standard initialization, a two-layer, single-head transformer converges to a $k$-gram estimator. Our theoretical construction is consistent with the empirical results, as described in lines 205 to 209 of the paper. Given the highly non-convex nature of the problem, there may be multiple valid constructions showing how two-layer, single-head transformers can represent induction heads. Our approach is one such construction. To further support this, we trained two models: (1) a standard two-layer, single-head transformer, and (2) a modified version where the vocabulary is embedded into the MLP, in line with our theoretical construction, and data generated from second order markov chains. We ran each model three times and computed the difference of their cross-entropy loss to the cross-entropy obtained using the true transition probabilities over a large data batch. The results are closely aligned. Additionally, the attention maps for both models  of the second layer closely match the ideal attention patterns of a $k$-gram estimator. We will include these results in the final version of the paper. We would also like to emphasize that while our construction demonstrates a specific way in which induction heads can be represented, it is not obvious that gradient descent will necessarily find this structure. Given the problem's non-convexity, multiple solutions are possible. Our main contribution is a representation result, and the fact that gradient descent empirically converges to a solution that closely matches our construction suggests that the relevant structure can emerge naturally during training.
>
> | Embedded Vocabulary     | Full Model (Standard Initialization) | Difference in Mean |
> |-|-|-|
> | 0.1004 ± 0.00128| 0.100223 ± 0.00110| 0.00017|
>
> **W3. The paper lacks empirical comparisons ..  real-world sequences that can be effectively modeled by the k-gram framework.**
>
> We would like to note that our work is primarily theoretical in nature, with the goal of obtaining the tightest possible characterization of the relationship between transformer depth and Markov order in the context of in-context learning for Markov processes. We acknowledge that the low-depth settings we study may not directly reflect practical applications, as also suggested in prior work ([3]), where larger depths are required to solve realistic tasks. Nonetheless, we believe that these simplified settings provide valuable insights into the fundamental mechanisms by which transformers can implement learning algorithms such as $k$-gram estimation. Extending these results to more realistic settings is an important direction for future work.
>
> ---
>
> ## Questions
>
> **Q1. Could the authors provide ... optimization analysis to the case $k>1$?**
>
> As noted in lines 39–47 in the paper, prior works have also found optimization analysis for $k > 1$ to be challenging. [2] provides such analysis only for first-order Markov chains and on a specialized disentangled transformer. [1] gives a representation result for realistic architectures (with MLPs and LayerNorms), but no optimization analysis. Our work provides the tightest known construction relating transformer depth to Markov order under realistic assumptions and also initiates an optimization analysis in this setting. As Reviewer VB7N also observed, the problem is inherently difficult and progress requires reasonable assumptions. The main obstacles in extending to $k > 1$ is that the MLP contains layer norms and also that MLP's optimality depends on $v_n$ (the output from the first layer) . Due to this, the gradient descent now involves two things, the first being the coupled relationship of the first layer and the MLP and the second being performing optimization analysis with layer norms (which is shown to be difficult to do in [4]). Together, due to the coupled behavior as well as dependency on layer norms, makes the analysis challenging. While a full analysis for $k > 1$ may be possible, we leave this important direction to future work.
>
> **Q2. The reviewer ...  empirically?**
>
> Due to space constraints, we respectfully refer you to our responses to Questions 1 and 2 from Reviewer 2Q6Z, where we have tried to address a closely related question in detail and we hope our prior responses may also be helpful in this context. However, we would be happy to elaborate further if additional clarification is needed.
>
> ---
> **References**:
>
> [1] Rajaraman, Nived, et al. "Transformers on markov data: Constant depth suffices." NeurIPS 37
>
> [2] Nichani, Eshaan et. al. "How transformers learn causal structure with gradient descent."ICML 2024.
>
> [3] Zhao, Wayne Xin, et al. "A survey of large language models." (2023).
>
> [4] Xu, Jingjing, et al. "Understanding and improving layer normalization." NeurIPS 32.

---

> > ### Author Response · Authors · 2025-08-04
> >
> > We thank the reviewer for their constructive comments. Below we have some more experiments and clarifications. Please let us know if there are any further questions.
> >
> > ---
> >
> > **W3 (Followup): The paper lacks empirical comparisons .. real-world sequences that can be effectively modeled by the k-gram framework.**
> >
> > As suggested by Reviewer VB7N, we conducted additional experiments on noisy data that may more closely resemble real-world conditions. We introduced noise in two distinct ways, described below:
> >
> > We sampled sequences from second-order Markov chains and introduced noise with probability $p_n$: for each token, with probability $p_n$, we replaced it with a randomly selected token; otherwise, it was left unchanged. Each model (a two-layer, single-head transformer) was trained with three different seeds, and we compared its cross-entropy loss to the loss computed using the true transition probabilities over a large evaluation batch. We observe that in the noiseless setting ($p_n = 0$), the learned attention map in the second layer closely aligns with the ideal pseudo-attention map (i.e., the optimal attention pattern for a $k$-gram estimator). However, as $p_n$ increases, this alignment progressively degrades. Notably, even at $p_n = 0.1$, the pseudo-attention map begins to diverge significantly from the learned pattern. This degradation is also reflected in the model's performance: the gap between the model’s cross-entropy loss and the true loss steadily increases with higher noise levels.
> >
> >
> > | $p_n$ | Difference in Cross-Entropy Loss |
> > |-------|----------------------------------|
> > | 0     | $0.10 \pm 0.003$                 |
> > | 0.1   | $0.15 \pm 0.010$                 |
> > | 0.25  | $0.18 \pm 0.002$                 |
> > | 0.5   | $0.21 \pm 0.005$                 |
> >
> >
> > We also sampled sequences from second-order Markov chains, where prior to sampling, we interpolated between the true transition probability matrix and a randomly generated one. Let $P_d$ denote the transition matrix corresponding to the true sequence dynamics for a given sequence. At each prediction step, we used a perturbed matrix defined as $P = (1 - \alpha) P_d + \alpha P_r$, where $P_r$ is a randomly sampled transition matrix. Crucially, $P_r$ is resampled at every time step, introducing stochastic perturbations into the sequence dynamics. We report results for various values of $\alpha$. As in earlier experiments, we measure the difference between the cross-entropy loss of the model (a two-layer, single-head transformer) and the loss computed using the true transition probabilities over a large evaluation batch. We find that this loss difference increases with higher $\alpha$, reflecting degraded model performance under increased noise. Furthermore, when $\alpha = 0$ and $\alpha = 0.1$, the attention map in the second layer closely aligns with the pseudo-attention map (i.e., the optimal attention pattern for a $k$-gram estimator). In contrast, at $\alpha = 0.25$, this alignment is substantially degraded—corresponding to a noticeable increase in cross-entropy loss. (As with previous experiments, results are averaged over three random seeds.) Interestingly, the model shows greater robustness to this form of noise compared to the perturbations examined in earlier experiments.
> >
> > | α | Difference in Cross Entropy Loss |
> > |---|----------------------------------|
> > | 0.0 | $0.100 ± 0.003$ |
> > | 0.1 | $0.101 ± 0.003$ |
> > | 0.25 | $0.141 ± 0.005$ |
> > | 0.5 | $0.181 ± 0.004$ |
> >
> >
> > We would like to emphasize that our work is primarily theoretical in nature, aiming to provide the tightest possible characterization of the relationship between transformer depth and Markov order in the context of in-context learning for Markov processes. Additionally, we would like to clarify that the three-layer construction referenced in our work is not our own, but was introduced in [1]. We also note that similar to ours, [1] is a theoretical work and does not include experiments on real-world data. We will include the new experimental results discussed in the revised version of the paper.
> >
> > ---
> >
> > **References**
> >
> > [1] Rajaraman, Nived, et al. "Transformers on markov data: Constant depth suffices." Advances in Neural Information Processing Systems 37 (2024): 137521-137556.

---

> ### Author Response · Authors · 2025-08-05
> **Discussion period closing soon**
>
> Dear Reviewer GjjU,
>
> We sincerely appreciate the time you have taken to provide valuable feedback for our work. As we are getting closer to the end of the discussion period, could you let us know if our responses above have adequately addressed your concerns? We remain at your disposal for any further questions.
>
> If you agree that our responses to your reviews have addressed the concerns you listed, we kindly ask if raising your score would more accurately reflect your updated evaluation of our paper. Thank you again for your time and thoughtful comments!
>
> Sincerely, The Authors

---

> > ### Author Response · Authors · 2025-08-07
> > **Discussion Window Closing – Final Check-In**
> >
> > Dear Reviewer GjjU,
> >
> > We would like to clarify a minor typo in our response to Q1: the correct references are lines 39–47 and 240–264 in the paper (not just 39–47 as originally stated). Additionally, we believe our response to W1 from Reviewer VB7N offers relevant context for this question, and we respectfully refer you to that reply for further clarification.
> >
> > We appreciate the time you've taken to provide feedback on our work. As the discussion period comes to a close, we would value your thoughts on whether our responses have addressed your concerns. We remain available to clarify any remaining points if needed.
> >
> > If you feel that our rebuttal has addressed the issues raised in your initial review, we respectfully ask that you consider whether an updated score would better reflect your current assessment of the paper.
> >
> > Thank you again for your time.
> >
> > Sincerely,
> >
> > *The Authors*

---

> ### Comment · Area_Chair_JVKk · 2025-08-07
>
> Dear reviewer GjjU,
>
> This is a kind reminder that reviewers should participate in discussions with authors. Please provide explanation about your updated (or not-updated) score and submit the “Mandatory Acknowledgement”. Please try to engage with the authors based on their rebuttal before the deadline. Your feedback is essential for the decision on this paper.
>
> Thank you!
> AC

---

### Official Review · Reviewer_hmwe · 2025-06-30

**Clarity:** 3
**Significance:** 4
**Originality:** 4
**Rating:** 5
**Confidence:** 3

**Summary:**

This paper aims to prove that a transformer with two layers within which contains a single head self-attention is sufficient to realise the k-th order in-context learning for any k. This is so far the tightest (2 layers and 1 head in each layer) theoretical solution against previous works where either more layers or more heads are needed.

**Questions:**

As suggested above, can you provide a more friendly description on how the MLP weights are constructed?

**Ethical Concerns:**

["NO or VERY MINOR ethics concerns only"]

**Final Justification:**

I’ve read the authors’ rebuttal and rebuttals to other reviews. The concerns have been addressed. Through this is a theoretical work and its setting is significantly simpler to real-world LLM corpus, it reveals the functions of both attentions and FFNs are interconnected on a common transformer structure. Therefore I would consider maintaining the score

**Limitations:**

yes

**Quality:**

4

**Strengths And Weaknesses:**

Strengths

- This paper provides a rigorous proof to the requirements of induction heads in the transformer architecture.
- This paper highlights the importance of the MLP module in a transformer layer. With a proper construction of MLP weights, the tightest results are obtained. Whereas previous works which requires more layers just ignored MLPs. This is also an important evidence suggesting that the MLPs in the pratical LLMs can play roles in structural abilities other than merely Key-Value stores.
- Empirical results in a simplified 1st-order markov analysis have accorded with the theorical prediction by a large margin.

Weaknesses

- Despite the simplification of the experiment as stated by the authors, the analysis still relies on synthesized data training other than induction heads in real world problems. The gaps are underexplored.
- In order to convey a concise and clear thought, this paper sacrifices much details. While most of them are acceptable, how the 1st layer MLP constructs $u_n$ is not easy to understand. Since this part is the key difference of this paper's constructional proof, it may worth more spaces (spared from the warm-up section) to describe (or even with more figures).

---

> ### Author Rebuttal · Authors · 2025-07-31
>
> We sincerely thank the reviewer for their thoughtful comments, valuable suggestions, and recognition of our work. Below, we provide detailed responses to each of the individual comments and questions.
>
> ---
>
> ## Comments on weaknesses
>
> **W2. In order to convey a concise and clear thought, this paper sacrifices much details. While most of them are acceptable, how the 1st layer MLP constructs $u_n$ is not easy to understand. Since this part is the key difference of this paper's constructional proof, it may worth more spaces (spared from the warm-up section) to describe (or even with more figures).**
>
> We respectfully apologize for not elucidating the result more clearly. Below, we try to provide a more accessible description of how the MLP functions in our construction. We will also make sure to elaborate on this in the revised version of the paper to improve the readability of this section.
>
> ---
>
> ## Questions
>
> **Q1. As suggested above, can you provide a more friendly description on how the MLP weights are constructed?**
>
> We clarify below how the MLP is used in our construction. We first provide a brief sketch of what the function of the MLP is and then dive into the details.
>
> **Brief Sketch:** In the first layer of attention, we obtain: $v_{n} = \sum_{i=1}^{\min(n, k)}  att_{n, n-i} e^x_{n-i}$ where $att_{n, n-i}$ is the attention weight at position $n - i$, when the token at position $n$ is in consideration, $k$ refers to the order of the markov chain, and $e^x_{n-i}$ is the value vector (one hot vector of the symbol associated with $x_{n-i}$ by construction) for token $x_{n-i}$. The MLP is then used to compute $u_n = \sum_{i=0}^{\min(n, k-1)} att_{n, n-i} e^x_{n-i}$ by isolating $e^x_{n-k}$ from $v_n$, subtracting it, and subsequently adding $e^x_{n}$. This is achieved via a weighted linear combination using the MLP and skip connections. Specifically, $e^x_{n}$ is introduced through skip connections, while the MLP plays a central role in isolating and removing $e^x_{n-k}$ with an appropriate weight.
>
>
> **More Details:** The first layer of the MLP is designed to extract the identity of the symbol that appeared $k$ steps before the current position $n$, i.e., to identify $x_{n-k}$. Recall that the attention mechanism in the previous stage computes: $v_n = \sum_{i=1}^{\min(n, k)}att_{n, n-i} e^x_{n-i}$ where each $att_{n, n-i} = \frac{3^i}{C_{attn}}$ (where $C_{attn}$ is the normalization constant of attention i.e. $C_{attn}=\sum_{i=1}^k 3^i$), and $e^x_{n-i}$ is the value vector for token $x_{n-i}$, represented as a one-hot vector by construction. Because the attention weights grow exponentially with $i$, the contribution from $x_{n-k}$ is strictly larger than the total contribution from all more recent symbols. That is:
>
> $att_{n, n-k} = \frac{3^k}{C_{attn}} > \sum_{i=1}^{k-1} att_{n, n-i} = \sum_{i=1}^{k-1} \frac{3^i}{C_{\texttt{attn}}} = \frac{3^{k} - 3}{2C_{\texttt{attn}}}$ ... (Inequality - 1)
>
> This inequality ensures that the coordinate in $v_n$ corresponding to $x_{n-k}$ is strictly larger than any combination of more recent contributions.
>
> *Layer-1:* To isolate $x_{n-k}$, the first MLP layer contains the one hot encodings corresponding to the vocabulary and hence one can take a dot product with this to $v_n$. After taking the dot product, the coefficients associated with each symbol can be distinctly identified. Since the vocabulary is embedded in the first layer, these coefficients naturally reside in orthogonal subspaces. The bias of the first layer is designed in a specific way to be able to isolate the symbol $x_{n-k}$ and this is done  by using the result shown in Inequality-1). Hence, the bias is a constant vector where each dimension has the same constant: $-\frac{3^{k} - 3}{2C_{attn}}$.  After adding this bias, the components corresponding to $x_{n-1}, x_{n-2}, \ldots$ become negative while the component for $x_{n-k}$ remains positive. Applying ReLU zeroes out the negative parts, leaving only the positive coordinate, corresponding to $x_{n-k}$, and after passing it through the norm layer, we obtain a one hot vector that that indicates the identity of $x_{n-k}$ (which is basically $e^x_{n-k}$ as its a one hot vector). Note that the first layer of the MLP is constructed in such a way that $e^x_{n-k}$ and $v_n$ are present in orthogonal subspaces.
>
> *Layer-2:* Since, the first layer extracts the one hot vector that indicates the identity of $x_{n-k}$, the second layer is then used to obtain $u_n$. The weights of the second layer are then designed to not only multiply  $e^x_{n-k}$, $v_n$ as well as $e^x_{n}$ (which is obtained via skip connections) with appropriate weights, but also to extract and aggregate them within the same subspace to obtain $u_n$. This is done by, designing the second layer wieghts of the MLP, that multiply $v_n$ with $3$ and $e^x_{n-k}$ with $-3^k/C_{attn}$ and finally $e^x_{n}$ with $1/C_{attn}$. Hence, once the weighted aggregation is computed, one can observe that
>
> $u_n = \frac{1}{C_{attn}} e^x_{n}  + 3 v_n -\frac{3^k}{C_{attn}} e^x_{n-k}$
>
> Thus, the first  two layers of the MLP layer performs the task of obtaining $u_n$. We will revise the manuscript to include a more accessible explanation in the main text.

---

> > ### Comment · Reviewer_hmwe · 2025-08-07
> > **Response to Rebuttal**
> >
> > Thanks for the explanation, I would like to maintain the score.

---

> ### Author Response · Authors · 2025-08-05
> **Discussion period closing soon**
>
> Dear Reviewer hmwe,
>
> We sincerely appreciate the time you have taken to provide valuable feedback for our work. As we are getting closer to the end of the discussion period, could you let us know if our responses above have adequately addressed your concerns? We remain at your disposal for any further questions.
>
> Sincerely, The Authors

---

> > ### Author Response · Authors · 2025-08-07
> > **Discussion Window Closing – Final Check-In**
> >
> > Dear Reviewer hmwe,
> >
> > We appreciate the time you've taken to provide feedback on our work. As the discussion period comes to a close, we would value your thoughts on whether our responses have addressed your concerns. We remain available to clarify any remaining points if needed.
> >
> > Thank you again for your time.
> >
> > Sincerely,
> >
> > *The Authors*

---

> ### Comment · Area_Chair_JVKk · 2025-08-07
>
> Dear reviewer hmwe,
>
> This is a kind reminder that reviewers should participate in discussions with authors. Please provide explanations about your updated (or not-updated) score and submit the “Mandatory Acknowledgement”. Your feedback is essential for the decision on this paper.
>
> Thank you!
> AC

---

### Official Review · Reviewer_2Q6Z · 2025-07-02

**Clarity:** 4
**Significance:** 3
**Originality:** 3
**Rating:** 5
**Confidence:** 4

**Summary:**

The paper shows how a k-th order induction head can be represented by a 2-layer transformer with a single attention head and MLP in each layer. The key idea is that the tokens can be expanded to a sufficiently high dimensional space where they are sparse (one-hot encoded, in this case) and attention weights can be used to combine them in superposition with an exponential moving average -- to represent k-grams. Prior constructions required two different moving averages (of window length k) shifted by one, but this paper shows how one can be used to derive the other since the token embedding at a timestep $x_{n-k}$ can be estimated using a suitable bias & ReLU on the moving average quantity $v_n$, and these can then be linearly combined to estimate $u_n$. In the second layer, the similarity between $u_T$ and $v_n$ is used to trigger an induction-based completion mechanism for $x_{T+1}$ copying $x_{n}$. While that shows that k-th order Markov chains are representable, it leaves open the question of whether they can actually be discovered during training by optimizing the cross-entropy loss on tokens. For this, the paper performs a perturbative analysis of stability of the proposed architecture weights (actually a subset, thereof) to show how optimization (at least within this subspace) converges to the anticipated weight patterns. The paper also points out an apparent "depth-width tradeoff" given that a k-gram model could be represented with a 3-layer transformer with a single attention head in each layer or alternatively a 2-layer transformer with two attention heads in the first layer.

**Questions:**

1. Why should the attention weights have the form $3^i$ instead of any other scale for the exponential? Is this precise number validated by experiment, or only the exponential behavior?
2. In Figure 2, layer 1 -- might it be interesting to have an additional plot depicting the attention map intensities on a logarithmic scale? Or maybe plot the variation across different diagnoals and inspect whether the weights are exponential as proposed (or whether there might be some other interesting pattern).
3. In Appendix E (fig 5/9/13), why not compute a pseudo-attention map for the first layer?
4. In Fig 2 Layer 2 (likewise figs 6-15) -- is it correct to say that the first "k" columns should actually be ignored when comparing with the pseudo-attention because the k-th order context is actually ill-defined for the first k tokens? If so, inspecting the "difference" between the "learned attention" and "Optimal"/"pseudo" attention might be misleading. The comparison might be less misleading (and also more favorable!) if you consider a sequence of length k+N and showed the attention matrix for a window of length N starting with the (k+1)-th token. Would you agree that's a more interesting plot?
5. Have you considered the implications for inference with quantized weights/activations? This proposal might point to specific patterns of failure, which could then be compared against LLMs for more circumstantial evidence on the representations within LLMs.

**Ethical Concerns:**

["NO or VERY MINOR ethics concerns only"]

**Final Justification:**

I am satisfied with the clarifications provided by the authors in the rebuttal. I believe that adding all those details to the final version of the paper would strengthen in -- making it more interesting and also more readable. On further reflection, I am happy to raise the score (from 4 to 5) for the promised improved/final version.

**Limitations:**

Yes

**Quality:**

4

**Strengths And Weaknesses:**

## Strengths
1. The proposal is conceptually elegant, including the minimality in components needed to represent k-th order Markov chains.
2. The experimental results seem closely matched with the theoretical model.
3. Explores not just in-principle representability, but also feasibility of discovery by training/optimization.
4. Such principled understanding will hopefully have a positive impact on architectures and training methods over time.

## Weaknesses
1. With this framework, the largest allowed "k" will be practically limited by the finite-precision of activations. This is even more of a problem when inference uses quantized activations. [addressed in rebuttal]
2. The observation regarding the depth-width tradeoff seems quite weak / superficial, and of unclear significance. It would be more interesting if the authors could flesh this out further. [partially addressed in rebuttal]

---

> ### Author Rebuttal · Authors · 2025-07-30
>
> We sincerely appreciate the reviewer’s thoughtful comments, valuable suggestions, and recognition of our work. Below, we provide detailed responses to each of the individual comments and questions.
>
> ---
>
> ## Comments about Weaknesses
>
> **W1. With this framework, the largest allowed "k" will be practically limited by this …  quantized activations.**
>
> We agree with the reviewers perspective. To clarify and provide additional context, the bit precision in our representation results follows prior work on constructions for the Markov in-context learning setting, namely Rajaraman et al. [1], where $\Omega(\log T + k)$ bits per parameter are sufficient to achieve an additive approximation error of $O(1/T)$. Where $T$ denotes the sequence length and $k$ denotes the order of the markov chain.
>
> **W2. The observation regarding the depth-width tradeoff seems quite weak / superficial, and of unclear significance. It would be more interesting if the authors could flesh this out further.**
>
> We agree that the most compelling result is the two-layer, single-head construction. The depth-width tradeoff presented in Section 3 serves two purposes: first, to provide context and continuity with prior work, thereby setting the stage for the subsequent constructive proofs; and second, to highlight a potentially interesting direction for the community. Specifically, the ability to trade depth for width may offer opportunities for inference-time speedups in other settings. While the two-head, two-layer construction may appear redundant for the specific task considered, we believe that exploring such architectural variations could prove valuable in broader applications.
>
> ---
>
> ## Questions
>
> **Q1. & Q2.   Why should the attention weights have the form ... only the exponential behavior? In Figure 2, layer 1 -- might it be interesting  ... whether the weights are exponential as proposed?**
>
> We address both the questions together below. We would like to note that there may exist multiple possible constructions demonstrating how two-layer, single-head transformers can represent induction heads due to the highly non-convex nature of the problem; our approach is one such construction. Our attention map design is motivated by two key observations:
> 1. *Empirical Observation*: In the first layer, we consistently observe that the attention weights increase monotonically along the first *k* lower diagonals and sharply decay to near-zero beyond that (figure - 2 in the paper).
> 2. *Prior Theoretical Construction*: This stems from a theoretical construction described in [1], where the attention pattern constructed in that paper is proportional to $2^i$ (a dyadic sum representation).
>
> Following the reviewer’s valuable suggestion to rigorously examine whether the diagonals in the first layer actually adhere to this pattern, we conducted further experiments. Specifically, we trained two models: one on a second-order Markov chain and another on a third-order Markov chain. For each model, we sampled 10 sequences and extracted the lower-*k* diagonals ($k=2$ or $k=3$) of the first-layer attention maps (note that $k$ refers to the order of the markov chain). For each diagonal and across the sampled sequences, we computed both the mean and the standard deviation of the attention values. Below, we summarize the results for clarity. *(Note: L-k refers to the k-th diagonal below the main diagonal.)*
>
> ### Table -1
>
> | Order | L-2 Mean ± StdDev| L-1 Mean ± StdDev| Ratio L-2 / L-1 |
> |-|-|-|-|
> | 2| 0.514 ± 0.0006| 0.256 ± 6.2455e-05| 2.00394|
>
> | Order | L-3 Mean ± StdDev| L-2 Mean ± StdDev| L-1 Mean ± StdDev| Ratio L-2/L-1 | Ratio L-3/L-1 |
> |-|-|-|-|-|-|
> | 3| 0.510 ± 0.0033| 0.3012 ± 0.00068039 |  0.1122 ± 0.00094251| 2.68| 4.53|
>
> As observed in our experiments (Table -1), the attention values do not strictly follow an exponential pattern of the form $3^i$. While our construction uses attention weights proportional to $3^i$ for illustrative purposes, we would like to emphasize that the proof of our representation result relies only on a much weaker condition. Specifically, for a token at position $n$, it suffices that the attention weight at position $n - k$, denoted $att_{n, n-k}$, is greater than both:
>
> 1. The sum of the attention weights from positions $n - 1$ to $n - k + 1$, and
> 2. All other attention weights.
>
> To elaborate further, in the first layer of attention, we obtain: $v_{n} = \sum_{i=1}^{\min(n, k)}  att_{n, n-i} e^x_{n-i}$ where $att_{n, n-i}$ is the attention weight at position $n - i$, when the token at position $n$ is in consideration, $k$ refers to the order of the markov chain, and $e^x_{n-i}$ is the value vector (one hot vector of the symbol associated with $x_{n-i}$ by construction) for token $x_{n-i}$. The MLP is then used to compute $u_n = \sum_{i=0}^{\min(n, k-1)} att_{n, n-i} e^x_{n-i}$ by isolating $e^x_{n-k}$ from $v_n$, subtracting it, and subsequently adding $e^x_{n}$. This is achieved via a weighted linear combination using the MLP and skip connections. Specifically, $e^x_{n}$ is introduced through skip connections, while the MLP plays a central role in isolating and removing $e^x_{n-k}$ from $v_n$ with an appropriate weight, which helps in obtaining $u_n$. Therefore, the key condition required for this construction to function, as also noted in the supplementary material (lines 555–560), is: $att_{n, n-k} > \sum_{i=1}^{k-1} att_{n, n-i} $ which guarantees that $e^{x}_{n-k}$ can be effectively isolated. Our experiments (Table -1) confirm that this condition holds consistently across different settings. We will update the paper to include these additional experimental results, as well as an explicit statement of the weaker, but sufficient, theoretical condition.
>
> **Q3. In Appendix E (fig 5/9/13), why not compute a pseudo-attention map for the first layer?**
>
> The pseudo-attention map represents the idealized attention pattern required to approximate a conditional $k$-gram estimator. In other words, it highlights the tokens that need to be attended to in order to estimate the transition probabilities of a $k$-th order Markov chain, effectively functioning as a $k$-th order induction head. Please refer to Figure 1b (page 3) of the main paper for an illustration. While it is possible to compute the pseudo-attention map for the first layer as well, we note that the role of the first-layer head is primarily to copy the previous $k$ tokens (i.e., from positions $n-1$ to $n-k$) for the $n$-th token. For this reason, we believe that displaying the pseudo-attention map for the first layer may not offer significant additional insight. However, a potentially useful comparison would be to show the attention map of our theoretical construction in the first layer alongside the learned attention map. We agree that this could help clarify how closely the learned mechanism approximates the idealized behavior. We will include this plot in the final version of the paper.
>
> **Q4. In Fig 2 Layer 2 (likewise figs 6-15) -- is it correct ...  a more interesting plot?**
>
>  It is indeed correct that the pseudo attention is ill-defined for the first k tokens, and accordingly, these tokens must be excluded when comparing it with the learned attention. Furthermore, as the sequence length increases, the estimation becomes more accurate. We also greatly appreciate the thoughtful suggestion to consider sequences of length $k + N$ and to present the attention matrix for a window of length $N$ starting from the $(k + 1)$-th token. We will incorporate this into revised version of the paper.
>
> **Q5. Have you considered the implications for inference ... against LLMs for more circumstantial evidence on the representations within LLMs.**
>
> This is indeed an interesting direction, and while our primary focus was on deriving the tightest characterization of the relationship between transformer depth and the Markov order required to solve the Markov in-context learning task, we agree that exploring quantized inference could offer valuable insights. For additional context, our representational results adopt the formulation from [1], which shows that $\Omega(\log T + k)$ bits per parameter are sufficient to achieve an additive approximation error of $O(1/T)$, where $T$ represents the sequence length and $k$ is the Markov order. That said, we did conduct a preliminary analysis of weight and activation quantization on a trained model using a second-order Markov chain with sequence length 32. We computed the Frobenius norm between the pseudo attention map (optimal attention map) and the attention map of the second layer of the quantized model across 10 random sequences, and report the mean and variance below:
>
> | Quantization (bits) | Frobenius Norm (Mean ± Std) |
> |-|-|
> | 2                   | 4.6108 ± 0.7190             |
> | 4                   | 4.5842 ± 0.9826             |
> | 8                   | 3.8615 ± 0.4590             |
> | 32 (No quantization)| 3.7700 ± 0.4725 |
>
> These results suggest that 8-bit quantization yields performance that is quite close to full-precision (32-bit), at least in terms of preserving the attention structure as captured by the Frobenius norm, which is indeed an interesting implication and also seems to match the theoretical bounds. Due to the updated NeurIPS policy, we are unable to share the attention maps directly. However, we will include these quantization results as well as the attention maps and their implications in the revised version of the paper.
>
> ---
>
> **References**
>
> [1] Rajaraman, Nived, et al. "Transformers on markov data: Constant depth suffices." NeurIPS 37

---

> > ### Author Response · Authors · 2025-08-05
> >
> > Dear Reviewer 2Q6Z,
> >
> > We sincerely appreciate the time you have taken to provide valuable feedback for our work. As we are getting closer to the end of the discussion period, could you let us know if our responses above have adequately addressed your concerns? We remain at your disposal for any further questions.
> >
> > Sincerely, The Authors

---

> > ### Comment · Reviewer_2Q6Z · 2025-08-05
> >
> > Apologies for the delayed comment; I took a few days to think over what you had written. I appreciate the clarifications and additional analyses/experiments, and believe that adding these to the final version of the paper would definitely strengthen it.
> >
> > Regarding W2: While I am still not fully satisfied about the significance of the observation, I accept that it is worth stating explicitly, as you have in the paper -- if only to spur future work. With that motivation, I suggest that you also contextualize the claim by adding to the conclusions section (a suitably adapted version of) your response to W2.

---

> ### Author Response · Authors · 2025-08-05
>
> Dear Reviewer 2Q6Z,
>
> We thank you for your response and are grateful that our responses appear to have addressed most of your comments and concerns. We also appreciate your suggestion regarding the clarification of W2 in the conclusions, which we will incorporate into the revised version of the paper.
>
> We respectfully ask whether you might consider updating your score to more accurately reflect your current evaluation of our submission. Thank you once again for your time and thoughtful feedback. We remain at your disposal should you have any further questions or require additional clarification.
>
> Sincerely, The Authors

---

### Comment · Area_Chair_JVKk · 2025-08-06

Dear Reviewer,

Please review the authors’ rebuttal, engage in discussions, and finalize your scores. Please write explanations about your updated (or not-updated) scores and submit the Mandatory Acknowledgement. If you have done so, thank you!

Your effort is greatly appreciated for the conference.
Thanks, AC

---

### Note · Authors · 2025-08-15

Dear AC and Reviewers,

We thank you for your constructive feedback, which has strengthened our work. We show that a two-layer, single-head transformer can represent any conditional k-gram, challenging the view that deeper or multi-head architectures are required for in-context learning of higher-order Markov processes. This is the tightest known construction, and we also make partial progress toward understanding the learning dynamics.

---
### Key Strengths Highlighted:
**Conceptual elegance and minimality, and novel MLP role (2Q6Z, hmwe, GjjU):** Achieves k-th order Markov chain representation with the fewest known layers/heads, and the MLP performs structural functions beyond key–value storage.

**Theory–experiment alignment (2Q6Z, hmwe):** Theoretical construction closely matches empirical observations.

**Clarity of exposition (2Q6Z, GjjU, VB7N):** Well written and clearly presented results.

---
### Key Concerns Addressed:

**Rev. 2Q6Z:** Questioned theory–experiment agreement; we added supporting experiments and clarified weaker theoretical conditions.

**Rev. hmwe:** Requested a more accessible proof; we expanded explanations and will further improve clarity in the final version.

**Rev. GjjU:**
1. We have clarified any misunderstandings regarding our reliance on the universal approximation theorem and parameter count; our approach does not use that theorem, and the count remains correct.
2. We addressed the suggestion that our experiments used handcrafted initializations by noting that all experiments employ standard initializations, yielding results that align closely with theory, as also observed by 2Q6Z and hmwe.
3. Although our work is primarily theoretical, we also evaluated the model on datasets that may exhibit real-world-like distributions to better understand its performance in such settings.

**Rev. VB7N:**
1. We have clarified misunderstandings regarding the lower bound; we explicitly cite prior work that establishes it and reinforced this point with additional experiments consistent with that theory.
2. On theory–experiment alignment, we confirmed that all experiments use standard initializations, ensuring direct comparability to the theoretical model.
3. Although testing on noisy data is not central to our contributions, we conducted such experiments and found consistent degradation under different noise schemes, illustrating how the architecture behaves in noisy environments.

Thank you once again.

Sincerely,
The Authors

---

### Decision · Program_Chairs · 2025-09-17

**Decision:**

Accept (spotlight)

**Comment:**

This work showed that a two-layer transformer with a single head can realise any conditional k-gram for in-context learning for any k, which is so far the tightest theoretical construction. It also provided experimental results that support the theoretical analysis. It also explores the feasibility of discovery by training dynamics.

The discussions have helped clarify/address a few concerns from the reviewers, e.g., clarifying the experimental setup, adding experiments. While the setting is simpler than real-world scenarios, the novel and tight theoretical construction and the empirical support are solid contributions to the community. Please incorporate the clarifications/revisions/added experiments into the final version if accepted.